# Expansion of outer cortical CUX2 neurons requires adaptations for DNA repair

Wenlong Xia[1], Laura Morcom[2,3], Zhaoyang Xu[2,3,4,5], I-Ling Lu[6], Qing Wang[7], Kimberly K. Hoi[1], Mingming Wei[1], Keying Zhu[1], Gregory Jordan[2,3], Xiao-Yan Tang[1], Julio Gonzalez-Maya[1], Vanesa S. Mattera[1], Sophia M. Panigrahi[1], Riki Kawaguchi[7], Ben Emery[8], Santos J. Franco[9], Daniel H. Geschwind[7], Brian Popko[10], David H. Rowitch[2,3,4,5,6 ✉] & Stephen P. J. Fancy[1 ✉]

During mammalian evolution, excitatory neurons in upper cortical layer 2 and layer 3 (L2/3) have shown a disproportionate expansion compared with other layers[1–4]. Replicative expansion of cortical neural progenitors is associated with considerable oxidative DNA damage. Here we show that activating transcription factor 4 (ATF4) has roles as a critical regulator of the DNA damage response, directly activating components of double-stranded DNA repair, including CIRBP, UBA52 and EBF1. Notably, pan-cortical knockout (*Emx1-Cre;Atf4*[fl/fl]) demonstrates that ATF4 is required specifically for the development of upper layer 2/3 neurons, marked by the expression of cut-like homeobox 2 protein, CUX2. ATF4 functions to repair DNA damage and attenuate cell death of embryonic radial glial progenitors in a p53-dependent manner. In particular, we show that cold inducible RNA-binding protein (CIRBP) is a transcriptional target of ATF4 that is required for normal phosphorylation of the key double-strand DNA repair factor ataxia telangiectasia mutated (ATM). These findings establish that ATF4 is an essential regulator of the DNA damage response. They further indicate that there are extraordinary requirements for DNA repair after replicative stress in CUX2[+] neurons during mammalian brain development.

Corticogenesis during mammalian embryonic development requires a massive expansion of neural progenitors (NPs) and their subsequent development into the mature neuron subtypes that make up the six neocortical layers[1]. During mammalian evolution, upper cortical L2/3 show a disproportionate expansion compared with other layers[2], implying that there are higher-order functions as well as increased replicative stress during development. In the mature cortex, these upper cortical layers (L2/3) are occupied predominantly by pyramidal projection neurons that send and receive mainly cortico-cortical connections[3,4], and the extent and complexity of these interhemispheric connections is closely associated with higher cognitive functions in mammals[5]. CUX2 is a CUT-homeodomain transcription factor that is expressed during early development in a subset of cortical neuronal progenitors, and remains an identity marker of these upper L2/3 projection neurons in the mature cortex[6,7]. It has been suggested that these L2/3 neurons are fate restricted, even in early cortical development, and that a radial glial (RG)-cell lineage is intrinsically specified to generate only CUX2[+] upper-layer neurons independent of niche and birth date[8], but this has been contested[9,10]. Several studies now suggest that these L2/3 CUX2[+] cortical neurons are fundamentally affected in a multitude of human disorders. Loss of CUX2[+] neurons has been reported in multiple sclerosis[11,12], head trauma in young human athletes[13], Alzheimer's disease[14] and frontotemporal dementia[15]. Dysfunction of L2/3 neurons is also consistently linked to temporal lobe epilepsy[16] and schizophrenia[17], and they show most pathway dysregulation in human autism[18]. This indicates an intrinsic vulnerability to various types of central nervous system dysfunction. Important aspects of CUX2[+] neuron basic biology, and the reason for this vulnerability, are not understood.

During corticogenesis, the rapid hyperproliferation of NPs in the subventricular zone demands a huge energy supply, and the reactive oxygen species (ROS) generated by mitochondrial respiration and cellular metabolism[19] are a main cause of DNA damage in proliferating NPs[20]. The preservation of genome stability is crucial[21–23], because transmission of genetic errors during early progenitor expansion leads to a variety of severe neurodevelopmental neurological conditions and neurodegeneration. Eukaryotic cells, including NPs, have evolved a finely tuned signalling network called the DNA damage response (DDR)[24,25], which detects DNA damage and then integrates cell-cycle control with DNA repair, or triggers apoptosis through p53-induced cell-cycle arrest in cells in which the DNA cannot be repaired. The DDR has evolved repair mechanisms specific for many types of DNA lesion, the most common of which during neurogenesis are DNA double-strand

[1]Division of Neuroimmunology and Glial Biology, Department of Neurology, University of California, San Francisco, San Francisco, CA, USA. [2]Department of Paediatrics, University of Cambridge, Cambridge, UK. [3]Cambridge Stem Cell Institute, University of Cambridge, Cambridge, UK. [4]Department of Pediatrics, Cedars Sinai Guerin Children's, Los Angeles, CA, USA. [5]Department of Neurosurgery, Cedars Sinai Guerin Children's, Los Angeles, CA, USA. [6]Institute for Regenerative Medicine, University of California, San Francisco, San Francisco, CA, USA. [7]Department of Neurology, Semel Institute for Neuroscience and Human Behavior, David Geffen School of Medicine, University of California, Los Angeles, Los Angeles, CA, USA. [8]Jungers Center for Neurosciences Research, Oregon Health & Science University, Portland, OR, USA. [9]Department of Pediatrics, University of Colorado Anschutz School of Medicine, Aurora, CO, USA. [10]Department of Neurology, Feinberg School of Medicine, Northwestern University, Chicago, IL, USA. ✉e-mail: dhr25@cam.ac.uk; stephen.fancy@ucsf.edu

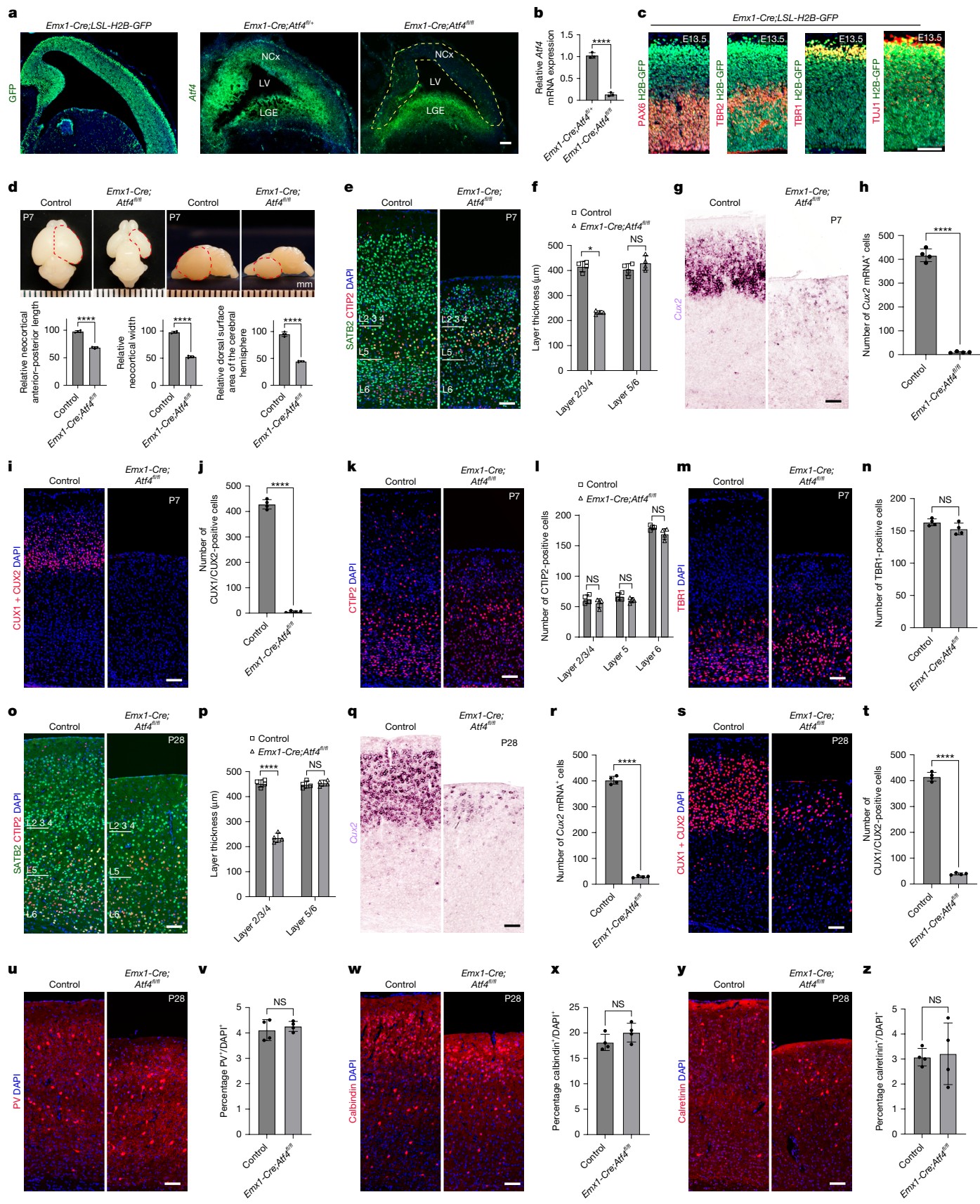

**Fig. 1 | See next page for caption.**

**Fig. 1 | Selective loss of CUX2⁺ upper-layer neurons in the ATF4-ablated mouse cortex. a**, Representative sagittal sections of E13.5 *Emx1-Cre;LSL-H2BGFP* embryos showing GFP expression to delineate the *Emx1-Cre* expression pattern. *Atf4* mRNA expression was visualized by RNAscope in E13.5 *Emx1-Cre;Atf4^fl/fl* embryos and controls; yellow dotted lines indicate *Atf4* knockout throughout the neocortex (NCx). LV, lateral ventricle; LGE, lateral ganglionic eminence. Scale bar, 100 µm. **b**, Quantification of *Atf4* mRNA levels in E13.5 *Emx1-Cre;Atf4^fl/fl* embryos relative to controls (*n* = 3 mice per genotype). **c**, Co-immunostaining of PAX6, TBR2, TBR1 and TUJ1 with GFP in E13.5 *Emx1-Cre;LSL-H2BGFP* cortex (*n* = 3 littermate mice) Scale bar, 50 µm. **d**, Representative images of P7 forebrains from *Emx1-Cre;Atf4^fl/fl* mice and littermate controls; dashed lines indicate the forebrain outline. Scale bar, 1 mm. Bottom panels show quantification of neocortical anterior–posterior length, width and dorsal surface area (*n* = 4 mice per genotype). **e,f**, Representative images of SATB2, CTIP2 and DAPI staining (**e**) and quantification of cortical layer 2/3/4 and layer 5/6 thickness (**f**) in P7 *Emx1-Cre; Atf4^fl/fl* and control cortices (*n* = 4 mice per genotype). Scale bar, 100 µm. **g,h**, *Cux2* mRNA ISH in P7 *Emx1-Cre;Atf4^fl/fl* and control cortices (**g**) and quantification of *Cux2⁺* neurons within a 500 × 500-µm² cortical region (**h**) (*n* = 4 mice per genotype). Scale bar, 100 µm. **i,j**, Immunostaining of CUX1 and CUX2 in P7 *Emx1-Cre;Atf4^fl/fl* and control cortices (**i**) and quantification of CUX1 and CUX2 double-positive neurons within a 500 × 500-µm² cortical region (**j**) (*n* = 4 mice per genotype). Scale bar, 100 µm. **k,l**, CTIP2 immunostaining in P7 *Emx1-Cre;Atf4^fl/fl* and control cortices (**k**) and quantification of CTIP2⁺ neurons in layers 2/3/4, 5 and 6 within a 500 × 500-µm² cortical region (**l**) (*n* = 4 mice per genotype). Scale bar, 100 µm. **m,n**, TBR1 immunostaining in P7 *Emx1-Cre;* *Atf4^fl/fl* and control cortices (**m**) and quantification of TBR1⁺ neurons within a 500 × 500-µm² cortical region (**n**) (*n* = 4 mice per genotype). Scale bar, 100 µm. **o,p**, SATB2, CTIP2 and DAPI staining in P28 *Emx1-Cre;Atf4^fl/fl* and control cortices (**o**) and quantification of cortical layer thickness (**p**) (*n* = 4 mice per genotype). Scale bar, 100 µm. **q,r**, *Cux2* mRNA ISH in P28 *Emx1-Cre;Atf4^fl/fl* and control cortices (**q**) and quantification of *Cux2⁺* neurons within a 500 × 500-µm² cortical region (**r**) (*n* = 4 mice per genotype). Scale bar, 100 µm. **s,t**, Immunostaining of CUX1 and CUX2 in P28 *Emx1-Cre;Atf4^fl/fl* and control cortices (**s**) and quantification of CUX1 and CUX2 double-positive neurons within a 500 × 500-µm² cortical region (**t**) (*n* = 4 mice per genotype). Scale bar, 100 µm. **u,v**, Parvalbumin (PV) staining in P28 *Emx1-Cre;Atf4^fl/fl* and control cortices (**u**) and percentage of PV⁺ neurons relative to total DAPI⁺ cells (**v**) (*n* = 4 mice per genotype). Scale bar, 100 µm. **w,x**, Calbindin staining in P28 *Emx1-Cre;Atf4^fl/fl* and control cortices (**w**) and percentage of calbindin⁺ neurons relative to total DAPI⁺ cells (**x**) (*n* = 4 mice per genotype). Scale bar, 100 µm. **y,z**, Calretinin staining in P28 cortices (**y**) and percentage of calretinin⁺ neurons relative to total DAPI⁺ cells (**z**) (*n* = 4 mice per genotype). Scale bar, 100 µm. Data are presented as mean ± s.d. (**b,d,f,h,j,l,n,p,r,t,v,x,z**). Statistical significance was determined by two-sided unpaired *t*-test (**b,d,h,j,n,r,t,v,x,z**) or two-sided multiple *t*-tests (**f,l,p**). *P* < 0.0001 (**b**), *P* < 0.0001 (**d**, left), *P* < 0.0001 (**d**, middle), *P* < 0.0001 (**d**, right), *P* = 0.000003 (**f**, layer 2/3/4), *P* = 0.210781 (**f**, layer 5/6), *P* < 0.0001 (**h**), *P* < 0.0001 (**j**), *P* = 0.24373 (**l**, layer 2/3/4), *P* = 0.125173 (**l**, layer 5), *P* = 0.062794 (**l**, layer 6), *P* = 0.1003 (**n**), *P* = 0.000002 (**p**, layer 2/3/4), *P* = 0.545881 (**p**, layer 5/6), *P* < 0.0001 (**r**), *P* < 0.0001 (**t**), *P* = 0.5465 (**v**), *P* = 0.1662 (**x**), *P* = 0.8374 (**z**); NS, not significant.

breaks (DSBs)[26]. ATM is a central regulator of DNA DSB repair and, once activated by DSBs, it phosphorylates many downstream factors to initiate the cascade of double-strand repair[27]. Phosphorylation of ATM at serine 1981 is required for the sustained retention of ATM at DSBs, and also for its ability to phosphorylate its downstream targets after DNA damage[28]. However, important gaps remain in our understanding of the factors that govern neural-progenitor genome integrity, and a more complete appreciation of how DNA damage-signalling pathways promote neural development is needed to understand the formation of the human brain.

ATF4 is a multifunctional transcription regulatory protein in the basic leucine zipper superfamily[29]. Its leucine zipper is highly amenable to heterodimerization with those of other basic leucine zipper family members, and the in vivo effects of ATF4 are largely, if not entirely, mediated by ATF4 heterodimers[30]. ATF4 participates in a variety of cellular responses to specific environmental stresses and derangements of the intracellular environment, and is a key transcription factor mediator of both the integrated stress response (ISR)[31] and the unfolded protein response (UPR)[32]. There is little evidence, however, to suggest that it is involved in the response to another type of cell stress, namely, genome instability. ATF4 has been shown to induce the DNA-repair enzyme APE1 in cultured fibroblasts in response to arsenite-mediated toxicity[33]. The PERK/ATF4 arm of the UPR has also been shown, in the context of tumour hypoxia, to induce the RNA/DNA helicase SETX, which is a replication stress-dependent DNA damage response[34]. Here we demonstrate new roles for ATF4 during neurogenesis as a crucial direct transcriptional regulator of the DDR in a select NP population in the developing brain. We find disrupted phosphorylation of ATM and catastrophic DNA damage and cell death of embryonic precursors of upper-layer neurons in *Emx1-Cre;Atf4* conditional-knockout (cKO) mice, indicating a selective requirement for ATF4 function in the DDR.

## Selective vulnerability of CUX2⁺ upper-layer neurons

L2/3 CUX2⁺ neurons are selectively vulnerable in multiple sclerosis[11,12] and demonstrate considerable dysregulation of *ATF4* expression. Given that ATF4 can function in different ways in cellular adaptation and has been suggested to be a redox-regulated prodeath transcriptional activator[35], we were initially interested in discovering its contribution to CUX2⁺ developmental vulnerability. In the mouse, embryonic cortical NPs divide rapidly between embryonic day 11.5 (E11.5) and birth (postnatal day 0, P0), giving rise to six neocortical layers[1]. We used a pan-cortical *Emx1-Cre* line, which becomes active in early embryogenesis and is expressed in all cortex progenitors and their progeny (Fig. 1c and Extended Data Fig. 1), to conditionally delete *Atf4* from all cortical NPs. The cortex in *Emx1-Cre;Atf4^fl/fl* mice showed a large reduction in volume compared with controls at P7 (Fig. 1d). Surprisingly, despite expression of *Emx1-Cre* throughout the cortex, and ablation of *Atf4* from all cortical NPs (Fig. 1a,b), we observed considerable thinning in cortex L2/3 at P7, whereas the thickness of other layers was unaltered (Fig. 1e,f). Neuronal occupants of L2/3 include CUX2⁺ excitatory neurons and interneurons. In situ hybridization (ISH) indicated that CUX2⁺ excitatory neurons were eliminated in *Emx1-Cre;Atf4^fl/fl* mice compared with controls (Fig. 1g,h), and this was confirmed by immunohistochemistry using anti-CUX1/CUX2 antibodies (Fig. 1i,j). By contrast, deep-layer CTIP2⁺ and TBR1⁺ neuron populations were intact (Fig. 1k–n). We confirmed the persistence of this L2/3 neuron-specific phenotype at P28 in *Emx1-Cre;Atf4^fl/fl* mice (Fig. 1o–t). Interneurons in different layers were unaffected at P28, including those expressing parvalbumin (Fig. 1u,v), calbindin (which predominantly occupies L2/3; Fig. 1w,x) and calretinin (Fig. 1y,z). All analyses were done in the primary somatosensory cortex, but the same phenotype was seen throughout the cortex at different rostral and caudal levels (Extended Data Fig. 2). These results identified specific requirements for *Atf4* function in L2/3 (CUX2⁺) excitatory neuron precursors.

## CUX2⁺ neuron loss in *Atf4* cKO due to DNA damage

We next investigated the mechanism for CUX2⁺ neuron elimination in the absence of ATF4 function. *Atf4* mRNA is expressed in the NP zone of E11 and E13 embryonic mouse telencephalon adjacent to the ventricle; by E16–P0, expression expands to the outer cortical layers (Fig. 2a). Similarly, *ATF4* mRNA is expressed in human cortex ventricular-zone NPs expressing PAX6 at gestational weeks 15 and 17 (Fig. 2b,c). At later embryonic times, *Atf4* mRNA seems to be more widely expressed by different neuronal subtypes (Extended Data Fig. 3a).

As shown in Fig. 2d–f, from early in corticogenesis we observed massive accumulation of DNA damage (a roughly 30-fold increase, γH2AX) and cell death (cleaved caspase 3 (CC3)) in NPs of *Emx1-Cre;Atf4^fl/fl* mice compared with controls. There was also a considerable reduction in

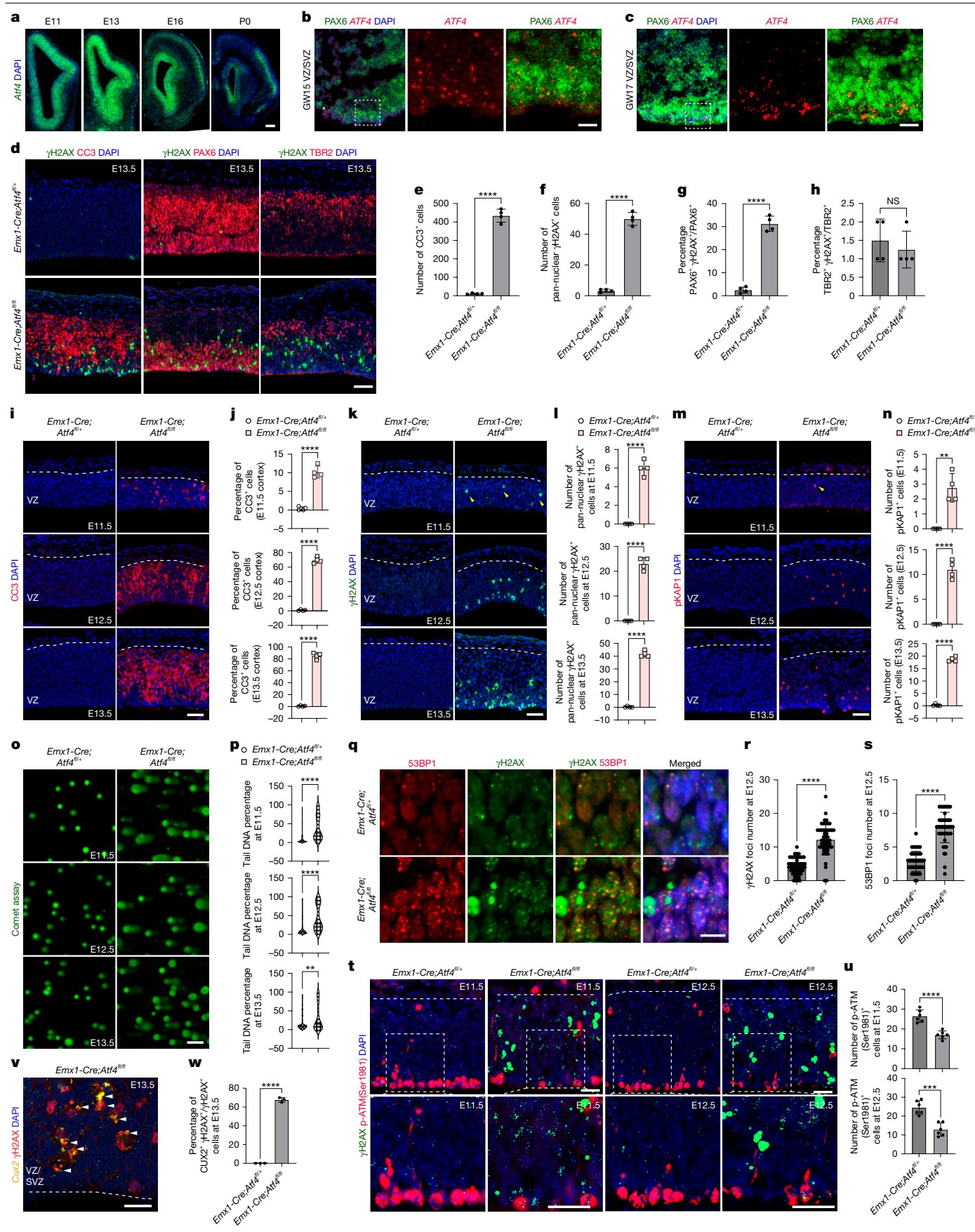

**Fig. 2** | See next page for caption.

**Fig. 2 | Selective loss of CUX2⁺ neurons in the absence of ATF4 originates early during corticogenesis, owing to impaired DNA damage repair.**
**a**, Representative images of *Atf4* RNAscope staining in mouse cortices at E11, E13, E16 and P0 ($n = 3$ littermates). Scale bar, 100 μm. **b,c**, *ATF4* RNAscope and PAX6 co-staining of human cortex at gestational week (GW) 15 (**b**) and 17 (**c**). VZ, ventricular zone; SVZ, subventricular zone ($n = 1$ for each time point). Scale bars, 10 μm. **d–h**, Co-immunostaining for CC3, PAX6 and TBR2 with γH2AX, in E13.5 *Emx1-Cre;Atf4[fl/fl]* and control mouse cortices (**d**), with quantification of CC3⁺ cell number (**e**), pan-nuclear γH2AX⁺ cell number (**f**) within a $200 \times 200$-μm² cortical region, and percentage of PAX6⁺γH2AX⁺ relative to total PAX6⁺ (**g**) and TBR2⁺γH2AX⁺ relative to total TBR2⁺ (**h**) cells ($n = 4$ mice per genotype). Scale bar, 50 μm. **i–n**, Immunostaining for CC3 (**i**), γH2AX (**k**) and pKAP1 (**m**), in E11.5, E12.5 and E13.5 *Emx1-Cre;Atf4[fl/fl]* and control cortices, with corresponding quantification of the percentage of CC3⁺ (**j**), pan-nuclear γH2AX⁺ cell number (**l**) and pKAP1⁺ cell number (**n**) within a 200-μm bin ($n = 4$ mice per genotype). Scale bars, 50 μm. **o,p**, Comet assay of acutely isolated neural stem cells (NSCs) from E11.5, E12.5 and E13.5 *Emx1-Cre;Atf4[fl/fl]* and control cortices (**o**) and quantification of tail DNA percentage (**p**) ($n = 4$ mice per genotype, with at least 58 cells per group). Scale bar, 100 μm. **q–s**, Immunostaining for γH2AX and 53BP1 in E12.5 *Emx1-Cre;Atf4[fl/fl]* and control cortices (**q**) and quantification of γH2AX (**r**) and 53BP1 (**s**) foci per cell ($n = 4$ mice per genotype, 50 cells per genotype). Scale bar, 10 μm. **t,u**, Immunostaining for p-ATM(Ser1981) (**t**) and quantification of p-ATM⁺ cells in a 200-μm-wide VZ region (**u**) at E11.5 and E12.5 in *Emx1-Cre;Atf4[fl/fl]* and control cortices ($n = 6$ mice per genotype). Scale bar, 25 μm. **v,w**, Co-staining of *Cux2* RNAscope and γH2AX in *Emx1-Cre;Atf4[fl/fl]* (**v**) and percentage of *Cux2*⁺γH2AX⁺ cells relative to total γH2AX⁺ (**w**) at E13.5 ($n = 4$ mice per genotype). Scale bar, 20 μm. All bar graphs represent mean ± s.d. (**e–h,j,l,n,p,r,s,u** and **w**). Statistical significance by two-sided unpaired t-test (**e–h,j,l,n,p,r,s,u,w**). $P < 0.0001$ (**e**), $P < 0.0001$ (**f**), $P < 0.0001$ (**g**), $P = 0.537$ (**h**), $P < 0.0001$ (**j**, E11.5), $P < 0.0001$ (**j**, E12.5), $P < 0.0001$ (**j**, E13.5), $P < 0.0001$ (**l**, E11.5), $P < 0.0001$ (**l**, E12.5), $P < 0.0001$ (**l**, E13.5), $P = 0.0012$ (**n**, E11.5), $P < 0.0001$ (**n**, E12.5), $P < 0.0001$ (**n**, E13.5), $P < 0.0001$ (**p**, E11.5), $P < 0.0001$ (**p**, E12.5), $P = 0.0011$ (**p**, E13.5), $P < 0.0001$ (**r**), $P < 0.0001$ (**s**), $P < 0.0001$ (**u**, E11.5), $P = 0.0002$ (**u**, E12.5), $P < 0.0001$ (**w**).

PAX6⁺ RG cells and TBR2⁺ intermediate progenitor cells (Extended Data Fig. 3b–d), and a roughly 30-fold increase in the percentage of PAX6⁺ NP cells with DNA damage (Fig. 2d,g). By contrast, the percentage of TBR2⁺ intermediate progenitor cells with DNA damage were similar (Fig. 2d,h). EdU labelling showed no significant difference in the proliferation of PAX6⁺ or TBR2⁺ cells between E13.5 *Emx1-Cre;Atf4[fl/fl]* mice and control groups (Extended Data Fig. 3e,f). These findings indicate sensitivity to DNA damage in primitive (PAX6⁺ TBR2⁻) NPs of mouse telencephalon.

The cortical phenotype in these mice begins as early as E11.5 and continues through E13.5, marked by significant increases in cell death (CC3; Fig. 2i,j) and pyknosis, indicative of chromatin condensation during apoptosis (Extended Data Fig. 3g,h). Furthermore, evidence of DNA damage is demonstrated by staining for γH2AX and pKAP1 (Fig. 2k–n). We performed comet assays with acutely isolated NP at three time points, all of which showed a significant increase in DNA damage in neural progenitors lacking *Atf4* (Fig. 2o,p). The loss of *Atf4* was associated with DNA damage-induced apoptosis in newly born NPs. This was supported by a significant overlap between cells positive for pan-nuclear γH2AX and proliferating cell nuclear antigen (PCNA) from E11.5 to E13.5 (Extended Data Fig. 3i,j). By contrast, no significant difference was observed in the ratio of Ki67⁺PAX6⁺ cells between *Emx1-Cre;Atf4[fl/fl]* and control mice from E11.5 to E13.5 (Extended Data Fig. 3k,l). Pan-nuclear staining of γH2AX and pKAP1 represent cells undergoing apoptosis because of overwhelming DNA damage[36]. To assess DNA damage more accurately, we also assessed the number of γH2AX punctae in the nuclei of cells without pan-nuclear staining. This showed significant increases in punctae for γH2AX and 53BP1 (which colocalized) in NPs at E12.5 from *Emx1-Cre;Atf4[fl/fl]* mice compared with controls (Fig. 2q–s).

ATM is a central kinase in the cellular response to DNA DSBs[27]. On detection of DSBs, ATM is activated and phosphorylated at Ser1981, a modification required for its stable retention at DNA breaks and for phosphorylation of downstream repair targets[28]. We observed a very significant decrease at E11.5 and E12.5 in the number of RG adjacent to the ventricle with phosphorylated ATM(Ser1981) in mutants compared with controls (Fig. 2t,u). The proliferation and the number of RG were unaltered in the absence of *Atf4* (Extended Data Fig. 3m–p), but this loss of ATM phosphorylation was associated with greatly increased γH2AX punctae. This indicates that a subset of RG have disrupted ATM phosphorylation in *Emx1-Cre;Atf4[fl/fl]* mice. It also indicates a fundamental failure at the initiation of DSB DNA repair in NPs lacking *Atf4*.

As described above, later postnatal times show a selective elimination of *Cux2*-expressing L2/3 neurons in *Emx1-Cre;Atf4[fl/fl]* mice. It has been suggested that L2/3 neurons are fate restricted, and that an RG cell lineage is intrinsically specified to generate only CUX2⁺ upper-layer neurons[8], but this concept remains controversial[9,10]. In support of an RG cell lineage intrinsically specified to generate CUX2⁺ neurons, we found expression of *Cux2* mRNA in a subset of NPs at E13.5, and a preferential accumulation of DNA damage in these compared with other neural progenitors (Fig. 2v,w). Together with the subsequent selective loss of L2/3 neurons at later embryonic times, these results indicate that a subset of cortical NPs, destined to become CUX2⁺ L2/3 excitatory neurons, have a selective requirement for *Atf4* to mitigate DNA damage.

## CUX2⁺ neuron loss in *Atf4* cKO is p53 dependent

DNA damage-induced apoptosis is often p53 dependent[37] in the nervous system. We found a massive induction of p53 expression at E12.5 associated with cell death in the cortices of *Emx1-Cre;Atf4[fl/fl]* mice, compared with controls (Fig. 3a,b). We also found multiple p53 target or p53-related genes upregulated in NPs in the absence of *Atf4*, such as *Cdkn1a*, *Eda2r*, *Ccng1*, *Zmat3*, *Ano3* and *Dglucy* (Fig. 3c). To assess whether cell death in NPs after *Atf4* loss of function is p53 dependent, we used a *p53[−/−]* mouse, in which apoptosis is prevented. Comparison of *Emx1-Cre;Atf4[fl/fl];p53[+/−]* mice with *Emx1-Cre;Atf4[fl/fl];p53[−/−]* mice at P7 showed remarkably complete rescue of upper cortical layer thinning (Fig. 3d,e), in particular regarding CUX2⁺ neurons (Fig. 3f–i). Analysis at P28 in *Emx1-Cre;Atf4[fl/fl];p53[−/−]* mice confirmed perdurance of normalized cortical size and layer thickness similar to that of control animals (Fig. 3j–l). We also found very high levels of DNA damage persisting in NPs of *Emx1-Cre;Atf4[fl/fl];p53[−/−]* mice (Fig. 3m–q), demonstrating that the elimination of CUX2⁺ cells lacking *Atf4* is p53 dependent, with apoptosis following the accumulation of DNA damage.

## ATF4 loss causes global alterations of DDR in NPs

ATF4 is a key transcription factor mediator of the ISR and the UPR, as well as of metabolic control pathways[38]. We performed quantitative PCR of whole cortex samples from E13.5 *Atf4[fl/fl]* (control) and *Emx1-Cre;Atf4[fl/fl]* mice, which showed significant loss in expression of *Atf4*, but did not show alterations in key components or target genes of the UPR and ISR (Extended Data Fig. 4a). This indicated that the function of ATF4 in the repair of DNA damage may be independent of its roles in the ISR and UPR.

We performed single-nucleus RNA sequencing (snRNA-seq) on E11.5 cortex from *Emx1-Cre;Atf4[fl/fl]* mice versus controls (Fig. 4a–c). Gene-set score analysis of the UPR and ISR gene sets (139 genes) across all clusters showed no significant changes that associate with UPR or ISR activity (Fig. 4d,e and Extended Data Fig. 4b). ATF4 has also been shown to increase transcription of the redox regulator NRF2 through GSH and regulate amino acid import into cells by increasing cystine uptake through the glutamate–cystine antiporter xCT[38]. We did a gene-set score analysis for the amino acid-transport gene set (192 genes), oxidative stress-related peroxidase activity gene set

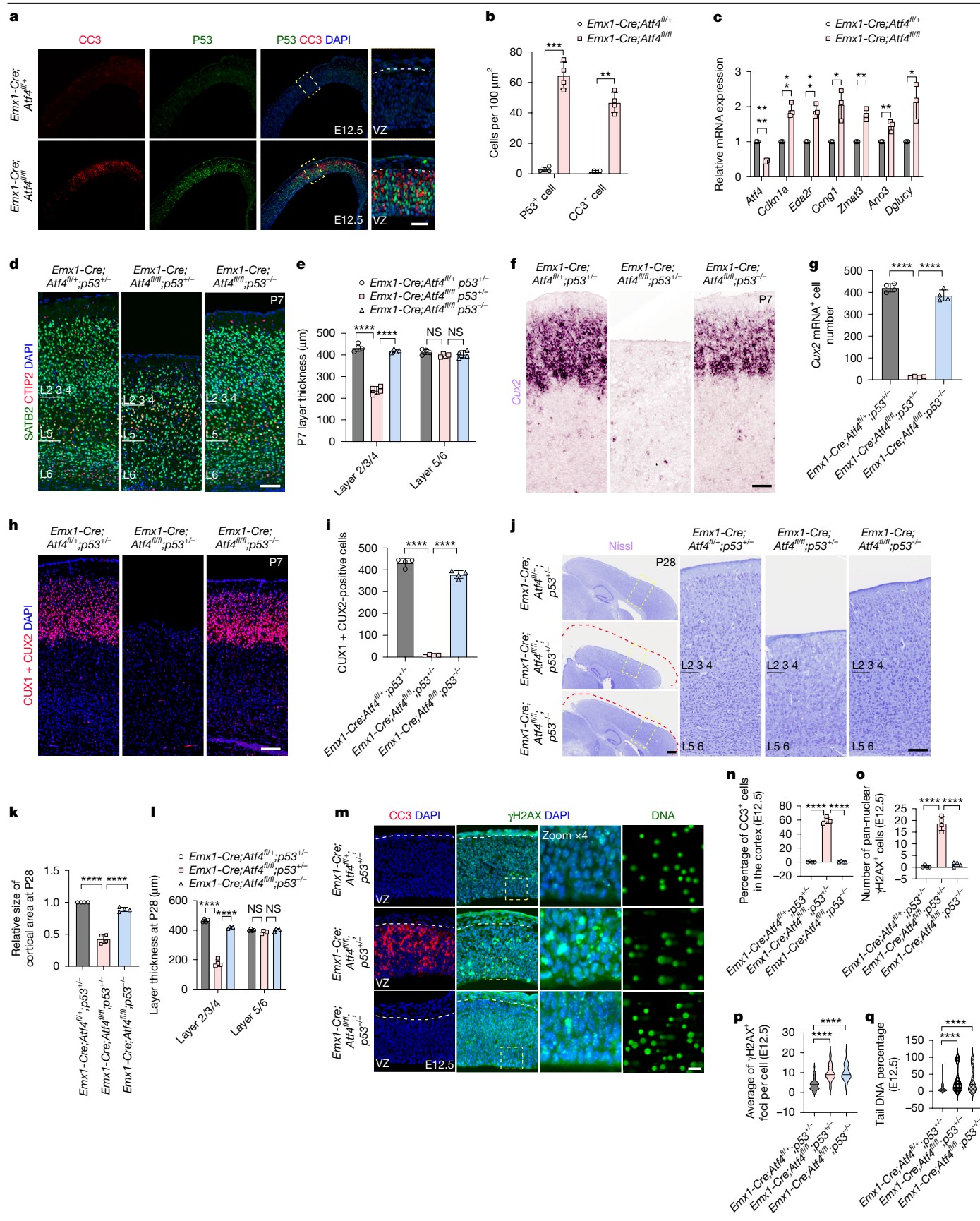

**Fig. 3** | See next page for caption.

**Fig. 3 | Selective CUX2⁺ neuron loss in the absence of ATF4 is p53-dependent after DNA damage. a,b,** Immunostaining for CC3 and p53 in E12.5 *Emx1-Cre; Atf4^fl/fl* and control cortices (**a**) and quantification of positive cells (**b**) (*n* = 4 mice per genotype). Scale bar, 25 μm. **c,** Quantitative PCR with reverse transcription (RT–qPCR) analysis of p53 target genes in E12.5 *Emx1-Cre;Atf4^fl/fl* and control cortices (*n* = 3 independent experiments). **d,e,** SATB2, CTIP2 and DAPI staining in P7 cortices of the indicated genotypes (**d**) and layer thickness quantification (**e**) (*n* = 4 mice per genotype). Scale bar, 100 μm. **f,g,** *Cux2* mRNA ISH in P7 cortices of the indicated genotypes (**f**) and quantification of *Cux2*⁺ neurons within a 500 × 500-μm² cortical region (**g**) (*n* = 4 mice per genotype). Scale bar, 100 μm. **h,i,** CUX1/CUX2 immunostaining in P7 cortices of the indicated genotypes (**h**) and quantification of CUX1 and CUX2 double-positive neurons within a 500 × 500-μm² cortical region (**i**) (*n* = 4 mice per genotype). Scale bar, 100 μm. **j–l,** Nissl staining in P28 cortices (**j**) with quantification of relative cortical area (**k**) and layer thickness (**l**) of the indicated genotypes (*n* = 4 mice per genotype). Scale bar, 100 μm. **m–q,** Immunostaining (CC3 and γH2AX) and NSC comet assay in E12.5 cortices of the indicated genotypes (**m**). Quantifications include CC3⁺ percentage (**n**), pan-nuclear γH2AX⁺ cell number (**o**), γH2AX⁺ foci per cell (**p**) and comet tail DNA percentage (**q**) (*n* = 4 mice per genotype). Scale bar, 100 μm. All bar graphs represent mean ± s.d. (**b,c,e,g,i,k,l,n–q**). Statistical significance by two-sided unpaired *t*-test (**b,g,i,k,n–q**) or two-sided multiple *t*-tests (**c,e,l**). *P* = 0.0008 (**b**, P53), *P* = 0.0012 (**b**, CC3), *P* = 0.000003 (**c**, *Atf4*), *P* = 0.001378 (**c**, *Cdkn1a*), *P* = 0.001104 (**c**, *Eda2r*), *P* = 0.013362 (**c**, *Ccng1*), *P* = 0.001082 (**c**, *Zmat3*), *P* = 0.006917 (**c**, *Ano3*), *P* = 0.019874 (**c**, *Dglucy*), *P* = 0.000003 (**e**, layer 2/3/4, left), *P* = 0.000001 (**e**, layer 2/3/4, right), *P* = 0.050816 (**e**, layer 5/6, left), *P* = 0.539494 (**e**, layer 5/6, right), *P* < 0.0001 (**g**, left), *P* < 0.0001 (**g**, right), *P* < 0.0001 (**i**, left), *P* < 0.0001 (**i**, right), *P* < 0.0001 (**k**, left), *P* < 0.0001 (**k**, right), *P* < 0.000001 (**l**, layer 2/3/4, left), *P* < 0.000001 (**l**, layer 2/3/4, right), *P* = 0.039932 (**l**, layer 5/6, left), *P* = 0.054179 (**l**, layer 5/6, right), *P* < 0.0001 (**n**, left), *P* < 0.0001 (**n**, right), *P* < 0.0001 (**o**, left), *P* < 0.0001 (**o**, right), *P* < 0.0001 (**p**, left), *P* < 0.0001 (**p**, right), *P* < 0.0001 (**q**, left), *P* < 0.0001 (**q**, right)).

(64 genes) and *Nrf2* target genes (37 genes) in E11.5 single-cell sequencing data. However, no significant changes in the activity of these gene sets were observed (Fig. 4g–i). Furthermore, we assessed translation in NPs in the absence of *Atf4* at E11.5. Gene-set score analysis of the eukaryotic translation-factor gene set (69 genes, including initiation, elongation, termination and ribosome recycling factors) revealed no significant changes in activity (Extended Data Fig. 5a). To assess protein synthesis directly, we performed *O*-propargyl-puromycin (OPP) labelling on acutely isolated NP cells from E11.5 *Atf4^fl/fl* (control) and *Emx1-Cre;Atf4^fl/fl* embryos. Relative OPP intensity showed no difference between groups, indicating that global translation rates were unchanged despite the observed increase in DNA damage (Extended Data Fig. 5b–d). Instead, in the E11.5 *Emx1-Cre;Atf4^fl/fl* RG clusters, we noted significant differential expression in a large number of factors in the DDR (Fig. 4f). Of the 20 most significantly altered biological process gene ontology (BPGO) enrichment terms (10 activated and 10 suppressed) for differentially expressed genes (DEGs) in RG, 9 of the 20 BPGO terms were directly related to the DDR (Fig. 4j).

We performed a protein–protein interaction (PPI) network functional enrichment analysis on all upregulated DDR-related genes (Fig. 4k). We observed significant upregulation of key genes associated with DNA repair, and widespread upregulation of accessory genes involved in DDR, such as chromatin-remodelling factors, histone-modification factors, cell-cycle regulation factors and programmed cell-death factors. A much smaller number of factors were downregulated in this cluster after ATF4 loss in *Emx1-Cre;Atf4^fl/fl* mice, as shown in the volcano plot in Fig. 4l. Of these downregulated genes, 20% were also associated with the DDR (Fig. 4m). Together, our snRNA-seq data at E11.5 identified widespread in vivo changes in key components as well as in accessory factors required for the DDR, in cortical NPs lacking ATF4.

Similar to previous reports[39], we found that *Cacna1i* was significantly upregulated in our *Atf4* mutant mice, as a result of an abnormal *Atf4* exon 1 to *Cacna1i* exon 2 fusion transcript, producing an out-of-frame fusion without altering full-length *Cacna1i* mRNA. *Atf4* single-allele deletion showed a similar *Cacna1i* increase as the double-knockout, but increased DNA damage was seen only in double-knockouts (Extended Data Fig. 4c–f), indicating that this transcript is unlikely to have functional consequences. Moreover, no differences were found in *Setx* expression or R-loop foci[34] between *Emx1-Cre;Atf4^fl/fl* and control mice (Extended Data Fig. 4g–i).

### ATF4 is a transcriptional regulator of DDR genes

Our companion paper[12] indicates that ATF4 acts not to prevent, but rather to repair, DNA damage, comprising a mechanism of neuronal resilience. We postulated that most of the DDR-related genes upregulated at E11.5 in cortical NPs lacking *Atf4* were a compensatory response by the cell to overwhelming DNA damage, but it ultimately fails owing to disrupted ATM phosphorylation at the initiation of double-strand DNA repair (Fig. 2t, u). The upregulated genes of the DDR in *Emx1-Cre; Atf4^fl/fl* mice were mostly accessory factors, and there are other arms of the DDR that may attempt to compensate for the loss of phospho-ATM to rescue the cell. For instance, the mechanisms that sense DNA damage may upregulate alternative arms, such as ataxia telangiectasia and RAD3-related (ATR) kinase, which is a master regulator of single-strand DNA break repair. This will be accompanied by significant upregulation of accessory factors in the DDR, but will ultimately fail to repair double-strand DNA breaks in the absence of phosphorylated ATM. A smaller number of factors were downregulated in *Emx1-Cre; Atf4^fl/fl* neuronal progenitors, and we proposed that ATF4 is a direct transcriptional regulator of these. We focused on four downregulated genes, *Cirbp*, *Uba52*, *Ebf1* and *Bcl6*, based on the presence of putative ATF4-binding sites in their upstream promoter regions. Cold inducible RNA binding protein (CIRBP) is a key regulator of double-strand DNA repair and modulates association of the crucial MRN (MRE11, RAD50 and NBS1) complex and ATM kinase with chromatin at the initiation of double-strand DNA repair[40]. The transcription factor EBF1 (early B cell factor 1) controls DNA damage repair in a dose-dependent fashion by directly regulating expression of its target gene, *Rad51* (ref. 41), and UBA52 is required for DNA repair and fine-tunes the spatiotemporal regulation of DNA-repair proteins at DNA damage sites through its interaction with RNF168 (ref. 42).

Chromatin immunoprecipitation with quantitative PCR (CHIP–qPCR) showed significant enrichment of ATF4 at promoter regions upstream of *Cirbp*, *Uba52* and *Ebf1* loci, but not *Bcl6*, perhaps because the predicted *Bcl6* ATF4 binding site is too far upstream (at around 5.5 kilobases) to be a functional site (Fig. 5a,b). We showed direct ATF4 transcriptional activation of the three candidate target genes using luciferase reporter assays (Fig. 5c). We also confirmed significant downregulation of mRNA and protein of these three genes (by qPCR and western blot) in NPs isolated from E11.5 *Emx1-Cre;Atf4^fl/fl* mice compared with controls (Fig. 5d,e).

### Targets of ATF4 are key components in DSB repair

To assess the function of the three direct ATF4 target genes in cortical NPs, we constructed lentiviral plasmids for *Atf4* overexpression, as well as for the overexpression or knockdown of its three target genes (*Cirbp*, *Uba52* and *Ebf1*) (Extended Data Fig. 6). Knockdown of *Cirbp*, *Uba52* or *Ebf1* in neural stem cells isolated from E11.5 cortex led to significant increases in cell death (Fig. 5f,g) and significant increases in DNA damage, assessed both by γH2AX foci staining (Fig. 5f,h) and comet assay (Fig. 5f,i).

In an alternative approach, we applied aphidicolin (APH)[43] to NPs isolated from the E11.5 cortex, resulting in DNA DSBs (γH2AX staining, comet assay) and cell death (CC3). Notably, individual overexpression

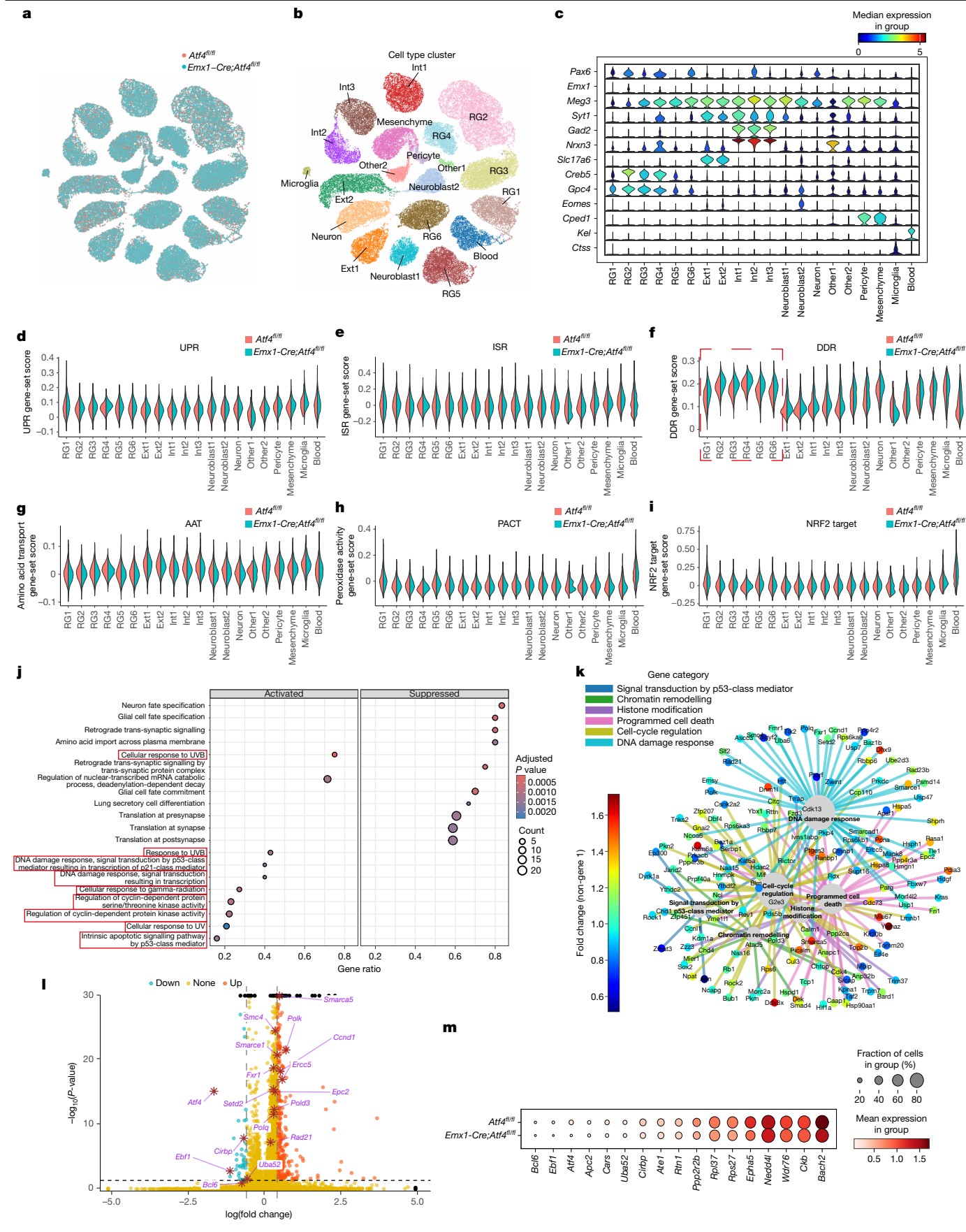

**Fig. 4** | See next page for caption.

**Fig. 4 | Loss of ATF4 in cortical NPs leads to global alterations in the DDR.**
**a**,**b**, Uniform manifold approximation and projection (UMAP) of integrated E11.5 snRNA-seq data from *Atf4^fl/fl^* and *Emx1-Cre;Atf4^fl/fl^* cortices, colour-coded by genotype (**a**) and cell type (**b**) (*n* = 3 biological repeats). **c**, Violin plots of normalized expression for selected cell-type marker genes. **d**–**i**, Gene-set score analysis for unfolded protein response (UPR) (**d**), integrated stress response (ISR) (**e**), DDR (**f**) (red box highlights changes of DDR in the RG cell clusters), amino acid transport (AAT) (**g**), peroxidase activity (PACT) (**h**) and NRF2 targets (**i**) across clusters. **j**, GO biological process terms enriched in RG2 cells (activated and suppressed). Circle size indicates gene count; colour intensity indicates adjusted *P* value; one-tailed exact hypergeometric test by ClusterProfiler with default BH-FDR adjustment for *P* value (red boxes highlight

the DDR-related BPGOs) **k**, STRING functional protein association network of upregulated genes in the RG2 cluster. **l**, Volcano plot of DEGs in E11.5 snRNA-seq data. Differential expression between the two conditions was computed using the Wilcoxon rank-sum test implemented in Scanpy's rank_genes_groups function, with *P* values corrected for multiple testing using the Benjamini–Hochberg false discovery rate (FDR) procedure. **m**, Dot-plot depicting downregulated DNA damage-related genes in the RG2 cluster from E11.5 snRNA-seq data. RG1, RG cell group 1; RG2, RG cell group 2; RG3, RG cell group 3; RG4, RG cell group 4; RG5, RG cell group 5; RG6, RG cell group 6; Ext1, excitatory neuron group 1; Ext2, excitatory neuron group 2; Int1, inhibitory neuron group 1; Int2, inhibitory neuron group 2; Int3, inhibitory neuron group 3.

of *Atf4* or its target genes (*Cirbp*, *Uba52* and *Ebf1*) partly mitigated the detrimental effects caused by APH (Fig. 5j–m). We also combined *Atf4* overexpression with the knockdown of each of these three target genes under APH treatment. The protective effect of *Atf4* under APH-induced stress was significantly diminished when its target genes (*Cirbp*, *Uba52* and *Ebf1*) were knocked down (Fig. 5n–q).

Considering our earlier findings of disrupted ATM phosphorylation in RG in *Emx1-Cre;Atf4^fl/fl^* mice, we were particularly interested in the function of CIRBP as a transcriptional target of ATF4, because CIRBP has been shown to promote the association of phosphorylated ATM with DNA[40]. To validate the function of CIRBP in vivo, we used in utero electroporation (IUE) of *Cirbp* shRNA (in a GFP expression plasmid) to knock down *Cirbp* in E14.5 wild-type brain cortex. At E15.5, we observed a very significant reduction in the total number of RG with phosphorylated ATM(Ser1981) (compared with IUE with a control empty vector), and also the proportion of GFP and phospho-ATM(Ser1981) double-positive cells relative to the total phospho-ATM(Ser1981)⁺ population along the ventricular surface (Fig. 5r–t). We also performed *Cirbp* shRNA IUE at E14.5 and assessed the tissue at the later time of postnatal day 2 (P2), and found significant thinning of GFP⁺ L2/3 thickness, as well as significant reductions in CUX1/CUX2⁺ and GFP⁺ cells (Fig. 5u–w). This demonstrates that *Cirbp* knockdown alone in RG reduces their ability to phosphorylate ATM, as well as causing subsequent thinning of L2/3. GFP⁺PAX6⁺ RG cells and GFP⁺TBR2⁺ intermediate progenitor cells were reduced with *Cirbp* knockdown, owing to increased cell death and DNA damage, rather than impaired neuronal proliferation (Extended Data Fig. 7). Together, these results show that *Cirbp* knockdown mimics the phenotype of *Atf4* loss in cortical neuronal progenitors, both in terms of disruption to phosphorylation of ATM in RG and also subsequent thinning of L2/3.

We also performed IUE of *Uba52* shRNA (in a GFP expression plasmid) to knock down *Uba52* in E13.5 wild-type brain cortex. At E16.5, we observed a significant reduction in the number of GFP⁺ and GFP⁺PAX6⁺ cells in the ventricular zone and subventricular zone region (Extended Data Fig. 8a–d), resulting from DNA damage-associated cell death (Extended Data Fig. 8e,f), rather than defects in NP proliferation or neuronal migration (Extended Data Fig. 8g–j). We also performed *Uba52* knockdown IUE in E13.5 p53-null mice, and found the GFP⁺ cell loss at E16.5 was rescued in the absence of p53 (Extended Data Fig. 8k,l). *Uba52* shRNA IUE at E14.5 and analysis at P2 showed a significant thinning of the GFP⁺ L2/3 and marked reductions in both CUX1/CUX2⁺ and GFP⁺ cells (Extended Data Fig. 8m–p). GO term analysis on our sequencing results (Fig. 4j and Extended Data Fig. 9a) had identified a small group of amino acid transporters, so we also performed IUE from E13.5 to E16.5 in wild-type cortex with shRNAs against *Slc1a3*, *Slc1a4*, *Slc7a1* and *Slc38a2* (Extended Data Fig. 9b) but observed no changes in GFP⁺ cell patterns, PAX6⁺GFP⁺ cell ratios or markers of cell death and DNA damage (Extended Data Fig. 9c–j). Together, these results demonstrate that ATF4 is a direct transcriptional regulator of genes that are critical for the DDR and implicate *Cirbp* as a key ATF4 target that is required for normal phosphorylation of ATM and the initiation of double-strand DNA repair.

## Discussion

The evolution of CUX2⁺ cortical neurons is of particular relevance to understanding advanced human cognitive abilities. Although cortical expansion is the main contributor to advanced human cognition, the supragranular cortex L2/3 and their CUX2⁺ pyramidal neurons show the greatest expansion during human brain evolution. Here we demonstrate that ATF4 is a crucial transcription factor in brain development by virtue of its highly specific role in repairing DNA damage and thereby rescuing cell death of neuronal progenitors for L2/3 excitatory neurons. We show that from their earliest origins, CUX2⁺ neuronal progenitors, destined to become L2/3 excitatory projection neurons of the adult cortex, are selectively vulnerable to *Atf4* loss of function. Remarkably, global loss of *Atf4* from cortical progenitors in *Emx1-Cre;Atf4^fl/fl^* mice leads to specific deletion of L2/3 CUX2⁺ excitatory neurons. We show at early stages that *Atf4* loss-of-function results in massive DNA damage and p53-dependent cell death. In a companion paper[12], we further show analogous vulnerability of CUX2⁺ neuron populations that also require ATF4 function in postnatal neuroinflammatory injury. Taken together, our findings show that *Cux2*-expressing neuronal progenitors of cortex require ATF4 for survival and to repair DNA damage during development and in disease.

### High burden of DNA damage in select neuronal progenitors
A likely common source of DNA damage in CUX2⁺ neuronal progenitors in development, as well as CUX2⁺ projection neurons in adult neuroinflammation, is ROS. ROS are by-products of aerobic metabolism and can act in redox biology as signalling molecules in the maintenance of physiological functions. In excess, however, they cause oxidative stress and damage to DNA. During neurogenesis, rapidly proliferating neuronal progenitors generate high levels of ROS as a by-product of mitochondrial respiration and activity[19,20], and the oxidative stress imposed by ROS is an important source of DNA damage. It is thought that cortical progenitors destined for L2/3, because they are the last to emerge from the ventricular zone, spend longer in the ventricular zone than progenitors destined for deeper cortical layers[9]. They are therefore probably exposed to ROS for a greater length of time in the ventricular zone, and to a greater cumulative oxidative stress and DNA damage load. We suggest that these neuronal progenitors recruit specific additional elements into their DDR, such as ATF4 and CUX2 (ref. 12), and that recruitment of this additional suite of DNA repair proteins was crucial to the expansion of L2/3 excitatory neurons. ROS also have a vital role in the pathophysiology of neurodegeneration in the adult, and glial and inflammatory cell sources contribute to excessive ROS in neuroinflammation[44]. It will be important to explore whether the vulnerability of CUX2⁺ lineage cells is contributed to by a heightened sensitivity to ROS and/or is due to a cell-intrinsic elevated production of ROS, owing to some property of their physiological function. It is also noteworthy that human CUX2⁺ pyramidal neurons differ in their gene expression, morphological complexity and electrophysiological activity from those of other mammalian species[45]. Their increased complexity may also be associated with inter-species differences in their generation of, or vulnerability to, ROS.

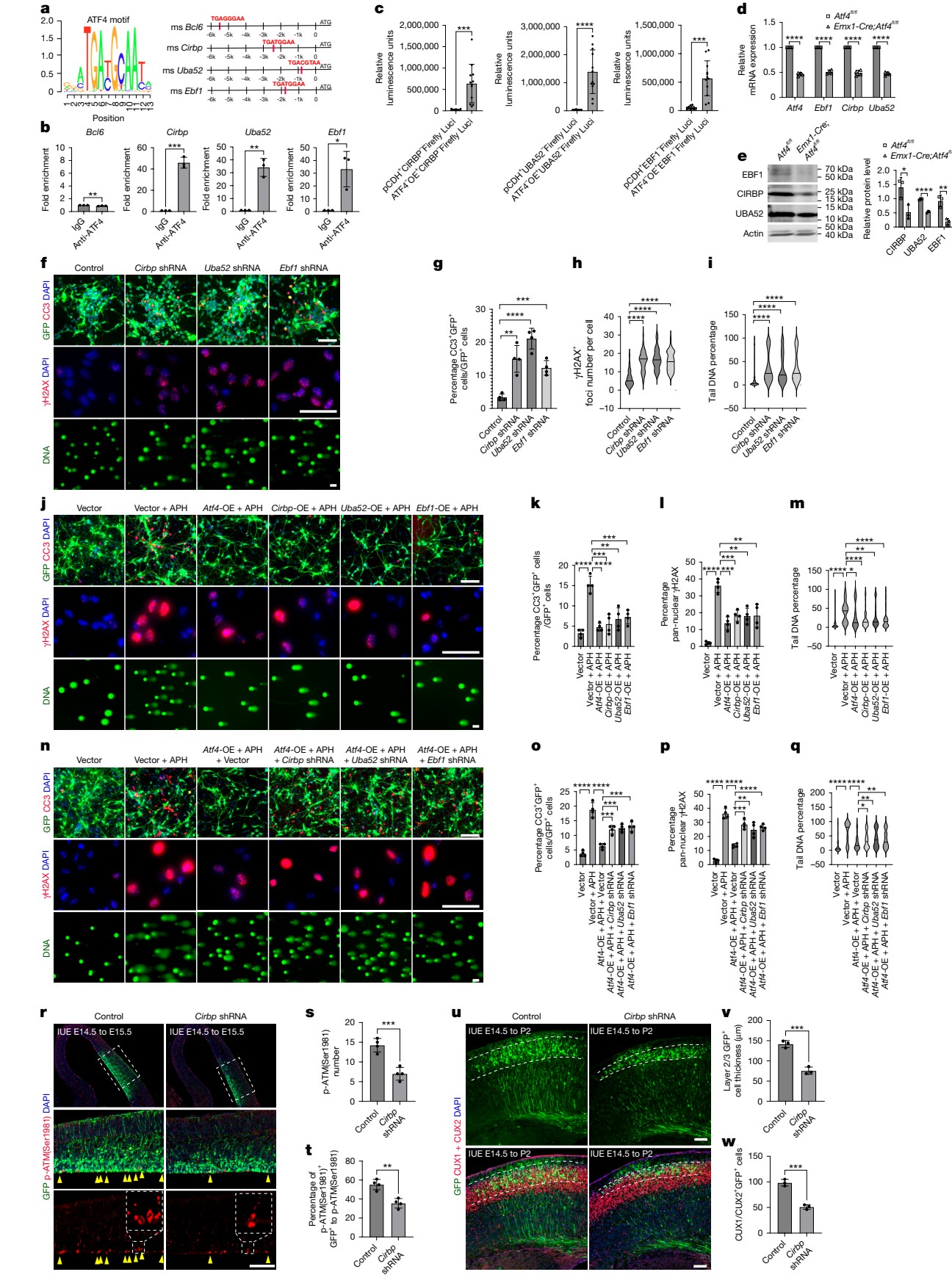

**Fig. 5** | See next page for caption.

**Fig. 5 | ATF4 is a crucial direct transcriptional regulator of the neuronal DDR. a**, ATF4 DNA-binding motifs[48] (from ISMARA, https://ismara.unibas.ch/supp/dataset3_v2/ismara_report/pages/ATF4.html) and analysis of promoter-binding sites in target genes (*Bcl6*, *Cirbp*, *Uba52* and *Ebf1*). ms, mouse. **b**, ChIP–qPCR showing ATF4 enrichment at target gene promoters (*n* = 3 independent experiments). **c**, Luciferase (Luci) reporter assays validating binding activity (*n* = 12 wells analysed across 3 independent experiments). **d**, RT–qPCR analysis of relative mRNA levels of *Atf4*, *Cirbp*, *Uba52* and *Ebf1* in the cortex of E11.5 *Emx1-Cre;Atf4*[fl/fl] cortex and controls (*n* = 4 independent experiments). **e**, Western blot analysis of CIRBP, UBA52 and EBF1 levels in E11.5 *Emx1-Cre;Atf4*[fl/fl] and control cortices (*n* = 3 independent experiments). **f–i**, Immunostaining for CC3/GFP and γH2AX, and comet assays in E11.5 NSCs infected with the indicated short hairpin RNA (shRNA) (**f**). Quantifications show percentage of CC3⁺GFP⁺ cells to total GFP⁺ cells (**g**) (*n* = 4 independent experiments), γH2AX⁺ foci per cell (**h**) (*n* = 50 cells from 4 independent experiments), and comet tail DNA percentage (**i**) (*n* ≥ 50 cells from 4 independent experiments). Scale bar, 50 μm. **j–m**, Immunostaining for CC3/GFP and γH2AX and comet assays in E11.5 NSCs overexpressing (OE) *Atf4*, *Cirbp*, *Uba52* or *Ebf1* following 500 nM APH treatment (**j**). Quantifications show percentage of CC3⁺GFP⁺ cells to total GFP⁺ cells (**k**) (*n* = 4 independent experiments), percentage of pan-nuclear γH2AX⁺ cells (**l**) (*n* = 4 independent experiments) and tail DNA percentage (**m**) (*n* = 52 cells from 4 independent experiments). Scale bar, 50 μm. **n–q**, Immunostaining for CC3/GFP and γH2AX, and comet assays in E11.5 NSCs under the indicated experimental conditions (**n**). Quantifications show the percentage of CC3⁺GFP⁺ cells to total GFP⁺ cells (**o**) (*n* = 4 independent experiments), the percentage of pan-nuclear γH2AX⁺ cells (**p**) (*n* = 4 independent experiments), and tail DNA percentage (**q**) (*n* ≥ 61 cells from 4 independent experiments). Scale bar, 50 μm. **r–t**, IUE of wild-type cortex from E14.5 to E15.5 with *Cirbp* knockdown and control; yellow arrowheads indicate GFP and p-ATM(Ser1981) double-positive cells (**r**). Quantifications show p-ATM⁺ cells in the VZ within a 500-μm bin (**s**) and percentage of p-ATM⁺GFP⁺ cells to total p-ATM⁺ cells (**t**) (*n* = 3 independent cortices). Scale bar, 100 μm. **u–w**, Co-staining of GFP and CUX1/CUX2 in P2 cortices following IUE at E14.5 with the indicated conditions (**u**). Quantifications show GFP⁺ cell layer thickness (**v**) (*n* = 3 independent cortices) and number of CUX1/CUX2⁺GFP⁺ cells within 200-μm bin (**w**) (*n* = 3 independent cortices). Scale bar, 100 μm. All data represent mean ± s.d. (**b–e,g–i,k–m,o–q,s,t,v,w**). Statistical significance by two-sided unpaired *t*-test (**b,c,g–i,k–m,o–q,s,t,v,w**) or two-sided multiple *t*-tests (**d,e**). *P* = 0.0061 (**b**, *Bcl6*), *P* = 0.0001 (**b**, Cirbp), *P* = 0.0012 (**b**, *UbA52*), *P* = 0.0158 (**b**, *Ebf1*), *P* = 0.0001 (**c**, left), *P* < 0.0001 (**c**, middle), *P* = 0.0001 (**c**, right), *P* < 0.000001 (**d**, *Atf4*), *P* < 0.000001 (**d**, *Ebf1*), *P* = 0.000001 (**d**, *Cirbp*), *P* < 0.00001 (**d**, *UbA52*), *P* = 0.024709 (**e**, *CIRBP*), *P* = 0.000078 (**e**, UBA52), *P* = 0.005096 (**e**, EBF1), *P* = 0.0014 (**g**, *Cirbp* shRNA), *P* < 0.0001 (**g**, *Uba52* shRNA), *P* = 0.0003 (**g**, *Ebf1* shRNA), *P* < 0.0001 (**h**, *Cirbp* shRNA), *P* < 0.0001 (**h**, *Uba52* shRNA), *P* < 0.0001 (**h**, *Ebf1* shRNA), *P* < 0.0001 (**i**, *Cirbp* shRNA), *P* < 0.0001 (**i**, *Uba52* shRNA), *P* < 0.0001 (**i**, *Ebf1* shRNA), *P* < 0.0001 (**k**, vector + APH), *P* < 0.0001 (**k**, *Atf4*-OE + APH), *P* = 0.0008 (**k**, *Cirbp*-OE + APH), *P* = 0.0026 (**k**, *UbA52*-OE + APH), *P* = 0.001 (**k**, *Ebf1*-OE + APH), *P* < 0.0001 (**l**, vector + APH), *P* = 0.0002 (**l**, *Atf4*-OE + APH), *P* = 0.0004 (**l**, *Cirbp*-OE + APH), *P* = 0.0011 (**l**, *UbA52*-OE + APH), *P* = 0.004 (**l**, *Ebf1*-OE + APH), *P* < 0.0001 (**m**, vector + APH), *P* = 0.0394 (**m**, *Atf4*-OE + APH), *P* < 0.0001 (**m**, *Cirbp*-OE + APH), *P* = 0.0037 (**m**, *UbA52*-OE + APH), *P* < 0.0001 (**m**, *Ebf1*-OE + APH), *P* < 0.0001 (**o**, vector + APH), *P* < 0.0001 (**o**, *Atf4*-OE + APH + vector), *P* = 0.0005 (**o**, *Atf4*-OE + APH + *Cirbp* shRNA), *P* = 0.0002 (**o**, *Atf4*-OE + APH + *Uba52* shRNA), *P* = 0.0004 (**o**, *Atf4*-OE + APH + *Ebf1* shRNA),), *P* < 0.0001 (**p**, vector + APH), *P* < 0.0001 (**p**, *Atf4*-OE + APH + vector), *P* = 0.0002 (**p**, *Atf4*-OE + APH + *Cirbp* shRNA), *P* = 0.0053 (**p**, *Atf4*-OE + APH + *UbA52* shRNA), *P* < 0.0001 (**p**, *Atf4*-OE + APH + *Ebf1* shRNA), *P* < 0.0001 (**q**, vector + APH), *P* < 0.0001 (**q**, *Atf4*-OE + APH + vector), *P* = 0.0227 (**q**, *Atf4*-OE + APH + *Cirbp* shRNA), *P* = 0.0019 (**q**, *Atf4*-OE + APH + *UbA52* shRNA), *P* = 0.0095 (**q**, *Atf4*-OE + APH + *Ebf1* shRNA), *P* = 0.0009 (**s**), *P* = 0.0018 (**t**), *P* = 0.0008 (**v**), *P* = 0.0009 (**w**).

## A role for ATF4 as a regulator of DNA damage repair

ATF4 is a well-established transcription factor that mediates the ISR and the UPR. We did not find alterations in classic components and targets of ISR or UPR, however, in CUX2⁺ neuronal progenitors lacking ATF4. Our data do not rule out effects on ISR and UPR in these cells, and indeed the UPR has been implicated in regulating cell-fate acquisition and generation of intermediate progenitors in developing cortex[19]. Although there has been little evidence that ATF4 directly regulates elements of the DDR[33], here we report a new function of ATF4 as a direct transcriptional regulator of the DDR in neuronal progenitors. We identify several ATF4 targets that are required for DNA repair, including UBA52, CIRBP and EBF1. It will be important to identify the full extent of ATF4 targets in the DDR and also the upstream signalling required for the activation of ATF4 in the context of DNA damage. ATF4 acts in vivo in heterodimers with other transcription factors and can have different effects in the same cell at the same time, depending on its choice of binding partner. It will be crucial to identify heterodimeric partners of ATF4 that act in the DDR, and how they may be regulated in different neuronal progenitor populations.

## CIRBP is required in neuronal progenitors for phosphorylation of ATM.

We demonstrate here a direct link between ATF4 transcriptional activity and DNA repair in neuronal progenitors of the cortex, through CIRBP. CIRBP was originally described as a DNA damage-induced transcript[46] and has been shown in cultured HeLa cells to promote phosphorylated ATM association with DNA[40]. CIRBP has also been implicated recently in the bowhead whale as being responsible for the exquisitely efficient DNA repair seen in this species, which allows for its long and cancer-free lifespan[47]. We show here that ATF4 transcriptionally activates CIRBP, which is subsequently required for the proper phosphorylation of ATM at Ser1981 in neuronal progenitors in vivo. Furthermore, knockdown of CIRBP alone in vivo led to significant cortical L2/3 thinning, highlighting the importance of this factor in corticogenesis. We find that CIRBP knockdown in vivo in developing cortex has the same phenotype as ATF4 loss in neuronal progenitors, both in terms of disrupted ATM phosphorylation in RG and subsequent thinning of L2/3. Considering that CIRBP is one of the most downregulated factors in human L2/3 cortex CUX2⁺ neurons in autism[18], it will be of great importance to fully understand its involvement in human neurological conditions.

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

# Methods

## Mouse experiments

All mice were handled in accordance with NIH guidelines and protocols approved by the UCSF Institutional Animal Care and Use Committee. Mice were housed under specific pathogen-free conditions in individually ventilated cages in a barrier facility on a 12 h:12 h light:dark cycle, with controlled temperature (20–26 °C) and humidity (30–70%). Housing density did not exceed five adult mice per cage; breeding cages (one male and up to two females) were maintained in a dedicated high-barrier area. Cages were changed weekly under laminar flow hoods, access was restricted with required PPE and colony health was monitored using sentinel mice. Both sexes were used, no sex-specific differences were observed and mice were randomly assigned to experimental groups.

C57BL/6 wild-type mice were obtained from the Jackson Laboratory (JAX:000664).

The *Emx1-Cre* line (*B6.129S2Emx1tm1(cre)Krj/J*, JAX:005628) has been previously described[49]. These mice were crossed with *Atf4* floxed mice to delete *Atf4* specifically in the early embryonic cortex. To assess phenotypes after blocking cell death, *Emx1-Cre* mice were also crossed with *Atf4* floxed and p53-null animals. *Emx1-Cre* mice were crossed with LSL-H2B-GFP mice for lineage tracing of EMX1+ cortical cells across developmental stages.

The *Atf4fl/fl* line (C57BL/6-Atf4tm1.1Cmad/J, JAX:033380) carries *loxP* sites flanking exons 2–3, which include the ATG start codon of the *Atf4* gene. These mice have been described previously[50] and were crossed with *Emx1-Cre* and/or p53-null mice to knock out the *Atf4* expression.

The p53-null (*p53−/−*) line (B6.129S2-Trp53tm1Tyj/J, JAX:002101) carries a neomycin cassette replacing exons 2–6 (including the start codon) of the *Trp53* gene. This line has been previously described[51]. These mice were crossed with *Emx1-Cre* and *Atf4* floxed mice to block p53-dependent cell death.

The LSL-H2B-GFP line (B6.Cg-Gt(ROSA)26Sortm8(CAG-HIST1H2BB/EGFP)Zjh/J JAX:036761) has a targeted mutation in the Gt(ROSA)26Sor locus with a *loxP*-flanked STOP cassette preventing transcription of a CAG promoter-driven enhanced green fluorescent protein (EGFP). EGFP expression occurs only after Cre-mediated recombination. This line has been previously described[52] and was crossed with *Emx1-Cre* mice for lineage-tracing experiments.

## Antibodies

For the immunostaining: GFP was detected with antibody GFP-1020 (Aves) at 1:1,000 dilution; mouse PAX6 was detected with antibody AB2237 (Millipore) at 1:500 dilution; mouse TBR2 was detected with antibody ab23345 (Abcam) at 1:500 dilution; mouse TBR1 was detected with antibody ab31940 (Abcam) at 1:1,000 dilution; mouse TUJ1 was detected with antibody T2200 (Sigma) at 1:1,000 dilution; mouse SATB2 was detected with antibody ab51502 (Abcam) at 1:500 dilution; mouse CTIP2 was detected with antibody ab18465 (Abcam) at 1:1,000 dilution; mouse CUX1 + CUX2 was detected with antibody ab309139 (Abcam) at 1:500 dilution; mouse calretinin was detected with antibody MAB1568 (Millipore) at 1:500 dilution; mouse parvalbumin was detected with antibody MAB1572 (Millipore) at 1:500 dilution; mouse calbindin was detected with antibody CB38a (Swant) at 1:500 dilution; human PAX6 was detected with antibody 901301 (Biolegend) at 1:500 dilution; mouse γH2A.X was detected with antibody ab2893 (abcam) at 1:500 dilution or 05-636 (Millipore) at 1:500 dilution; mouse CC3 was detected with antibody 9661 (Cell Signaling Technology) at 1:400 dilution; mouse Phospho-KAP-1 (Ser824) was detected with antibody A300-767A (Bethyl Laboratories) at 1:1,000 dilution; mouse p53 was detected with antibody 2524 (Cell Signaling Technology) at 1:500 dilution; mouse PCNA was detected with antibody 2586 (Cell Signaling Technology) at 1:500 dilution; mouse Ki67 was detected with antibody 550609 (BD Biosciences) at 1:500 dilution; mouse 53BP1 was detected with antibody NB100-304 (Novus Biologicals) at 1:500 dilution; mouse DNA-RNA Hybrid S9.6 was detected with antibody ENH001 (Kerafast) at 1:500 dilution; mouse p-ATM(Ser1981) was detected with antibody 05-740 (Millipore Sigma) at 1:500 dilution; mouse PHH3 was detected with Phospho-Histone H3 (Ser10) Antibody 9701 (Cell Signaling Technology at 1:500 dilution; mouse Nestin was detected with antibody MAB353 (Millipore Sigma) at 1:500 dilution; and mouse SOX2 was detected with antibody ab92494 (Abcam) at 1:500 dilution. All secondary antibodies for immunostaining were used at a dilution of 1:1,000. For immunoblotting: β-actin was detected with antibody 66009-1-Ig (Proteintech) at 1:1,000 dilution; EBF1 was detected with antibody AB10523 (Millipore) at 1:500 dilution; UBA52 was detected with antibody 18039-1-AP (Proteintech) at 1:500 dilution; CIRBP was detected with antibody 10209-2-AP (Proteintech) at 1:500 dilution; and ATF4 was detected with antibody 11815 (Cell Signaling Technology) at 1:500 dilution. All secondary antibodies used for immunoblotting were applied at a dilution of 1:20,000. For ChIP-qPCR, ATF4 antibody 11815 (Cell Signaling Technology) used at 1:50 dilution.

## In utero electroporation

Timed-pregnant C57BL/6 wild-type or other indicated genotype mice were anaesthetized and the uterine horns were exposed for the procedure. All plasmids used were Maxiprepped using the GeneJET Endo-Free Plasmid Maxiprep Kit (K0861) and then concentrated to 6 μg ul⁻¹ using NaCl and isopropanol. A total of 2 μl of *Cirbp* shRNAs or *Uba52* shRNAs or other indicated shRNA plasmid mixture (1:1 molar ratio, final concentration 3 mg ml⁻¹) mixed with pCAG-GFP at a 3:1 molar ratio and fast green (2 mg ml⁻¹) was microinjected into the fetal brain ventricles using a micropipette (made with Sutter P-30 Vertical Micropipette Puller). The plasmid mixture was then electroporated into the ventricular cells of the fetal brain using an electroporator (BTX ECM830). For each electroporation, five 40-V pulses of 50-ms duration were applied at 1-s intervals. After electroporation, the uterus was returned to the abdominal cavity and the abdominal wall and skin were sutured. EdU (50 mg per kg) was injected intraperitoneally before collecting the embryos, and pregnant mice were killed at different time points for phenotype analysis. Fetal brains were fixed in 4% paraformaldehyde (PFA) overnight, then dehydrated in 30% sucrose at 4 °C. Sub-regions of the cortex were identified based on cell density and visualized with DAPI nuclear staining. In all in IUE experiments, brains from at least three embryos were collected for analysis.

## RNA purification and RT–qPCR analysis

For RT–qPCR, the brain tissue or cell culture were homogenized and dissolved in 1 ml TRIZOL (Invitrogen) for 15 min on ice, then total RNA was extracted with Direct-zol RNA Miniprep kits, following the manufacturer's protocol. For reverse transcription, 2 μg RNA was used to synthesize the cDNA, cDNA was synthesized using the First-Strand cDNA Synthesis kit (APExBIO, K1072) and RT–qPCR was done using SYBR Green qPCR Master Mix (APExBIO, K1070) on a QuantStudio 5 system according to the manufacturer's instructions. The expression of specific mRNAs was quantified using the primer sequences listed in Supplementary Table 8. Relative mRNA expression levels were calculated using the comparative cycling threshold method (ΔΔCT), with β-actin serving as the normalization control.

## Cell culture

All cell cultures were maintained at 37 °C with 5% $CO_2$ in sterile, humidified incubators.

HEK293FT cells were cultured in DMEM medium supplemented with 10% FBS and penicillin–streptomycin.

NSCs were isolated from the embryonic cortex at stages E11.5, E12.5 and E13.5. Dissected embryonic cortices were collected in DMEM medium (Gibco) on ice and digested with 40 U ml⁻¹ papain (Worthington Biochemical) for 5 min at 37 °C in calcium- and magnesium-free EBSS (Gibco). The digested tissue was centrifuged at 500 r.p.m. for 30 s to remove papain, washed twice with DMEM and gently triturated in NSC

culture medium (50% DMEM/F12, 50% Neurobasal medium, 0.5% GlutaMAX, 1% non-essential amino acids, 10 ng ml⁻¹ bFGF, 10 ng ml⁻¹ EGF and 1% penicillin–streptomycin) to dissociate the cells. The cell suspension was then filtered through a 40-μm strainer to obtain single NSCs.

For the acute comet assay, NSCs collected at different time points were used directly, according to the manufacturer's instructions. For lentivirus infection experiments, NSCs were plated on plates pre-coated with poly-D-lysine (10 μg ml⁻¹) and laminin (10 μg ml⁻¹). After growing in proliferation medium for 12 h, cells were infected overnight with lentivirus and cultured for an additional 24 h before immunohistochemistry analysis. To induce DSBs, aphidicolin was added to the medium at a final concentration of 500 nM, and cells were cultured for 24 h before analysis.

## Plasmid construction, lentiviral production and transductions
The *Atf4*, *Cirbp*, *Uba52*, *Ebf1* full-length cDNA was subcloned into the pCDH-E2A-MCS-EGFP lentiviral overexpression vector with Takara PrimeSTAR Max DNA Polymerase, and all shRNAs targeting *Cirbp*, *Uba52*, *Ebf1* and four indicated amino acid transport genes were cloned to the pSicor knockdown lentiviral vector.

All batches of lentivirus were produced with target plasmid or control plasmid with psPAX2 (Addgene, 12260) and pMD2.G (Addgene, 12259) packaging plasmids. In brief, one million HEK293T cells were plated into six-well plates. After 12 h, two mixtures were made for the transfection. Mixture 1 contained 1.5 μg psPAX2, 1 μg pMD2.G and 2 μg target plasmid in 100 μl DMEM, and mixture 2 contained 13.5 μl PEI (1 mg ml⁻¹) in 100 μl DMEM. Mixture 2 was added to mixture 1 and gently mixed as the transfection mixture. The final mixture was incubated for 10 min before adding to 1 ml culture media in the six-well plate. Then, 6 h after the transfection, PEI-containing medium was replaced with 2 ml fresh medium. Two batches of viral media were collected 36 h and 72 h after the transfection. Then, the media were combined and centrifuged at 3,000*g* for 10 min to remove cell debris. For transduction of NSCs, the collected lentivirus was added to the NSC culture medium at a 1:1 ratio, along with polybrene (final concentration 800 ng ml⁻¹) to enhance infection efficiency. The medium was replaced 12 h after infection. To increase the knockdown efficiency, we mixed two lentiviruses at a 1:1 ratio for those target genes. The overexpression or knockdown efficiency for each virus (or virus mixture) was tested by western blot or RT–qPCR with infected NSCs. All shRNA oligonucleotide sequences are listed in Supplementary Table 8.

## Comet assay
After infection and treatments, cultured NSCs were trypsinized, suspended in prechilled PBS and diluted to a concentration of $1 \times 10^5$ cells per ml. For acutely isolated NSCs from cortex at different time points, cells were similarly diluted and directly subjected to the following steps. The cell suspension was mixed with an equal volume of 1% low-melting-point agarose, maintained at 37 °C, and immediately layered onto frosted glass slides (Fisher) precoated with 1% agarose. The cell–agarose mixture was gently compressed with a coverslip, then slides were placed flat on ice in the dark to allow the single-layer cell mixture to harden. Coverslips were removed after 10 min. Slides were kept in the dark on ice or at 4 °C for all subsequent steps.

Subsequent steps were performed following the Comet Assay Single Cell Gel Electrophoresis Assay manual (4250-050-K). In brief, slides were immersed in prechilled lysis buffer overnight at 4 °C, washed twice with prechilled distilled water (10 min each) and then placed in prechilled Alkaline Unwinding Solution for 30 min. Electrophoresis was done at 1 V cm⁻¹ in alkaline electrophoresis solution, adjusted according to the size of the electrophoresis chamber, for 30 min. After electrophoresis, the slides were neutralized in 0.4 M Tris-HCl (pH 7.0). Comets were stained with SYBR Gold (1:10,000 in PBS) for 10 min.

All experiments were done in triplicate, and a minimum of 50 comet-tail moments were imaged using a Zeiss apotome microscope and analysed using the ImageJ OpenComet plugin for quantification.

## Chromatin immunoprecipitation assay and qPCR
NSCs were acutely isolated from the cortex of E11.5 wild-type embryos. The isolated NSCs were fixed in 1% formaldehyde by adding 550 μl of 37% formaldehyde to 20 ml of growth medium and incubated at room temperature for 10 min with gentle swirling to ensure even mixing. Unreacted formaldehyde was quenched by adding glycine to a final concentration of 0.125 M and incubating at room temperature for 5 min.

Cells were collected by centrifugation at 1,500 r.p.m. for 10 min at 4 °C, followed by two washes with ice-cold PBS. The cell pellet was resuspended in ice-cold PBS containing 1× protease inhibitor cocktail, then centrifuged at 800*g* for 5 min at 4 °C to pellet the cells. Cells were lysed using cell lysis buffer followed by nuclear lysis buffer, as described in the EZ-Magna ChIP A/G Chromatin Immunoprecipitation Kit manual (17-10086). Chromatin was sheared into manageable sizes (200–1,000 base pairs) by sonication.

For downstream analysis, 5% of the lysate was set aside as input control, and the remaining 95% was used for immunoprecipitation following the kit's protocol. Immunoprecipitation was done using either IgG or anti-ATF4 antibodies. After elution of the protein–DNA complexes, reverse crosslinking was done to free the DNA, which was then purified using the columns provided in the kit.

The purified DNA was subjected to real-time qPCR analysis. Primers used for cloning and specific gene promoters are listed in Supplementary Table 8.

## Immunoblotting
Cell lysates were extracted using RIPA buffer supplemented with a protease inhibitor cocktail (CST) at 4 °C for 30 min, followed by sonication to ensure complete lysis. The supernatant was collected after centrifugation at 10,000 r.p.m. for 10 min at 4 °C. Proteins were separated on 10–15% SDS-PAGE gels and transferred onto 0.2 μm PVDF membranes. Blots were blocked with 5% BSA for 30 min at room temperature. Standard immunoblotting procedures were subsequently performed, followed by development using Li-COR IRDye secondary antibodies. Membranes were imaged using a Li-COR Odyssey CLx imaging system, and band intensities were quantified using ImageJ.

## Immunohistochemistry
Embryonic tissues were isolated and fixed overnight in 4% PFA. Postnatal tissues were collected after perfusion and fixed in 4% PFA overnight. After fixation, the tissues were washed with PBS and equilibrated in 30% sucrose (prepared with DEPC-PBS) at 4 °C. The samples were then cryopreserved in Tissue-Tek OCT Compound (Sakura Finetek) and sectioned at a thickness of 12 μm. For consistency in phenotypic analysis, we focused on primary somatosensory cortex areas across all postnatal stages shown in the figures, using rostral-to-caudal level-matched coronal brain sections for all data.

For immunostaining, tissue sections underwent antigen retrieval using pH 6.0 sodium citrate buffer, followed by washes in PBST (PBS containing 0.1% Triton X-100). Sections were then incubated in a blocking buffer (PBST with 10% normal goat serum (NGS); Millipore) for 1 h at room temperature. Next, sections were incubated overnight at 4 °C with primary antibodies diluted in PBST containing 1% NGS. After three washes, sections were incubated with secondary antibodies (Invitrogen) diluted in blocking buffer for 1 h at room temperature. Finally, sections were washed and mounted with Fluoromount-G.

For cultured cells, cells were washed with PBST, fixed with 4% PFA at room temperature for 15 min, and then washed with PBS. The same blocking and staining protocol used for tissue sections was applied to the cultured cells.

Images were acquired using a Zeiss apotome microscope, with identical parameters applied to all samples from the same experiment.

## Human brain tissue

Human specimens were collected from autopsy, with previous patient consent to institutional ethical regulations of the University of California San Francisco Committee on Human Research, as previously reported[53].

## EdU labelling and staining

EdU powder was dissolved in PBS (10 mg ml$^{-1}$) and incubated on a shaker at 37 °C until fully dissolved. Cumulative EdU labelling was done by administering intraperitoneal injections of 50 mg per kg EdU in sterile PBS to pregnant mice at the time points indicated in the figures. Tissues were collected at the corresponding time points, as shown in the figures. Before dissection, all embryos were transferred to ice-cold PBS to halt further EdU incorporation. Mouse brains were then dissected and fixed in ice-cold 4% PFA for 8–10 h. Cryostat sections were prepared and incorporated EdU was detected using the Click-iT EdU Alexa Fluor 594/647 imaging kit, following the manufacturer's protocol. Before imaging, the sections were incubated with primary antibodies overnight at 4 °C, followed by incubation with secondary antibodies at room temperature for 1 h. After three washes, the sections were stained with DAPI, mounted and prepared for imaging and analysis using the Zeiss Apotome and ZEN software.

## OPP labelling and staining

NSCs were isolated from E11.5 control and *Emx1-Cre;Atf4$^{fl/fl}$* cortices by 5 min of papain digestion at 37 °C, followed by gentle trituration and filtration through a cell strainer. Cells were immediately plated on glass coverslips pre-coated with poly-D-lysine (10 µg ml$^{-1}$) and laminin (10 µg ml$^{-1}$). After 3 h in proliferation medium (50% DMEM/F12, 50% Neurobasal medium, 0.5% GlutaMAX, 1% non-essential amino acids, 10 ng ml$^{-1}$ bFGF, 10 ng ml$^{-1}$ EGF and 1% penicillin–streptomycin) to allow attachment, OPP was added according to the Click-&-Go Plus 647 OPP kit instructions (CCT-1496) for 30 min (cells without OPP served as negative controls). Cells were then fixed with 4% PFA and subjected to immunostaining. The OPP intensity of each cell was measured using Zeiss Apotome and ZEN software.

## Isolation of single nuclei from embryonic cortices for snRNA-seq

Nuclei were isolated by pooling frozen brain tissue from two samples of the same genotype. All procedures were done on ice or at 4 °C in an RNase-free environment. In brief, frozen mouse brain cortex tissue was gently lysed in 3 ml of homogenization buffer (250 mM sucrose, 150 mM KCl, 30 mM MgCl$_2$, 60 mM Tris, 0.01% (v/v) Triton X-100, 0.001% (v/v) Digitonin, 0.01% (v/v) NP40, 1 mM DTT), supplemented with 0.2 U ml$^{-1}$ RNase inhibitor (NEB, M0314) and complete protease inhibitor cocktail (Roche, 11697498001). Lysis was done using a Wheaton Dounce Tissue Grinder (30 strokes with pestle B).

The lysate was filtered through a 40-µm cell strainer and centrifuged at 1,000*g* for 8 min to collect a nuclear pellet. To remove debris, the pellet was resuspended in 350 µl of homogenization buffer, mixed 1:1 with 50% iodixanol buffer (iodixanol 60% (v/v) in a buffer containing 250 mM sucrose, 150 mM KCl, 3 mM MgCl$_2$, 60 mM Tris) and layered over 600 µl of 29% iodixanol buffer (iodixanol 29% (v/v) in the same buffer). The sample was centrifuged at 13,500*g* for 20 min.

The supernatant was discarded and the nuclei were gently resuspended and washed in 1 ml of 1% BSA/PBS. Nuclear integrity and complete lysis were confirmed visually. Nuclei were counted manually and diluted to a concentration of 1,000 nuclei per µl in 1% BSA/PBS.

## Single-nucleus RNA-seq library preparation and sequencing

Single-nucleus RNA-seq libraries were prepared using Chromium Next GEM Single Cell 3′ Reagent Kit v.3.1 (10x Genomics), following the manufacturer's instructions. In brief, single-nucleus gel-bead-in-emulsions (GEMs) were generated in Chromium Controller with NextGEM Chip G (10x Genomics). The first-strand cDNA was synthesized on beads, followed by clean-up and amplification. The amplified cDNAs were examined in Bioanalyzer with a high-sensitivity DNA chip (Agilent). Then, 10 µl of cDNAs was forwarded to the library preparation. The dual-indexed libraries were pooled and sequenced on an Illumina NovaSeq 6000 (Illumina).

## Single-nucleus RNA-seq alignment and filtering

Demultiplexed FASTQ files were aligned to the *Mus musculus* reference genome (mm10-2020-A assembly) using the cellranger count pipeline provided by 10x Genomics (v.7.0.1) with default parameters, unless otherwise specified. Cell Bender (v.0.3.0) was used to eliminate technical artefacts, specifically ambient RNA contamination and barcode swapping, from the raw gene-expression counts.

## Cell-type annotation and clustering

Downstream analyses were done mainly using Scanpy (v.1.8.1) and DESC (v.2.1.1). Highly variable genes were identified using the highly_variable_genes function in Scanpy, and the top 2,048 genes were selected for subsequent analyses. These genes were used as input for DESC, a deep learning-based framework for dimensionality reduction, batch correction and unsupervised clustering. DESC learns a nonlinear low-dimensional representation of the data using a three-layer encoder architecture (1,024, 256 and 32 nodes). Cell clusters were annotated based on canonical marker gene expression. Differentially expressed genes for each cluster were identified using the rank_genes_groups function in Scanpy, and cell identities were assigned by comparison with established cell-type markers reported in the literature.

## Gene-set score analysis

Gene-set score analysis was done using the score_genes function in Scanpy (v.1.8.1). For each cell, a gene-set score was calculated based on an enhanced gene list curated from public databases and recent publications. Parameters were set to ctrl_size = 500 and n_bins = 25. The resulting scores reflected the relative expression levels of the corresponding gene sets.

The source of the genes for each gene set is listed here:

DNA damage repair gene set: a comprehensive list of 698 DNA damage-repair genes was assembled by merging genes annotated under GO:0006281 (DNA repair) and HALLMARK_DNA_REPAIR (M5898), and integrating this with the curated Human DNA Repair Genes list by R. Wood and M. Lowery[54].

UPR gene set: this set includes 116 genes associated with GO:0006986, sourced from the Mouse Genome Informatics (MGI) database.

ISR gene set: this comprises 23 genes, based on GO:0140467 from the MGI database.

Amino acid transport gene set: this set includes 193 genes annotated under GO:0006865, sourced from the MGI database.

Peroxidase Activity Gene Set: Consisting of 64 genes, this set is derived from GO:0004601 (peroxidase activity), also sourced from the MGI database.

NRF2 target gene set: this set includes 37 genes identified as NRF2 targets, based on a curated list from ref. 55.

Eukaryotic translation factors gene set: this set includes 69 genes including initiation, elongation, termination and ribosome recycling factors.

All genes for gene-set score analysis are listed in Supplementary Tables 1–7.

## DEG analysis, volcano plot and pathway analysis

DEGs were identified using the rank_genes_groups function in Scanpy with the Wilcoxon rank-sum test. Individual nuclei were treated as independent observations, and biological replicates (*n* = 3) were not

explicitly incorporated into the statistical model. Pathway and gene ontology enrichment analyses were done using the clusterProfiler R package (v.4.14.6), restricted to the biological process category and the top ten activated and suppressed pathways. Enrichment was assessed using the gseGO function with an adjusted *P* value < 0.05.

### PPI and gene ontology network analysis

PPI analysis of upregulated DEGs associated with the DDR was done using STRING (v.11.5). Analyses were conducted with the following settings: organism, *Mus musculus*; network edges, confidence-based; interaction sources, experiments and databases; minimum interaction score, high confidence; and no limit on maximum interactors. STRING networks were generated to represent both functional and physical protein associations, as specified in the corresponding figures.

Direct PPI enrichment *P* values were obtained from STRING. Gene ontology enrichment analyses in the PPI networks were prioritized, and FDR-corrected *P* values were reported using the whole genome as the background. Network visualization of the most significantly enriched GO terms and their associated genes was done using the GOenrich function in clusterProfiler (v.4.14.6).

### RNAscope

Mouse and human tissues were analysed using single-molecule fluorescence in situ hybridization (smFISH) with the RNAscope LS Multiplex kit, following the manufacturer's instructions. In brief, tissue sections were subjected to antigen retrieval by heating slides in buffer to unmask RNA targets, followed by protease treatment to enhance probe permeability. After washing, specific RNAscope probes were applied at the recommended concentration and hybridized at 40 °C. Signal-amplification reagents were then added sequentially, followed by fluorescent detection reagents (for example, tyramide-conjugated fluorophores). Slides were washed thoroughly between each step to remove unbound probes or reagents and counterstained with DAPI to visualize cell nuclei.

For combined RNAscope and immunostaining, standard immunohistochemistry procedures were followed. After completion of RNAscope, slides were protected from prolonged light exposure and incubated in blocking buffer (PBS containing 0.01% Triton X-100 and 10% normal goat serum; Millipore) for 1 h at room temperature. Sections were then incubated overnight at 4 °C with primary antibodies diluted in PBST containing 1% NGS. After three washes, sections were incubated with secondary antibodies (Invitrogen) diluted in blocking buffer for 1 h at room temperature. Finally, slides were mounted with ProLong Glass Antifade Mountant and imaged under a fluorescence microscope.

### In situ hybridization

Brain tissues were collected as previously described[53] and cryoprotected in 30% sucrose containing 0.1% diethyl pyrocarbonate (DEPC) to preserve mRNA integrity. Serial coronal or sagittal sections (13 μm thick) were prepared using a cryostat (Leica CM1950). For in situ hybridization, sections were incubated overnight at 65 °C with diluted denatured antisense probes. This was followed by three post-hybridization washes at 65 °C and two washes at room temperature. Sections were then incubated overnight at 4 °C with an anti-Digoxigenin-AP Fab fragments antibody (1:1,500, Sigma-Aldrich, 11093274910). DIG-labelled antisense CUX2 probes were used to identify CUX2-positive upper-layer neurons. The mRNA-expressing cells were visualized the next day as dark purple deposits using the NBT/BCIP–alkaline phosphatase substrate combination (Sigma-Aldrich, 11681451001). Images were captured using a Zeiss Axioscan microscope under bright-field illumination. To make the probe for the mouse *Cux2* mRNA, a partial mouse *Cux2* CDS sequence was cloned into the pGEM-T Easy Vector System (Promega) based on the sequence used in the Allen Brain Atlas ISH probe (https://mouse.brain-map.org/gene/show/12829). Images

were captured using a Zeiss Axioscan microscope under bright-field illumination.

### Nissl staining

We used 13-μm frozen brain sections, fixed in 4% PFA, for Nissl staining. The sections were washed in absolute alcohol for 5 min (twice), followed by 90% alcohol for 3 min and 70% alcohol for 3 min. Staining was done with 0.1% cresyl violet solution for 30 min at room temperature. After staining, the slides were rinsed in water to remove excess dye, followed by washes in graded ethanol (70%, 80%, 90% and 100%) to gradually dehydrate the tissue and enhance contrast. The slides were then immersed in xylene for 5 min to clear the tissue. A resin-based mounting medium was used to mount the slides. To improve imaging quality and help the automated imaging system to focus more accurately, coverslips were pressed with a heavy objective lens. Images were captured using a Zeiss Axioscan microscope under bright-field illumination.

### Quantification and statistical analysis

We followed standard practices for statistical analysis of biological data. Results are presented as the mean ± s.d. along with raw dot plots. Statistical significance between experimental and control groups was assessed using a two-tailed unpaired Student's *t*-test or two-tailed multiple *t*-test. Microsoft Excel and Graphpad Prism 10 were used to perform these statistical analyses unless specified otherwise. The threshold for statistical significance was set at 0.05, with significance levels denoted as follows: $P \geq 0.05$, not significant; *$P < 0.05$, **$P < 0.01$, ***$P < 0.001$, ****$P < 0.0001$. Details of the number of biological replicates (*n*) and their definitions and *P* values are given in the figure legends and source data. All data were analysed blindly, without consideration of genotype, to minimize bias.

### Reporting summary

Further information on research design is available in the Nature Portfolio Reporting Summary linked to this article.

## Data availability

The single-nucleus RNA sequencing data from E11.5 mouse brains generated in this study have been deposited in the Gene Expression Omnibus (GEO) under accession number GSE314470. All other data supporting the findings of this study are available from the corresponding authors upon reasonable request. Source data are provided with this paper.

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

**Acknowledgements** We thank members of the Fancy and Rowitch laboratories for comments and suggestions. This work was supported by funding from the European Research Council Advanced Grant (789054 to D.H.R.), the Wellcome Trust (to D.H.R.), the NIH (P01 NS083513 to D.H.R. and S.P.J.F., and R35 NS137478 to B.P.), the Dr. Miriam and Sheldon G. Adelson Medical Research Foundation (to D.H.R., D. H. G. and B. P.) and the NIHR Cambridge Biomedical Research Centre (NIHR203312). This work was supported by the NIH NINDS (R01 NS128021 and R21 NS133891 to S.P.J.F.), the US Department of Defence (MS230141 to S.P.J.F.), Alex's Lemonade Stand Foundation (to S.P.J.F.), the Race to Erase MS (to S.P.J.F.) and a gift from the Spangler Foundation (to S.P.J.F.).

**Author contributions** W.X., D.H.R. and S.P.J.F. conceived and designed all the experiments; W.X. did most of the experiments and analysed data; L.M., K.K.H., M.W., K.Z., X.-Y.T., G.J.,

J.G.-M., V.S.M. and S.M.P. assisted with experiments and sample preparation; Z.X. performed bioinformatic analysis on all single-cell sequencing data; I.-L.L. did staining on human developmental tissue; Q.W. and R.K. did single-cell sequencing and assisted with data analysis; L.M., Z.X., B.E., S.J.F., D.H.G. and B.P. contributed to discussions and design of the experiments. All authors provided input in the writing of the manuscript and W.X., D.H.R. and S.P.J.F wrote and edited the manuscript.

**Competing interests** The authors declare no competing interests.

**Additional information**

**Correspondence and requests for materials** should be addressed to David H. Rowitch or Stephen P. J. Fancy.

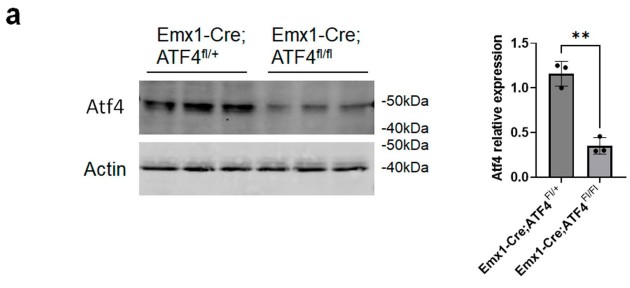

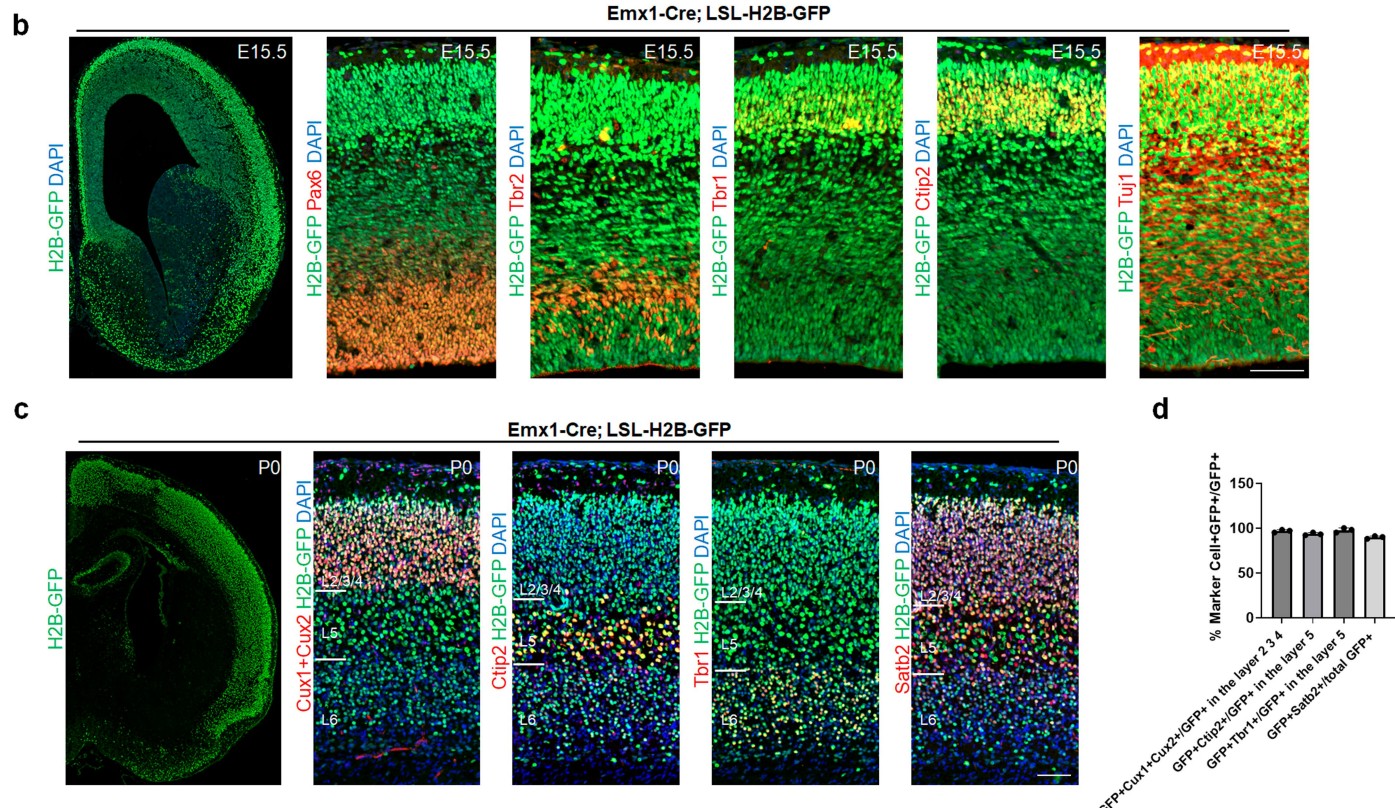

**Extended Data Fig. 1 | ATF4 gene knockout efficiency and Emx1-Cre expression pattern (related to Fig. 1). a**, Western blot analysis and quantification of Atf4 protein levels in E13.5 Emx1-Cre; Atf4$^{fl/fl}$ and control cortices (n = 3 mice per genotype). **b**, Immunostaining for GFP and Pax6, Tbr2, Tbr1, Ctip2, or Tuj1 in E15.5 Emx1-LSL-H2BGFP cortices. (n = 3 mice per genotype)

Scale bar, 100 µm. **c**,**d**, Representative images (**c**) and quantification (**d**) of the percentage of GFP+ neurons co-expressing Cux1/2 (layers 2/3/4), Ctip2 (layer 5), Tbr1 (layer 6), or Satb2 (all layers) in P0 Emx1-LSL-H2BGFP cortices (n = 3 mice per genotype). Scale bar, 100 µm. All bar graphs represent mean ± s.d. (**a**, **d**). Statistical significance by two-sided unpaired t-test (**a**). (P = 0.0011 (**a**)).

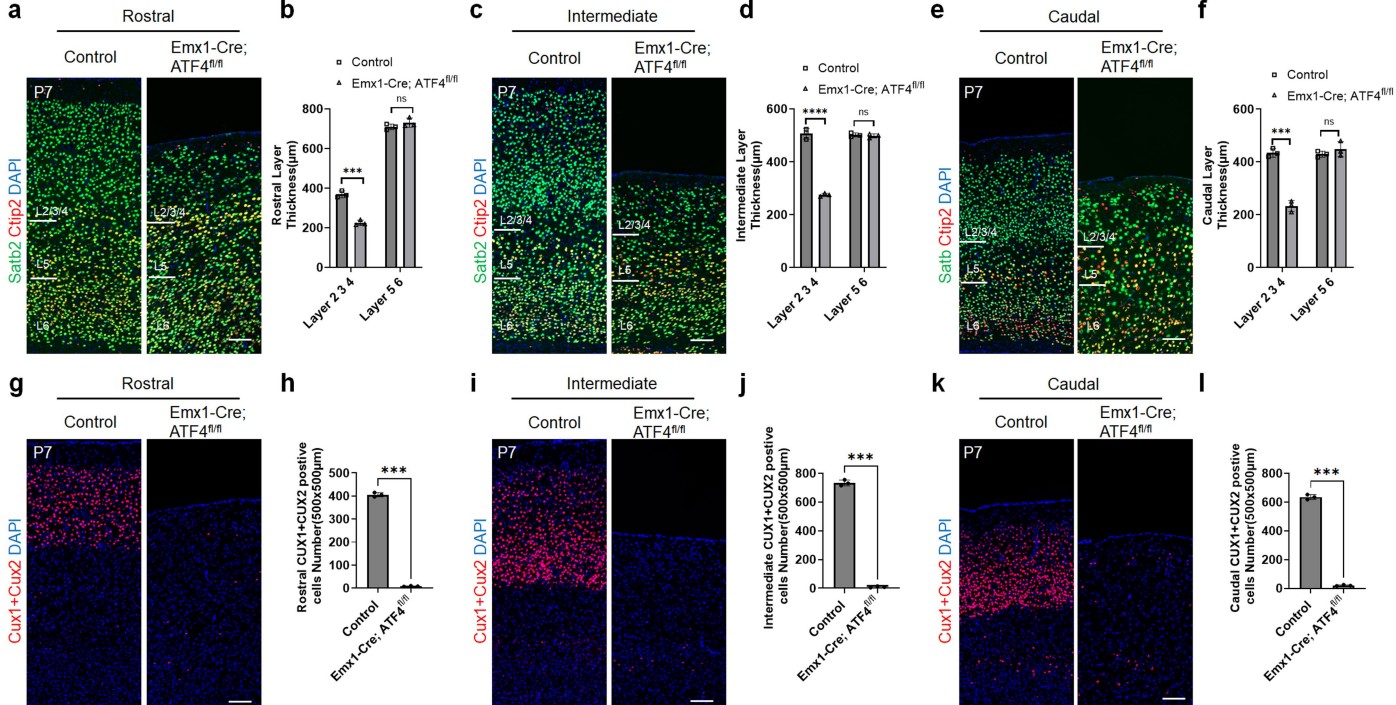

**Extended Data Fig. 2 | ATF4 knockout maintains phenotypic consistency across cortical levels (related to Fig. 1). a–f**, Representative images of Satb2, Ctip2, and DAPI staining in rostral (**a**), intermediate (**c**), and caudal (**e**) sections of P7 Emx1-Cre; Atf4$^{fl/fl}$ and control cortices, with corresponding quantification of layer 2/3/4 and layer 5/6 thickness (**b**, **d**, **f**) (n = 3 mice per genotype). Scale bars, 100 µm. **g–l**, Immunostaining of Cux1/2 in rostral (**g**), intermediate (**i**), and caudal (**k**) sections of P7 Emx1-Cre; Atf4$^{fl/fl}$ and control cortices, with corresponding quantification of Cux1 +/Cux2+ double-positive neurons (**h**, **j**, **l**) (n = 3 mice per genotype). Scale bars, 100 µm. All bar graphs represent mean ± s.d. (**b**, **d**, **f**, **h**, **j**, **l**). Statistical significance by two-sided unpaired t-test (**h**, **j**, **l**) or two-sided multiple t-tests (**b**, **d**, **f**). (P = 0.000335 (**b**, layer 2, 3, 4), P = 0.252074 (**b**, layer 5, 6), P = 0.000042 (**d**, layer 2, 3, 4), P = 0.621308 (**d**, layer 5, 6), P = 0.00019 (**f**, layer 2, 3, 4), P = 0.351271 (**f**, layer 5, 6), P = 0.0001 (**h**), P = 0.0001 (**j**), P = 0.0001 (**l**)).

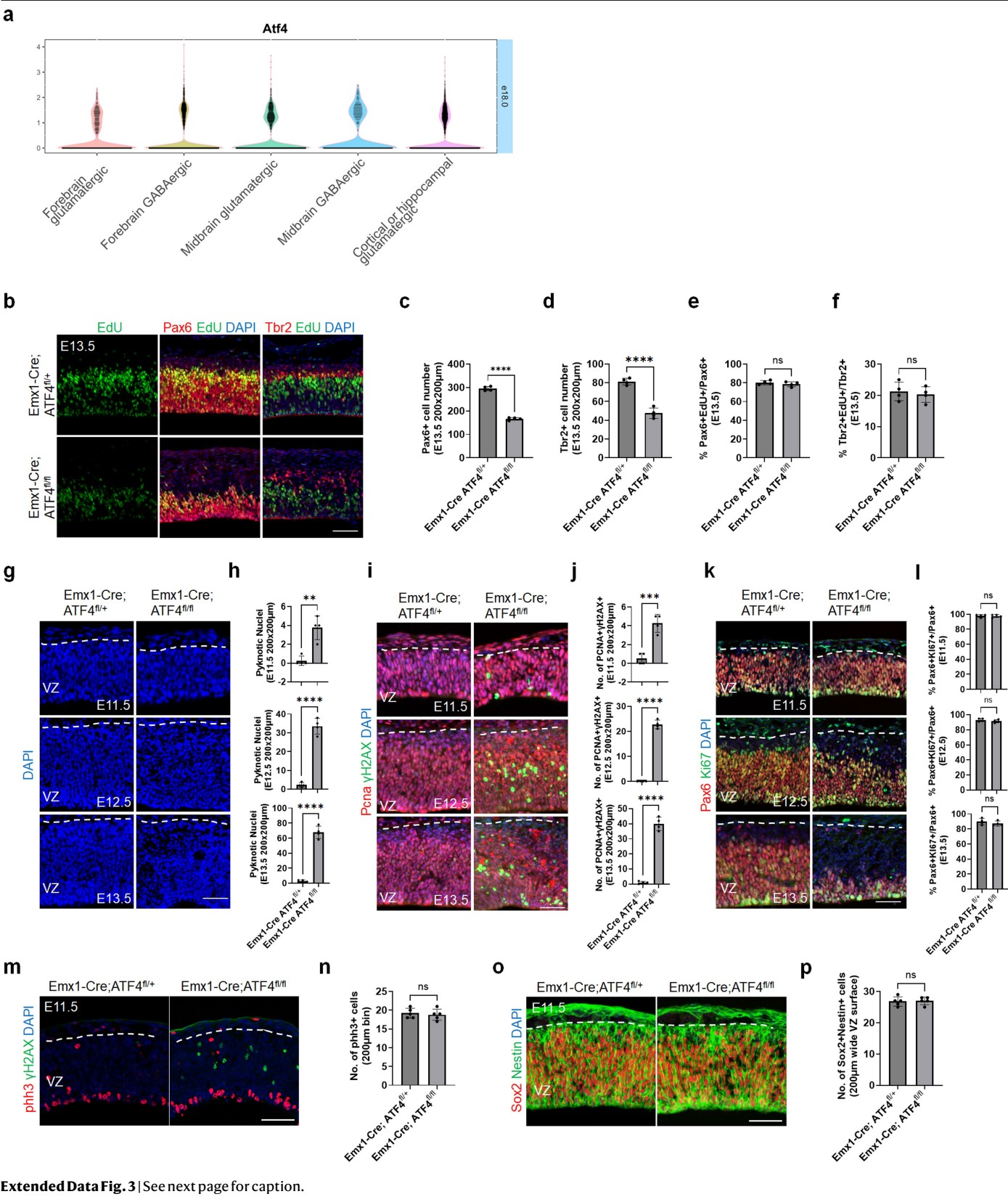

**Extended Data Fig. 3** | See next page for caption.

**Extended Data Fig. 3 | Replication-associated DNA damage accumulates in the ATF4-deficient cortex (related to Fig. 2). a**, Atf4 expression across neuronal subclasses. Gene expression visualization was performed in R (v4.3) using the plot1cell packages. The transcription factor Atf4 was used as the feature of interest. Embryonic cells from E18.0 were retained. Cells were further restricted to major glutamatergic and GABAergic subclasses across forebrain, midbrain, and cortex, as defined in the metadata, data coming from published paper[56]. **b**–**f**, Immunostaining for EdU(1 h labeling), Pax6, and Tbr2 in E13.5 Emx1-Cre; Atf4$^{fl/fl}$ and control cortices (**b**), with quantification of Pax6+ cells number (**c**), Tbr2+ cells number (**d**), percentage of Pax6+EdU+ cells (**e**), and percentage of Tbr2+EdU+ cells (**f**) (n = 4 mice per genotype). Scale bar, 50 μm. **g**,**h**, DAPI staining (**g**) and quantification of pyknotic nuclei (**h**) at E11.5, E12.5, and E13.5 (n = 4 mice per genotype) in the indicated genotypes. Scale bar, 50 μm. **i**,**j**, Immunostaining (**i**) and quantification (**j**) for γH2AX and PCNA at E11.5, E12.5, and E13.5 in the indicated genotypes (n = 4 mice per genotype). Scale bar, 50 μm. **k**,**l**, Immunostaining for Pax6 and Ki67 (**k**) and percentage of Pax6 + Ki67+ cells relative to total Pax6+ cells (**l**) in the indicated genotypes (n = 4 mice per genotype). Scale bar, 50 μm. **m**,**n**, Immunostaining for γH2AX and phh3 in E11.5 Emx1-Cre; Atf4$^{fl/fl}$ and control cortices (**m**) and quantification of phh3+ cells at E11.5 (**n**) (n = 5 mice per genotype). Scale bar, 50 μm. **o**,**p**, Immunostaining for Sox2 and Nestin in E11.5 Emx1-Cre; Atf4$^{fl/fl}$ and control cortices (**o**) and quantification of double-positive cells in the VZ at E11.5 (**p**) (n = 5 mice per genotype). Scale bar, 50 μm. All bar graphs represent mean ± s.d. (**c**, **d**, **e**, **f**, **h**, **j**, **l**, **n**, **p**). Statistical significance by two-sided unpaired t-test (**c**, **d**, **e**, **f**, **h**, **j**, **l**, **n**, **p**). (P < 0.0001 (**c**), P < 0.0001 (**d**), P = 0.3498 (**e**), P = 0.6259 (**f**), P = 0.0021 (**h**, E11.5), P < 0.0001 (**h**, E12.5), P < 0.0001 (**h**, E13.5), P = 0.0005 (**j**, E11.5), P < 0.0001 (**j**, E12.5), P < 0.0001 (**j**, E13.5), P = 0.8588 (**l**, E11.5), P = 0.2773 (**l**, E12.5), P = 0.3903 (**l**, E13.5), P = 0.6628 (**n**), P = 0.8222 (**p**)).

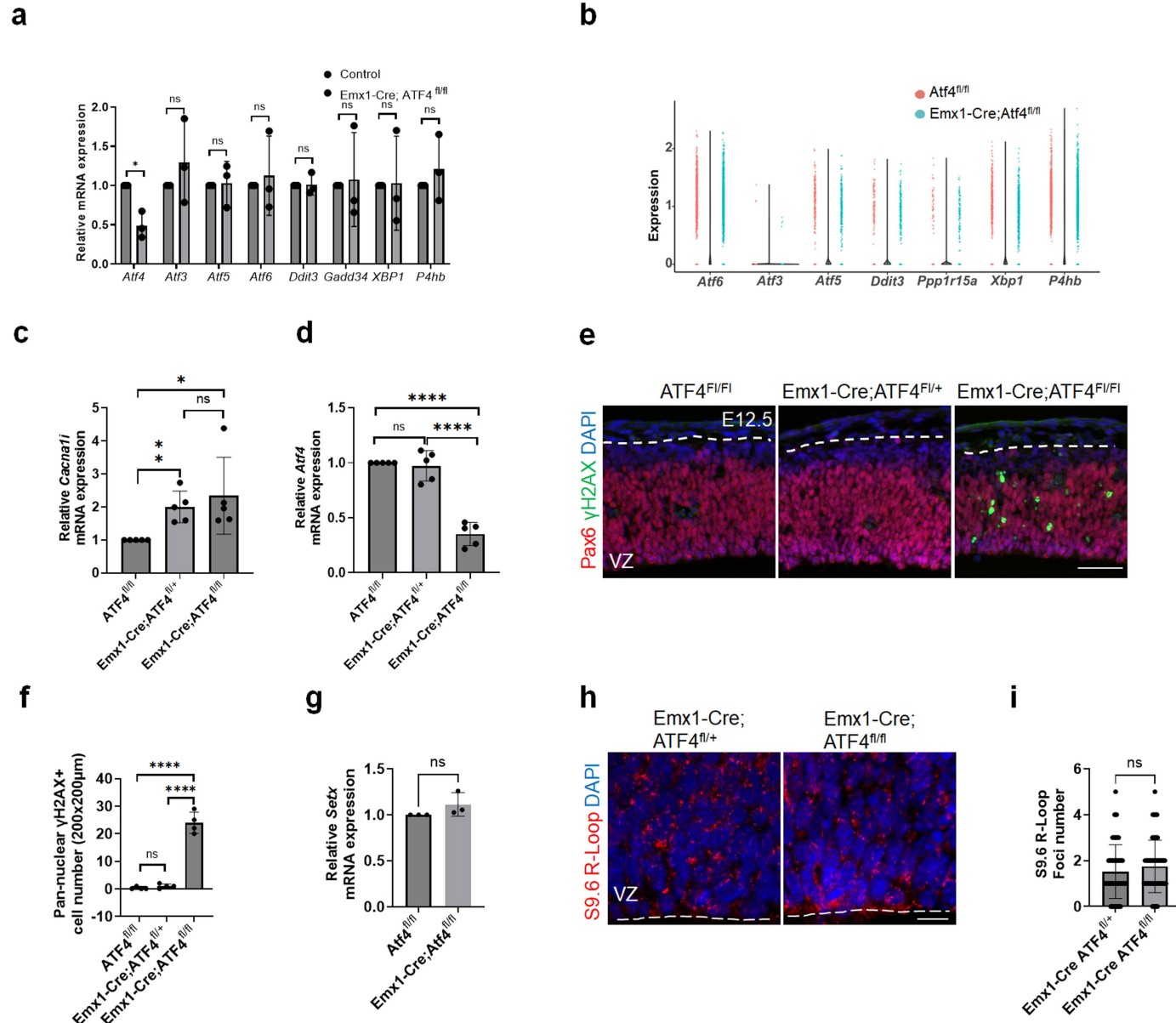

**Extended Data Fig. 4 | DNA damage was restricted to the cerebral cortex in Atf4 biallelic knockout mice (related to Fig. 4). a**, RT–qPCR analysis of ISR- and UPR-related genes in E13.5 Emx1-Cre; Atf4^fl/fl and control cortices (n = 3 independent experiments). **b**, Dot plot of ISR- and UPR-related gene expression in the RG2 (radial glial cell group 2) cluster from E11.5 snRNA-seq data. **c,d**, RT–qPCR analysis of *Cacna1i* (**c**) and *Atf4* (**d**) mRNA levels in E12.5 cortices of the indicated genotypes (n = 5 independent experiments). **e,f**, Immunostaining for Pax6 and γH2AX in E12.5 cortices of indicated genotypes (**e**) and quantification of pan-nuclear γH2AX+ cells (f) (n = 4 mice per genotype). Scale bar, 50 µm. **g**, RT–qPCR analysis of *Setx* mRNA levels in E12.5 Emx1-Cre; Atf4^fl/fl and control cortices (n = 3 independent experiments). **h,i**, S9.6 R-loop immunostaining in the VZ of indicated genotypes (**h**) and

quantification of S9.6 foci (**i**) (n = 3 mice per genotype, 60 cells). Scale bar, 10 µm. All bar graphs represent mean ± s.d. (**a, c, d, f, g, i**). Statistical significance by two-sided unpaired t-test (**a, c, d, f, g, i**). (P = 0.0346 (**a**, Atf4), P = 0.4479 (**a**, Atf3), P = 0.8695 (**a**, Atf5), P = 0.7094 (**a**, Atf6), P = 0.8775 (**a**, Ddit3), P = 0.8465 (**a**, Gadd34), P = 0.9402 (**a**, Xbp), P = 0.4816 (**a**, Ph4b), P = 0.0016 (**c**, Atf4 monoallelic knockout vs. ctrl), P = 0.0335 (**c**, Atf4 biallelic knockout vs. Ctrl), P = 0.5694 (**c**, Atf4 monoallelic knockout vs. biallelic knockout), P = 0.6641 (**d**, Atf4 monoallelic knockout vs. ctrl), P < 0.0001 (**d**, Atf4 biallelic knockout vs. Ctrl), P < 0.0001 (**d**, Atf4 monoallelic knockout vs. biallelic knockout), P = 0.1682 (**f**, Atf4 monoallelic knockout vs. ctrl), P < 0.0001 (**f**, Atf4 biallelic knockout vs. Ctrl), P < 0.0001 (**f**, Atf4 monoallelic knockout vs. biallelic knockout), P = 0.1975 (**g**), P = 0.272 (**i**)).

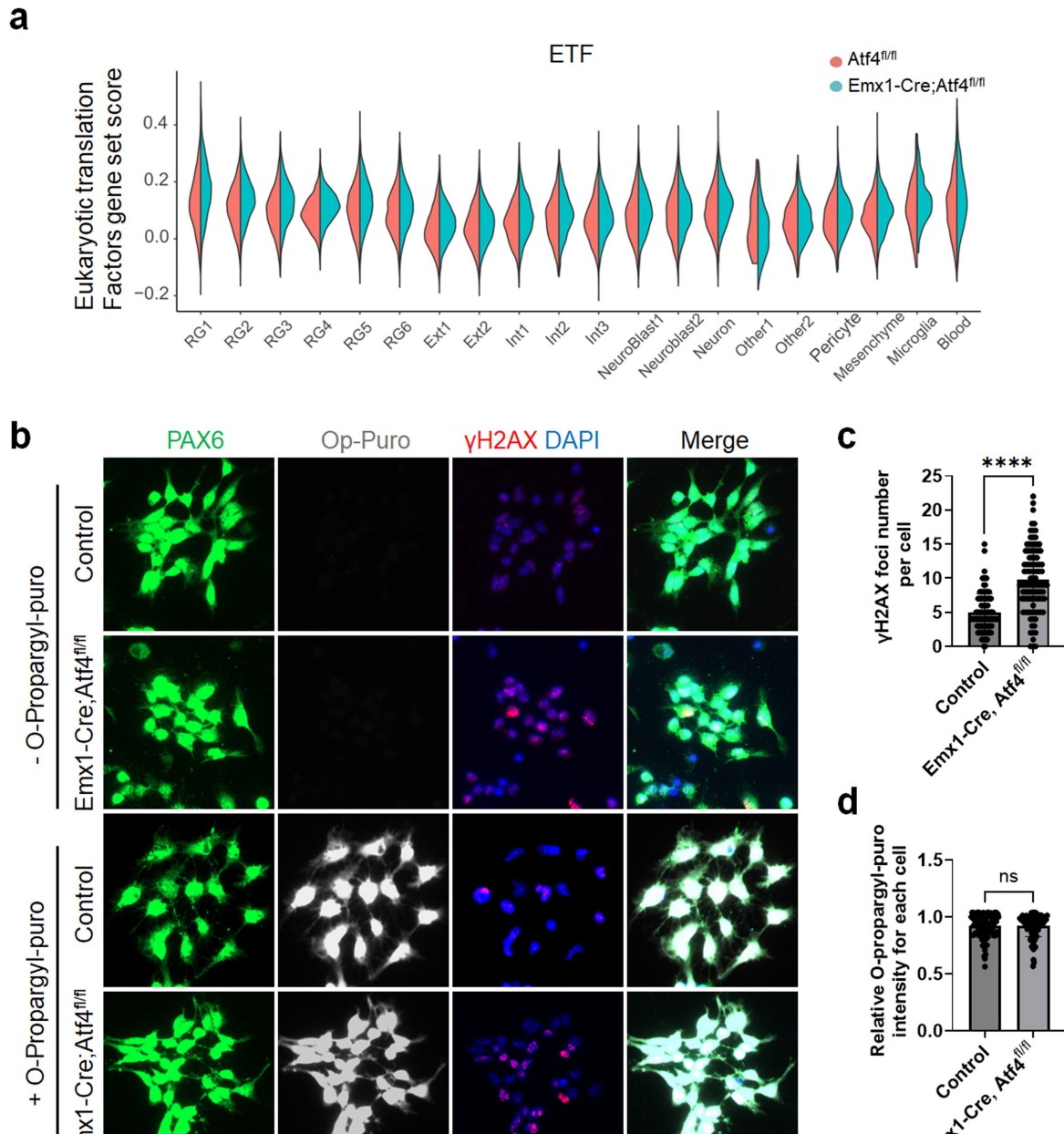

**Extended Data Fig. 5 | Protein translation was unchanged in the E11.5 ATF4-deficient cortex. (related to Fig. 4). a**, Gene set score analysis of DEGs with the Eukaryotic translation factors (EFT)-related gene set across different cell clusters. (RG1: radial glial cell group 1; RG2: radial glial cell group 2; RG3: radial glial cell group 3; RG4: radial glial cell group 4; RG5: radial glial cell group 5; RG6: radial glial cell group 6; Ext1: excitatory neuron group 1; Ext2: excitatory neuron group 2; Int1: inhibitory neuron group 1; Int2: inhibitory neuron group 2; Int3: inhibitory neuron group 3). **b**–**d**, Pax6, O-propargyl-puromycin (OPP), and γH2AX immunostaining of neural progenitor cells (NPCs) isolated from E11.5 cortices and treated ± OPP (**b**), with quantification of γH2AX+ foci (**c**) (n = 3 mice per genotype, 100 cells) and relative OPP intensity (**d**) (n = 3 mice per genotype, 100 cells). Scale bar, 50 μm. All bar graphs represent mean ± s.d. (**c, d**). Statistical significance by two-sided unpaired t-test (**c, d**). (P < 0.0001 (**c**), P = 0.8854 (**d**)).

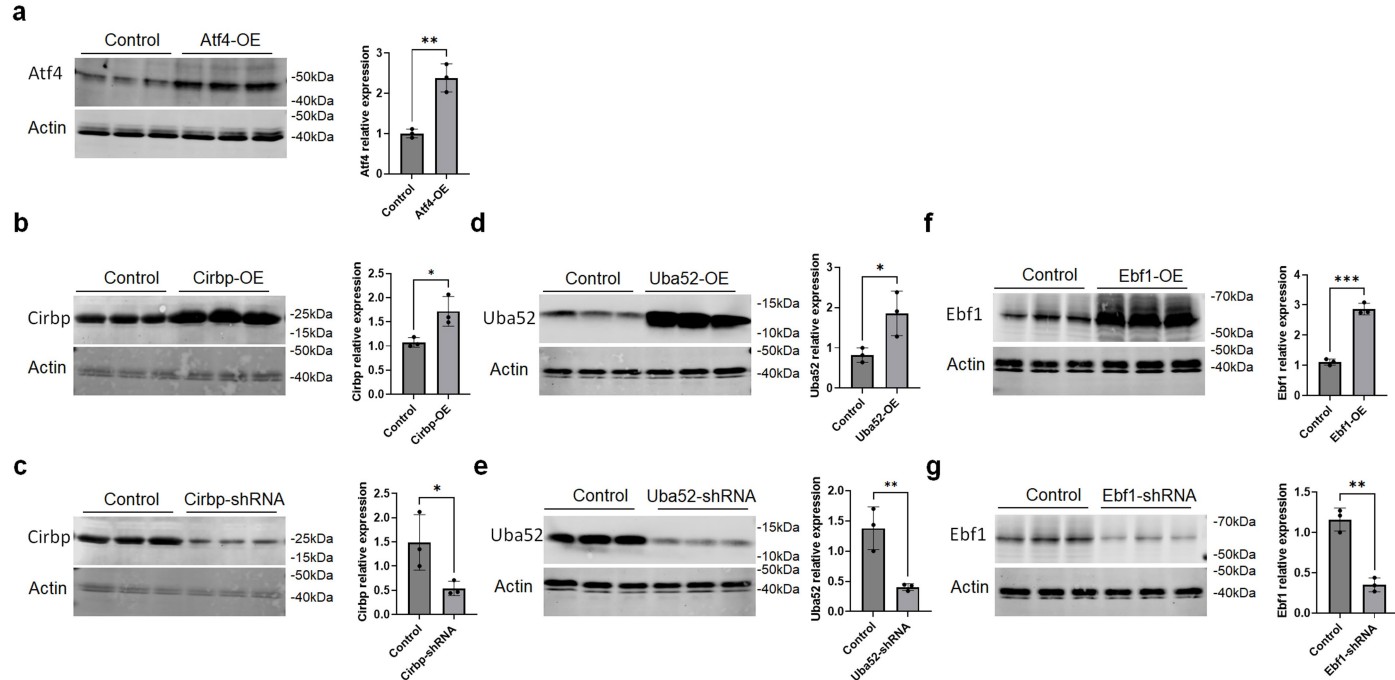

**Extended Data Fig. 6 | Western blot analysis of Atf4 overexpression and downstream genes overexpression and knockdown efficiency (related to Fig. 5). a–g**, Western blot analysis and quantification of Atf4 (**a**), Cirbp (**b,c**), Uba52 (**d,e**), and Ebf1 (**f,g**) protein levels in E11.5 NSCs following infection with the indicated overexpression or shRNA lentiviruses (n = 3 independent experiments). All bar graphs represent mean ± s.d. (**a, b, c, d, e, f, g**). Statistical significance by two-sided unpaired t-test (**a, b, c, d, e, f, g**). (P = 0.0028 (**a**), P = 0.0259 (**b**), P = 0.0498 (**c**), P = 0.0362 (**d**), P = 0.0092 (**e**), P = 0.0002 (**f**), P = 0.0011 (**g**).

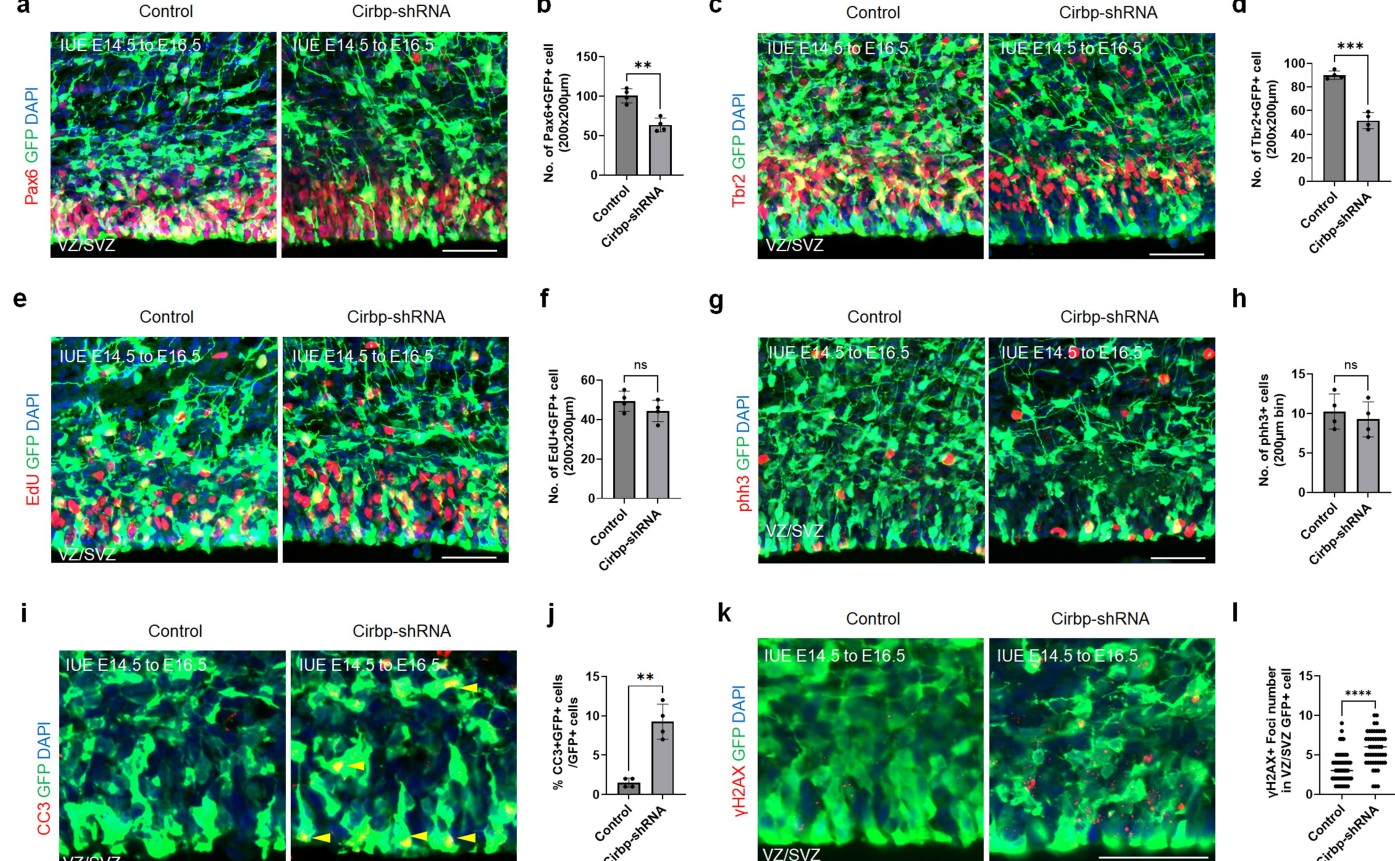

**Extended Data Fig. 7 | IUE Cirbp knockdown induces cell death and DNA damage. (related to Fig. 5). a–d**, Co-staining and quantification of GFP with Pax6 (**a,b**) or Tbr2 (**c,d**) in VZ/SVZ regions at E16.5 following IUE at E14.5 with Cirbp-shRNA or control (n = 4 independent cortices). Scale bars, 50 μm. **e,f**, EdU (1 h labeling) and GFP immunostaining (**e**) and quantification of double-positive cells (**f**) in the VZ/SVZ at E16.5 (n = 4 independent cortices). Scale bar, 50 μm. **g,h**, phh3 and GFP immunostaining (**g**) and quantification of phh3+ cells (**h**) (n = 4 mice per genotype). Scale bar, 50 μm. **i,j**, CC3 and GFP immunostaining (**i**) and percentage of CC3 + GFP+ cells (**j**) in the VZ/SVZ (n = 4 independent cortices). Scale bar, 50 μm. **k,l**, γH2AX and GFP immunostaining (k) and quantification of γH2AX+ foci (**l**) in VZ/SVZ GFP+ cells (n = 4 independent cortices, 51 cells). Scale bar, 50 μm. All bar graphs represent mean ± s.d. (**b, d, f, h, j, l**). Statistical significance by two-sided unpaired t-test (**b, d, f, h, j, l**). (P = 0.0011 (**b**), P = 0.0003 (**d**), P = 0.2271 (**f**), P = 0.5472 (**h**), P = 0.0044 (**j**), P < 0.0001 (**l**)).

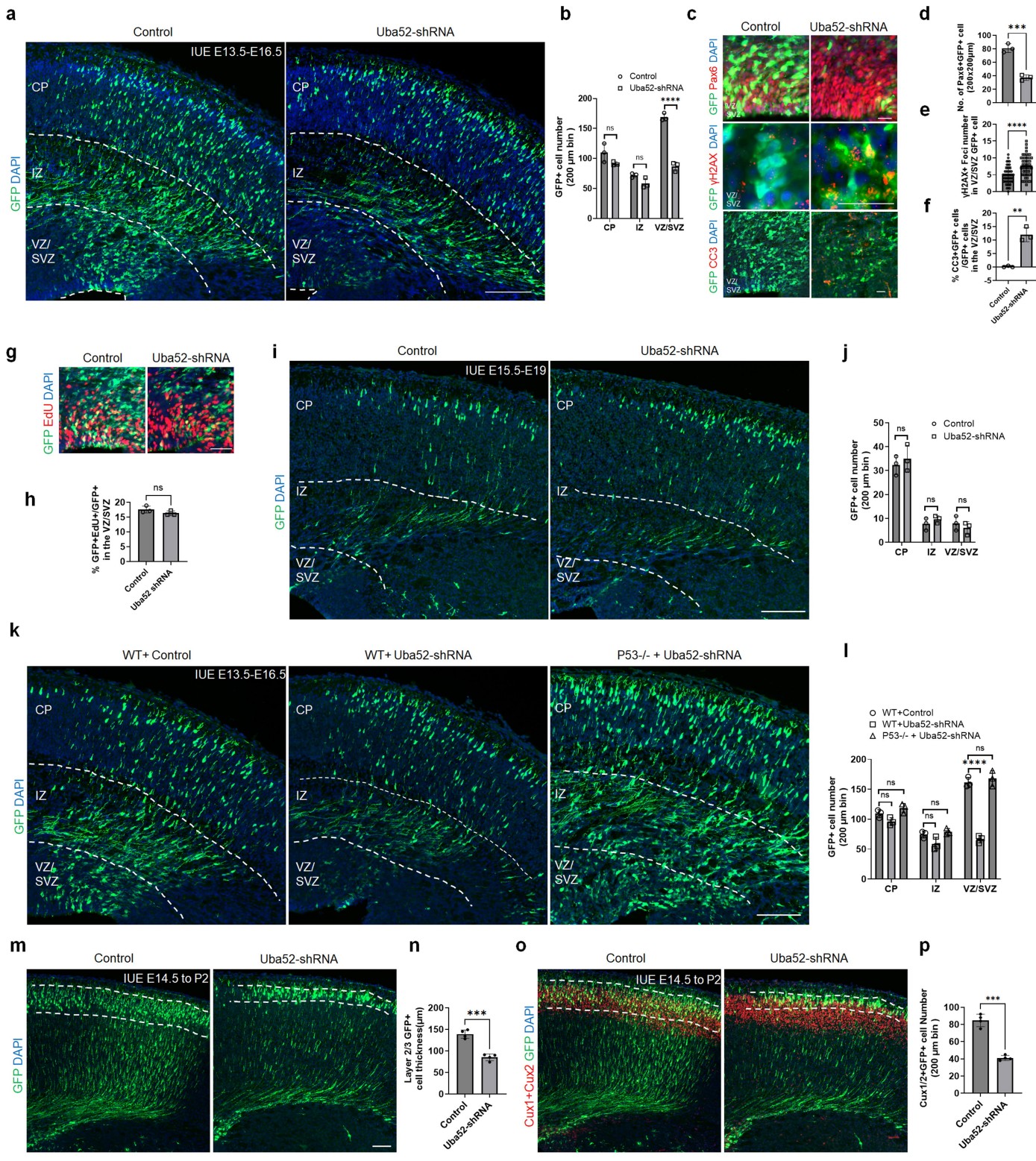

**Extended Data Fig. 8** | See next page for caption.

**Extended Data Fig. 8 | Cortical phenotypes after IUE-mediated Uba52 knockdown (related to Fig. 5). a,b,** IUE of E13.5 cortices with Uba52-shRNA or control, analyzed at E16.5 (**a**), and quantification of GFP+ cells across cortical zones (**b**) (n = 3 independent cortices). CP, cortical plate; IZ, intermediate zone; VZ/SVZ, ventricular–subventricular zone. Scale bar, 100 μm. **c–f,** Immunostaining of GFP with Pax6, γH2AX, or CC3 at E16.5 (**c**) and quantification of GFP+ Pax6+ cells (**d**), γH2AX+ foci (**e**), and percentage of CC3 + GFP+ cells (**f**) (n = 3 independent cortices). Scale bar, 25 μm. **g,h,** EdU (1 h labeling) and GFP immunostaining (**g**) and percentage of GFP+EdU+ cells (**h**) at E16.5 (n = 3 independent cortices). Scale bar, 50 μm. **i,j,** IUE of E15.5 cortices with control or Uba52-shRNA, analyzed at E19.5 (**i**), and quantification of GFP+ cells across zones (**j**) (n = 3 independent cortices). Scale bar, 100 μm. **k,l,** In utero electroporation (IUE) in wild-type and p53-/- embryonic cortices (**k**) and distribution of GFP+ cells across cortical layers (CP, IZ, VZ/SVZ) (**l**). Wild-type embryos were electroporated at E13.5 with control or Uba52 knockdown plasmids and analyzed at E16.5; p53-/- embryos received Uba52 knockdown from E13.5 to E16.5. (n = 3 independent cortices per condition). Scale bar, 100 μm. **m,n,** Immunostaining of GFP in wild-type cortical sections following in utero electroporation (IUE) at E14.5 with control or Uba52 knockdown constructs, analyzed at P2 (**m**), and quantification of GFP+ cell layer thickness (**n**) (n = 3 independent cortices). Scale bar, 100 μm. **o,p,** Immunostaining of GFP and Cux1/2 in wild-type cortical sections following IUE at E14.5 with control or Cirbp knockdown constructs, analyzed at P2 (**o**), and quantification of Cux1/2+GFP+ double-positive cells per 200-μm bin (**p**) (n = 3 independent cortices). Scale bar, 100 μm. All bar graphs represent mean ± s.d. (**b, d, e, f, h, j, l, n, p**). Statistical significance by two-sided unpaired t-test (**d**, **e**, **f**, **h, n, p**) or two-sided multiple t-tests (**b, j, l**). (P = 0.115371 (**b**, CP), P = 0.062473 (**b**, IZ), P = 0.000139 (**b**, VZ/SVZ), P = 0.0006 (**d**), P < 0.0001(**e**), P = 0.0015 (**f**), P = 0.1995 (**h**), P = 0.538778 (**j**, CP), P = 0.304559 (**j**, IZ), P = 0.435331 (**j**, VZ/SVZ), P = 0.068857 (**l**, CP Uba52-shRNA), P = 0.189228 (**l**, CP Uba52-shRNA+P53 null), P = 0.089776 (**l**, IZ Uba52-shRNA), P = 0.395612 (**l**, IZ Uba52-shRNA+P53 null), P = 0.000097 (**l**, VZ/SVZ Uba52-shRNA), P = 0.498589 (**l**, VZ/SVZ Uba52-shRNA+P53 null), P = 0.0002 (**n**), P = 0.0004 (**p**)).

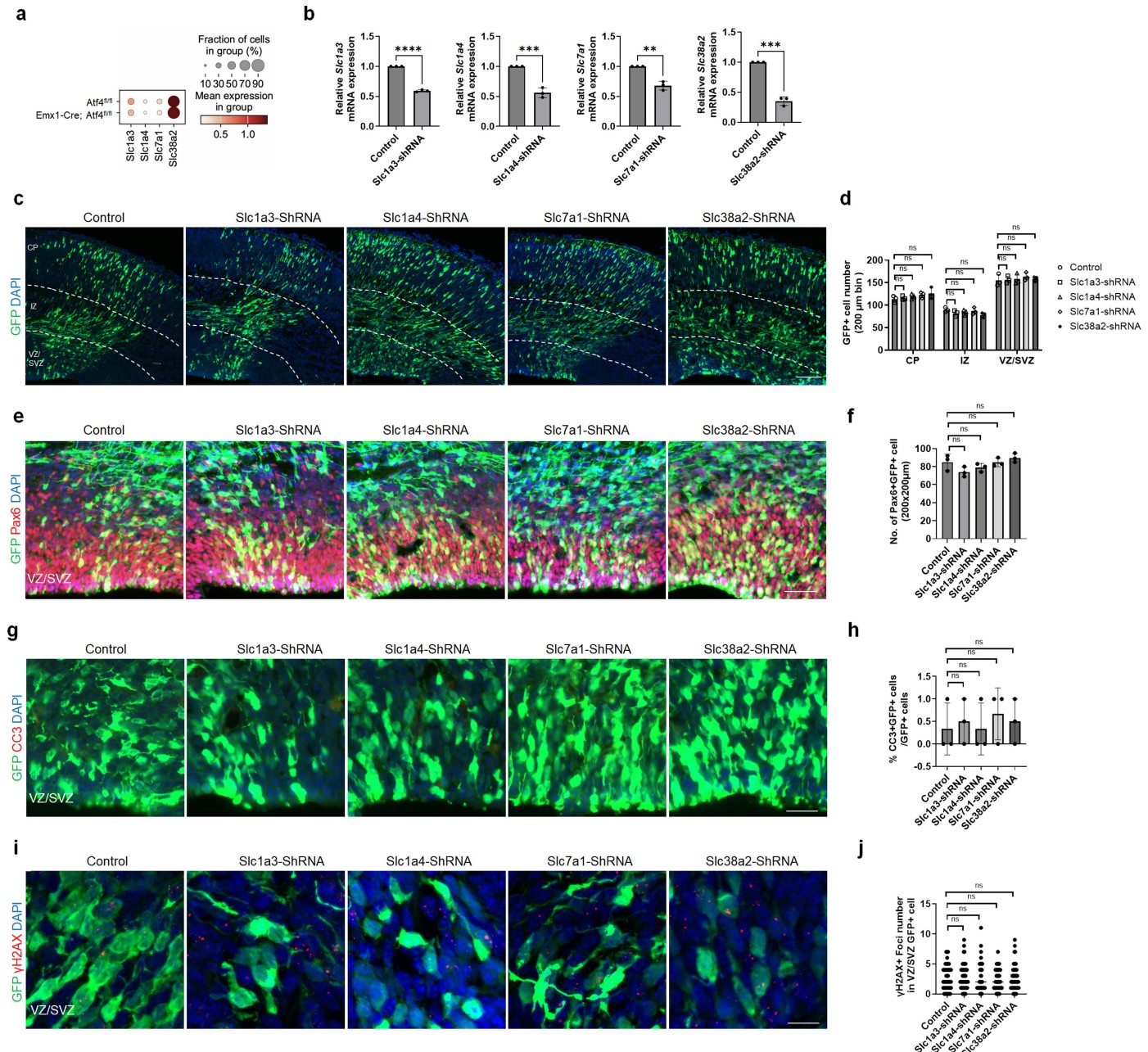

**Extended Data Fig. 9 | Atf4-regulated amino acid transporter knockdown does by IUE not impair cortical DNA integrity. (related to Figs. 4 and 5).**
**a**, Dot plot depicting downregulated amino acid transport genes (Slc1a3, Slc1a4, Slc7a1, and Slc38a2) in the RG2 cluster from E11.5 snRNA-seq data. **b**, RT–qPCR analysis of shRNA knockdown efficiency for Slc1a3, Slc1a4, Slc7a1, and Slc38a2 (n = 3 independent experiments). **c,d**, Distribution of GFP+ cells following IUE in E13.5–E16.5 cortices after knockdown of indicated genes compared to control (**c**), with quantification of GFP+ cells across cortical zones (**d**) (n = 3 independent cortices). Scale bar, 100 μm. **e,f**, Immunostaining for GFP and Pax6 (**e**) and quantification of GFP+Pax6+ cells in the VZ/SVZ (**f**) (n = 3 independent cortices). Scale bar, 50 μm. **g,h**, Immunostaining for GFP and CC3 (**g**) and percentage of CC3 + GFP+ cells (**h**) in the VZ/SVZ (n = 3 independent cortices). Scale bar, 25 μm. **i,j**, Immunostaining for GFP and γH2AX (**i**) and quantification of γH2AX+ foci in GFP+ cells (**j**) (n = 3 independent embryos, 50 cells). Scale bar, 20 μm. All

data represent mean ± s.d. (**b, d, f, h, j**). Statistical significance by two-sided unpaired t-test (**b, f, h, j**) or two-sided multiple t-tests (**d**). (P < 0.0001(**b**, Slc1a3-shRNA), P = 0.0007 (**b**, Slc1a4-shRNA), P = 0.0016 (**b**, Slc7a1-shRNA), P = 0.0001 (**b**, Slc38a2-shRNA), P = 0.676036 (**d**, CP- Slc1a3-shRNA), P = 0.369398 (**d**, CP-Slc1a4-shRNA), P = 0.189228 (**d**, CP-Slc7a1-shRNA), P = 0.214316 (**d**, CP-Slc38a2-shRNA), P = 0.201846 (**d**, IZ- Slc1a3-shRNA), P = 0.286066 (**d**, IZ-Slc1a4-shRNA), P = 0.642064 (**d**, IZ-Slc7a1-shRNA), P = 0.056209 (**d**, IZ-Slc38a2-shRNA), P = 0.86419 (**d**, VZ/SVZ- Slc1a3-shRNA), P = 0.786748 (**d**, VZ/SVZ-Slc1a4-shRNA), P = 0.428796 (**d**, VZ/SVZ-Slc7a1-shRNA), P = 0.687705 (**d**, VZ/SVZ-Slc38a2-shRNA), P = 0.1434 (**f**, Slc1a3-shRNA), P = 0.3574 (**f**, Slc1a4-shRNA), P = 0.957 (**f**, Slc7a1-shRNA), P = 0.473 (**f**, Slc38a2-shRNA), P = 0.7247 (**h**, Slc1a3-shRNA), P > 0.9999 (**h**, Slc1a4-shRNA), P = 0.5185 (**h**, Slc7a1-shRNA), P = 0.7247 (**h**, Slc38a2-shRNA), P = 0.6298 (**j**, Slc1a3-shRNA), P = 0.8911 (**j**, Slc1a4-shRNA), P = 0.4902 (**j**, Slc7a1-shRNA), P = 0.9228 (**j**, Slc38a2-shRNA)).

| | |
|---|---|

# Reporting Summary

## Statistics

For all statistical analyses, confirm that the following items are present in the figure legend, table legend, main text, or Methods section.

| n/a | Confirmed | |
|---|---|---|
| ☐ | ☒ | The exact sample size (*n*) for each experimental group/condition, given as a discrete number and unit of measurement |
| ☐ | ☒ | A statement on whether measurements were taken from distinct samples or whether the same sample was measured repeatedly |
| ☐ | ☒ | The statistical test(s) used AND whether they are one- or two-sided<br>*Only common tests should be described solely by name; describe more complex techniques in the Methods section.* |
| ☒ | ☐ | A description of all covariates tested |
| ☐ | ☒ | A description of any assumptions or corrections, such as tests of normality and adjustment for multiple comparisons |
| ☐ | ☒ | A full description of the statistical parameters including central tendency (e.g. means) or other basic estimates (e.g. regression coefficient) AND variation (e.g. standard deviation) or associated estimates of uncertainty (e.g. confidence intervals) |
| ☐ | ☒ | For null hypothesis testing, the test statistic (e.g. *F*, *t*, *r*) with confidence intervals, effect sizes, degrees of freedom and *P* value noted<br>*Give P values as exact values whenever suitable.* |
| ☒ | ☐ | For Bayesian analysis, information on the choice of priors and Markov chain Monte Carlo settings |
| ☒ | ☐ | For hierarchical and complex designs, identification of the appropriate level for tests and full reporting of outcomes |
| ☒ | ☐ | Estimates of effect sizes (e.g. Cohen's *d*, Pearson's *r*), indicating how they were calculated |

*Our web collection on statistics for biologists contains articles on many of the points above.*

## Software and code

Policy information about availability of computer code

| | |
|---|---|
| Data collection | Imaging data was collected using Zeiss Zen software (Blue edition 2.6 pro)for Zeiss Apotome, Western Blot data was collected using LICOR odyssey system with the Image Studio software(ver 5.2), BioTek Synergy H4 with Gen5 Data Analysis Software (ver 3.11) was used to collect the data of luciferase reporter assay, qPCR data was collected with ABI QuantStudio 5 with QuantStudio Design& Analysis Software(v1.4.3), ISH and Nissl staining data were collected with ZEISS Axioscan and Zeiss Zen software(v3.7 slidescan) |
| Data analysis | All the data were analysed using Excel 2016 (ver 2511) GraphPad Prism 10.6.0, Images were analysed using Zeiss Zen (Blue edition 2.6 pro) or Image J(ver. 2.9.0) software. Single-nucleus RNA-seq data were processed using Cell Ranger (v7.0.1, 10x Genomics) for read alignment and gene counting against the mm10-2020-A reference genome. Ambient RNA contamination and barcode swapping were removed using CellBender (v0.3.0). Downstream analyses were performed primarily in Scanpy (v1.8.1), including highly variable gene selection, differential gene expression analysis, and gene set scoring. Cell clustering and batch correction were carried out using DESC (v2.1.1). Differential expression was assessed using the Wilcoxon rank-sum test implemented in Scanpy. Volcano plots were generated using ggplot2 in R (v3.4.4). Pathway and gene ontology enrichment analyses were performed using the clusterProfiler R package (v4.14.6). Protein–protein interaction analysis was conducted using STRING (v11.5) and then visualized by GOenrich package in clusterProfiler(version 4.14.6) with a network diagram. |

For manuscripts utilizing custom algorithms or software that are central to the research but not yet described in published literature, software must be made available to editors and reviewers. We strongly encourage code deposition in a community repository (e.g. GitHub). See the Nature Portfolio guidelines for submitting code & software for further information.

## Data

Policy information about availability of data

All manuscripts must include a data availability statement. This statement should provide the following information, where applicable:
- Accession codes, unique identifiers, or web links for publicly available datasets
- A description of any restrictions on data availability
- For clinical datasets or third party data, please ensure that the statement adheres to our policy

Single-nucleus RNA sequencing data from E11.5 mouse brains generated in this study have been deposited in the Gene Expression Omnibus (GEO) under accession number GSE314470. Source data are provided with this paper. All other data supporting the findings of this study are available from the corresponding author upon reasonable request.

## Research involving human participants, their data, or biological material

Policy information about studies with human participants or human data. See also policy information about sex, gender (identity/presentation), and sexual orientation and race, ethnicity and racism.

| | |
|---|---|
| Reporting on sex and gender | No |
| Reporting on race, ethnicity, or other socially relevant groupings | No |
| Population characteristics | Postmortem fetal tissue (GW15 and GW17) were used in this study for RNAscope. For GW15 sample, the age is 15 Gestational Weeks, PMI 15 hours, Clinical history: Cervical insufficiency, Neuropathological diagnosis: Control. For GW17 sample, the age is 17 Gestational Weeks, PMI 1hour 20mins, Clinical history: Hypoplastic left heart syndrome, Neuropathological diagnosis: Control. Listed as Case No.5 and Case No.6 from published paper(PMID: 35084970). |
| Recruitment | Human specimens were collected from autopsy, with previous patient consent to institutional ethical regulations of the University of California San Francisco Committee on Human Research as previously reported(PMID: 26798014, PMID: 35084970), and the sample been used in the published paper and listed as Case No.5 and Case No.6(PMID: 35084970). |
| Ethics oversight | Human specimens were collected from autopsy, with previous patient consent to institutional ethical regulations of the University of California San Francisco Committee on Human Research as previously reported(PMID: 26798014, PMID: 35084970) |

Note that full information on the approval of the study protocol must also be provided in the manuscript.

# Field-specific reporting

Please select the one below that is the best fit for your research. If you are not sure, read the appropriate sections before making your selection.

☒ Life sciences ☐ Behavioural & social sciences ☐ Ecological, evolutionary & environmental sciences

For a reference copy of the document with all sections, see nature.com/documents/nr-reporting-summary-flat.pdf

# Life sciences study design

All studies must disclose on these points even when the disclosure is negative.

| | |
|---|---|
| Sample size | Minimum sample sizes were estimated based on previously published studies(PMID: 30787442, PMID: 41339559), and our experiences(PMID: 26798014, PMID: 36384142). Normally, at least 3 samples per group were used for statistical analyses. |
| Data exclusions | No exclusions |
| Replication | For all the graphical representations of the data, the number of and types of replicates used are mentioned in their respective figure legends and methods. All the experiments were performed with 3 or more biological replicates. |
| Randomization | Mice were assigned randomly to the experimental groups. For non-mouse experiments, samples were assigned to different experimental groups according to pre-defined experimental conditions. All samples were processed in parallel under identical conditions. To minimize potential batch effects, the order of sample assignment and data acquisition was balanced across groups. |
| Blinding | Blinding was employed for all analysis. During the data collection process, researchers were unaware of the group assignments. Samples were independently coded, and data collection was performed without knowledge of the genotype or experimental conditions. Group assignment information was only revealed after data collection and initial quantitative analysis were completed. |

# Reporting for specific materials, systems and methods

We require information from authors about some types of materials, experimental systems and methods used in many studies. Here, indicate whether each material, system or method listed is relevant to your study. If you are not sure if a list item applies to your research, read the appropriate section before selecting a response.

## Materials & experimental systems

| n/a | Involved in the study |
|-----|-----------------------|
| ☐ | ☒ Antibodies |
| ☐ | ☒ Eukaryotic cell lines |
| ☒ | ☐ Palaeontology and archaeology |
| ☐ | ☒ Animals and other organisms |
| ☒ | ☐ Clinical data |
| ☒ | ☐ Dual use research of concern |
| ☒ | ☐ Plants |

## Methods

| n/a | Involved in the study |
|-----|-----------------------|
| ☒ | ☐ ChIP-seq |
| ☒ | ☐ Flow cytometry |
| ☒ | ☐ MRI-based neuroimaging |

## Antibodies

Antibodies used

For the immunostaining: GFP was detected with Anti-Green Fluorescent Protein (GFP) Antibody GFP-1020 (Aves) at 1:1000 dilution, mouse Pax6 was detected with Anti-PAX6 Antibody AB2237 (Millipore) at 1:500 dilution, mouse Tbr2 was detected with Anti-TBR2 / Eomes antibody ab23345 (Abcam) at 1:500 dilution, mouse Tbr1 was detected with Anti-TBR1 antibody ab31940 (Abcam) at 1:1000 dilution, mouse Tuj1 was detected with Anti-β-Tubulin III Antibody T2200 (Sigma) at 1:1000 dilution, mouse Satb2 was detected with Anti-SATB1 + SATB2 antibody [SATBA4B10] - C-terminal ab51502 (Abcam) at 1:500 dilution, mouse Ctip2 was detected with Anti-Ctip2 antibody [25B6] ab18465 (Abcam) at 1:1000 dilution, mouse Cux1+Cux2 was detected with Anti-CUX1+CUX2 antibody [EPR26509-154] ab309139 (Abcam) at 1:500 dilution, mouse Calretinin was detected with Anti-Calretinin Antibody, clone 6B8.2 MAB1568 (Millipore) at 1:500 dilution, mouse Parvalbumin was detected with Anti-Parvalbumin Antibody MAB1572 (Millipore) at 1:500 dilution, mouse Calbindin was detected with Anti- Calbindin antibody CB38a (Swant) at 1:500 dilution, human PAX6 was detected with anti-Pax-6 Antibody 901301 (Biolegend) at 1:500 dilution, mouse gamma H2A.X was detected with Anti-gamma H2A.X (phospho S139) antibody ab2893 (abcam) at 1:500 dilution or Anti-phospho-Histone H2A.X (Ser139) Antibody, clone JBW301 05-636 (Millipore) at 1:500 dilution, mouse Cleaved Caspase-3 was detected with Anti-Cleaved Caspase-3 (Asp175) Antibody 9661(cell signaling technology) at 1:400 dilution, mouse Phospho-KAP-1 (Ser824) was detected with Anti-Phospho-KAP-1 (Ser824) Polyclonal antibody A300-767A (Bethyl Laboratories) at 1:1000 dilution, mouse P53 was detected with Anti-p53 (1C12) Mouse Monoclonal Antibody 2524 (cell signaling technology) at 1:500 dilution, mouse PCNA was detected with Anti-PCNA (PC10) Mouse Monoclonal Antibody 2586 (cell signaling technology) at 1:500 dilution, mouse Ki67 was detected with antibody 550609 (BD Biosciences) at 1:500 dilution, mouse 53bp1 was detected with Anti-53BP1 Antibody NB100-304 (Novus Biologicals) at 1:500 dilution, mouse DNA-RNA Hybrid S9.6 was detected with Anti-DNA-RNA Hybrid [S9.6] Antibody ENH001 (Kerafast) at 1:500 dilution, mouse p-ATM(Ser1981) was detected with Anti-phospho-ATM (Ser1981) Antibody, clone 10H11.E12 05-740 (Millipore Sigma) at 1:500 dilution. Mouse phh3 was detected with Phospho-Histone H3 (Ser10) Antibody 9701 (cell signaling technology) at 1:500 dilution. Mouse Nestin was detected with Anti-Nestin Antibody, clone rat-401 MAB353 (Millipore Sigma) at 1:500 dilution. Mouse Sox2 was detected with Anti-SOX2 antibody [EPR3131] ab92494 (abcam) at 1:500 dilution.

For immunoblotting: Beta-Actin was detected with Anti-Beta Actin Monoclonal antibody 66009-1-Ig (proteintech) at 1:1000 dilution, Ebf1 was detected with Anti-EBF-1 Antibody AB10523 (Millipore) at 1:500 dilution, UBA52 was detected with Anti-UBA52 Polyclonal antibody 18039-1-AP (proteintech) at 1:500 dilution, Cirbp was detected with CIRBP Polyclonal antibody 10209-2-AP (proteintech) at 1:500 dilution, Atf4 was detected with Anti-ATF-4 (D4B8) Rabbit Monoclonal Antibody 11815 (cell signaling technology) at 1:500 dilution.

For ChIP-qPCR: Anti-ATF-4 (D4B8) Rabbit Monoclonal Antibody 11815 (cell signaling technology) used at 1:50 dilution.

For the ISH experiment, Anti-Digoxigenin-AP, Fab fragments been used at 1;1500 dilution, 11093274910(ROCHE)

Secondary antibodies:
Goat anti-Chicken IgY (H+L) Secondary Antibody, Alexa Fluor™ 488 ; A11039; 1:1000 dilution; Invitrogen
Goat anti-Mouse IgG (H+L) Cross-Adsorbed Secondary Antibody, Alexa Fluor 488; A11001; 1:1000 dilution; Invitrogen
Goat anti-Mouse IgG (H+L) Cross-Adsorbed Secondary Antibody, Alexa Fluor 594; A11005; 1:1000 dilution; Invitrogen
Goat anti-Mouse IgG (H+L) Secondary Antibody, Alexa Fluor® 647 conjugate; A21236; 1:1000 dilution; Invitrogen
Goat anti-Rabbit IgG (H+L) Cross-Adsorbed Secondary Antibody, Alexa Fluor 488; A11008; 1:1000 dilution; Invitrogen
Goat anti-Rabbit IgG (H+L) Cross-Adsorbed Secondary Antibody, Alexa Fluor 594; A11012; 1:1000 dilution; Invitrogen
Goat anti-Rabbit IgG (H+L) Highly Cross-Adsorbed Secondary Antibody, Alexa Fluor 647; A21245; 1:1000 dilution; Invitrogen
Goat anti-Rat IgG (H+L) Cross-Adsorbed Secondary Antibody, Alexa Fluor™ 488, Invitrogen™; A11006; 1:1000 dilution; Invitrogen
Goat anti-Rat IgG (H+L) Cross-Adsorbed Secondary Antibody, Alexa Fluor 594; A11007; 1:1000 dilution; Invitrogen
Goat anti-Rat IgG (H+L) Secondary Antibody, Alexa Fluor® 647 conjugate; A21247; 1:1000 dilution; Invitrogen
Highly Cross-Adsorbed Goat (Polyclonal) Anti-Mouse IgG (H+L) Antibody Conjugated to IRDye 680RD; 926-68070; 1;20000 dilution; LICORbio;
Highly cross-adsorbed goat (polyclonal) anti-rabbit IgG (H+L) antibody conjugated to IRDye 680RD; 926-68071; ;20000 dilution; LICORbio;

Validation

All of the antibodies used were chosen from published research and have been validated by the manufacturer for the specific species and application. All information is listed here:

1: Anti-Green Fluorescent Protein (GFP) GFP-1020 (Aves): https://www.antibodiesinc.com/products/anti-green-fluorescent-protein-antibody-gfp?utm_source=citeab&utm_medium=affiliate&utm_campaign=product

2: Anti-PAX6 Antibody AB2237 (Millipore) https://www.sigmaaldrich.com/US/en/product/mm/ab2237

3: Anti-TBR2 / Eomes antibody ab23345 (Abcam)
https://www.abcam.com/en-us/products/primary-antibodies/tbr2-eomes-antibody-ab23345

4: Anti-TBR1 antibody ab31940 (Abcam)
https://www.abcam.com/en-us/products/primary-antibodies/tbr1-antibody-ab31940

5: Anti-β-Tubulin III Antibody T2200 (Sigma)
https://www.sigmaaldrich.com/US/en/product/sigma/t2200?srsltid=AfmBOopCTJbYTjkeTDPBEMiUK0Cz3DnWg1J7rHukdfANqxpWUvW9QC0f

6: Anti-SATB1 + SATB2 antibody [SATBA4B10] - C-terminal ab51502 (Abcam)
https://www.abcam.com/en-us/products/primary-antibodies/satb1-satb2-antibody-satba4b10-c-terminal-ab51502

7: Anti-Ctip2 antibody [25B6] ab18465 (Abcam)
https://www.abcam.com/en-us/products/primary-antibodies/ctip2-antibody-25b6-ab18465

8: Anti-CUX1+CUX2 antibody [EPR26509-154] ab309139 (Abcam)
https://www.abcam.com/en-us/products/primary-antibodies/cux1cux2-antibody-epr26509-154-ab309139

9: Anti-Calretinin Antibody, clone 6B8.2 MAB1568 (Millipore)
https://www.sigmaaldrich.com/US/en/product/mm/mab1568?srsltid=AfmBOoourzDuL7C-QliiWfRK2-JH0fQooDbIJYninIQpWAM0IlQXDSrf

10: Anti-Parvalbumin Antibody MAB1572 (Millipore)
https://www.sigmaaldrich.com/US/en/product/mm/mab1572?srsltid=AfmBOooU7Sxl7jSfVjjXQFlheVcVkUsgYsyeXsPk0xiQDntEyjKsz3xB

11: Anti- Calbindin antibody CB38a (Swant)
https://shop.swant.com/cb38a-calbindin.html

12: anti-Pax-6 Antibody 901301 (Biolegend)
https://www.biolegend.com/nl-nl/products/purified-anti-pax-6-antibody-11511?displayInline=true&filename=Purified%20anti-Pax-6%20Antibody.pdf&leftRightMargin=15&pdf=true&topBottomMargin=15&v=20250227010954

13: Anti-gamma H2A.X (phospho S139) antibody ab2893 (abcam)
https://www.abcam.com/en-us/products/primary-antibodies/gamma-h2ax-phospho-s139-antibody-ab2893

14: Anti-phospho-Histone H2A.X (Ser139) Antibody, clone JBW301 05-636 (Millipore)
https://www.sigmaaldrich.com/US/en/product/mm/05636?srsltid=AfmBOor46JY6MrIeXmATEXcY7DRo2Pq2QBdlMlRh7OE-UxNWKL7g7esd

15: Anti-Cleaved Caspase-3 (Asp175) Antibody 9661(cell signaling technology)
https://www.cellsignal.com/products/primary-antibodies/cleaved-caspase-3-asp175-antibody/9661?srsltid=AfmBOorED8btjid1f1KdGY2raxGyMt4VrQqxOkhxT0sl9VmpAit1staw

16: Anti-Phospho-KAP-1 (Ser824) Polyclonal antibody A300-767A (Bethyl Laboratories)
https://www.thermofisher.com/antibody/product/Phospho-KAP-1-Ser824-Antibody-Polyclonal/A300-767A

17: Anti-p53 (1C12) Mouse Monoclonal Antibody 2524 (cell signaling technology)
https://www.cellsignal.com/products/primary-antibodies/p53-1c12-mouse-monoclonal-antibody/2524?srsltid=AfmBOoqgnOE0bWhEDqWv1v73fciS9mN6Iiq7aK-AOpDx2XrcYRQjZHpj

18: Anti-PCNA (PC10) Mouse Monoclonal Antibody 2586 (cell signaling technology)
https://www.cellsignal.com/products/primary-antibodies/pcna-pc10-mouse-monoclonal-antibody/2586?srsltid=AfmBOooJR-o0wKAsUyA9k5JYHNoOB4XAdFiW1HHGoZO61XaVRugrTKb0

19: Anti-53BP1 Antibody NB100-304 (Novus Biologicals)
https://www.novusbio.com/products/53bp1-antibody_nb100-304?srsltid=AfmBOoqXirQRpT3GLjGQZ7PD_PNLhPZwRUC4r-jWVaKtFn9R3H1pnceA

20: Anti-DNA-RNA Hybrid [S9.6] Antibody ENH001 (Kerafast)
https://www.kerafast.com/productgroup/432/anti-dna-rna-hybrid-s96-antibody

21: Anti-phospho-ATM (Ser1981) Antibody, clone 10H11.E12 05-740 (Millipore Sigma)
https://www.sigmaaldrich.com/US/en/product/mm/05740?srsltid=AfmBOoqwv_9ymkVKzucidMuPt1UgRxMOZkXurWdxhwedwV6CY7uu8Xh8

22: Phospho-Histone H3 (Ser10) Antibody 9701 (cell signaling technology)
https://www.cellsignal.com/products/primary-antibodies/phospho-histone-h3-ser10-antibody/9701?srsltid=AfmBOor-gfGf97hsjIB4UH9EJ8aH2zrX4RSr1ZJx68YW94JhkBRSui5n

23: Anti-Nestin Antibody, clone rat-401 MAB353 (Millipore Sigma)
https://www.sigmaaldrich.com/US/en/product/mm/mab353?srsltid=AfmBOopoxvMbH0d7rpdoxh9-gYXjGkttSJ8h18j5Yq8RxRQWJ44AKqW-

24: Anti-SOX2 antibody [EPR3131] ab92494 (abcam)
https://www.abcam.com/en-us/products/primary-antibodies/sox2-antibody-epr3131-ab92494

25: Anti-Beta Actin Monoclonal antibody 66009-1-Ig (proteintech)
https://www.ptglab.com/products/Pan-Actin-Antibody-66009-1-Ig.htm?srsltid=AfmBOorrHJR5l4OquiWo9mFX7RN3L4BFWQfzDs3VdfROVqN_H21vZYvl

26: Anti-EBF-1 Antibody AB10523 (Millipore)

https://www.sigmaaldrich.com/US/en/product/mm/ab10523?
srsltid=AfmBOorckej_8O8PUbDeZ6iGNipTPzdi0yxmetE88QFMk_90aWjNrUG4
27: Anti-UBA52 Polyclonal antibody 18039-1-AP (proteintech)
https://www.ptglab.com/products/UBA52-Antibody-18039-1-AP.htm?
srsltid=AfmBOoqKzpFGxM2V8uvh_sMW0vo4G4YsV2oS3JPt7MRQTFZCnGGOGTOE
28: CIRBP Polyclonal antibody 10209-2-AP (proteintech)
https://www.ptglab.com/products/CIRBP-Antibody-10209-2-AP.htm?
srsltid=AfmBOookLo0lo4_mNkeGy9sgVA3nPNoVt8fpWTJ_LkWJjzwJu0lTKRp1
29: Anti-ATF-4 (D4B8) Rabbit Monoclonal Antibody 11815 (cell signaling technology)
https://www.cellsignal.com/products/primary-antibodies/atf-4-d4b8-rabbit-monoclonal-antibody/11815?
srsltid=AfmBOop2flu7FRJfrn5CJwrmp5LgeflJBzzy8GfvpundmXHyebMoJ5yE
30: Anti-Digoxigenin-AP, Fab fragments, 11093274910(ROCHE)
https://www.sigmaaldrich.com/US/en/product/roche/11093274910?srsltid=AfmBOoql-NORqVQ2jaa9uuU9GMv55yWRynlBx5-
Jrfae240YHtWd-jgf
31: Goat anti-Chicken IgY (H+L) Secondary Antibody, Alexa Fluor™ 488 ; A11039
https://www.thermofisher.com/antibody/product/A11039.html?ef_id=Cj0KCQiAkPzLBhD4ARIsAGfah8hiVZeF-
LweJHmhyhiF9AMb_adQY-kAtLIIM7zVwEgEL6TTyCbmteoaAt1xEALw_wcB:G:s&s_kwcid=AL!3652!3!!!!x!!
&gad_source=1&gad_campaignid=23463429566&gbraid=0AAAAADxi_GRlQD99T3m6b32wsvWiftVl1&gclid=Cj0KCQiAkPzLBhD4ARIsA
Gfah8hiVZeF-LweJHmhyhiF9AMb_adQY-kAtLIIM7zVwEgEL6TTyCbmteoaAt1xEALw_wcB
31: Goat anti-Mouse IgG (H+L) Cross-Adsorbed Secondary Antibody, Alexa Fluor 488; A11001
https://www.thermofisher.com/antibody/product/Goat-anti-Mouse-IgG-H-L-Cross-Adsorbed-Secondary-Antibody-Polyclonal/
A-11001
32: Goat anti-Mouse IgG (H+L) Cross-Adsorbed Secondary Antibody, Alexa Fluor 594; A11005
https://www.thermofisher.com/antibody/product/Goat-anti-Mouse-IgG-H-L-Cross-Adsorbed-Secondary-Antibody-Polyclonal/
A-11005
33: Goat anti-Mouse IgG (H+L) Secondary Antibody, Alexa Fluor® 647 conjugate; A21236
https://www.thermofisher.com/antibody/product/Goat-anti-Mouse-IgG-H-L-Highly-Cross-Adsorbed-Secondary-Antibody-Polyclonal/
A-21236
34: Goat anti-Rabbit IgG (H+L) Cross-Adsorbed Secondary Antibody, Alexa Fluor 488; A11008
https://www.thermofisher.com/antibody/product/Goat-anti-Rabbit-IgG-H-L-Cross-Adsorbed-Secondary-Antibody-Polyclonal/
A-11008
35: Goat anti-Rabbit IgG (H+L) Cross-Adsorbed Secondary Antibody, Alexa Fluor 594; A11012
https://www.thermofisher.com/antibody/product/Goat-anti-Rabbit-IgG-H-L-Cross-Adsorbed-Secondary-Antibody-Polyclonal/
A-11012
36: Goat anti-Rabbit IgG (H+L) Highly Cross-Adsorbed Secondary Antibody, Alexa Fluor 647; A21245
https://www.thermofisher.com/antibody/product/Goat-anti-Rabbit-IgG-H-L-Highly-Cross-Adsorbed-Secondary-Antibody-Polyclonal/
A-21245
37: Goat anti-Rat IgG (H+L) Cross-Adsorbed Secondary Antibody, Alexa Fluor™ 488, Invitrogen™; A11006
https://www.thermofisher.com/antibody/product/Goat-anti-Rat-IgG-H-L-Cross-Adsorbed-Secondary-Antibody-Polyclonal/A-11006
38: Goat anti-Rat IgG (H+L) Cross-Adsorbed Secondary Antibody, Alexa Fluor 594; A11007
https://www.thermofisher.com/antibody/product/Goat-anti-Rat-IgG-H-L-Cross-Adsorbed-Secondary-Antibody-Polyclonal/A-11007
39: Goat anti-Rat IgG (H+L) Secondary Antibody, Alexa Fluor® 647 conjugate; A21247
https://www.thermofisher.com/antibody/product/Goat-anti-Rat-IgG-H-L-Cross-Adsorbed-Secondary-Antibody-Polyclonal/A-21247
40: Highly Cross-Adsorbed Goat (Polyclonal) Anti-Mouse IgG (H+L) Antibody Conjugated to IRDye 680RD; 926-68070; 1;20000
dilution; LICORbio;
https://www.licorbio.com/support/contents/reagents/irdye-secondary-antibodies/680rd/goat-anti-mouse-igg.html
41: Highly cross-adsorbed goat (polyclonal) anti-rabbit IgG (H+L) antibody conjugated to IRDye 680RD; 926-68071
https://shop.licorbio.com/irdye-secondary-antibodies/irdye-680rd-goat-anti-rabbit-igg-secondary-antibody/

# Eukaryotic cell lines

Policy information about cell lines and Sex and Gender in Research

| Cell line source(s) | HEK293T (ATCC #CRL-3216) cell line was used for lenti-virus packaging. The 293T cell line, originally referred as 293tsA1609neo, is a highly transfectable derivative of human embryonic kidney 293 cells, and contains the SV40 T-antigen. |
|---|---|
| Authentication | HEK293T (ATCC #CRL-3216) cell line identity was authenticated by short tandem repeat (STR) profiling prior to use. STR profiling by ATCC(https://www.atcc.org/products/crl-3216). |
| Mycoplasma contamination | Lines tested negative for mycoplasma contamination |
| Commonly misidentified lines (See ICLAC register) | No. |

# Animals and other research organisms

Policy information about studies involving animals; ARRIVE guidelines recommended for reporting animal research, and Sex and Gender in Research

| Laboratory animals | This study used mice at various developmental stages and includes both males and females.<br><br>The mice age in between 2 to 6 months old were used for the breeding. |
|---|---|

All mice were handled in accordance with NIH guidelines and protocols approved by the UCSF Institutional Animal Care and Use Committee. Mice were housed under specific pathogen-free conditions in individually ventilated cages within a barrier facility on a 12-h light/dark cycle, with controlled temperature (68–79 °F) and humidity (30–70%). Housing density did not exceed five adult mice per cage; breeding cages (one male, up to two females) were maintained in a dedicated high-barrier area. Cages were changed weekly under laminar flow hoods, access was restricted with required PPE, and colony health was monitored using sentinel mice. Both sexes were used, no sex-specific differences were observed, and mice were randomly assigned to experimental groups.

C57BL/6 wild-type (WT) mice were obtained from The Jackson Laboratory (JAX:000664).

The Emx1-Cre line (B6.129S2Emx1tm1(cre)Krj/J, JAX:005628) has been previously described. These mice were crossed with Atf4 floxed mice to delete Atf4 specifically in the early embryonic cortex. To assess phenotypes after blocking cell death, Emx1-Cre mice were also crossed with Atf4 floxed and p53 null animals. In addition, Emx1-Cre mice were crossed with LSL-H2B-GFP mice for lineage tracing of Emx1+ cortical cells across developmental stages.

The Atf4fl/fl line (C57BL/6-Atf4tm1.1Cmad/J, JAX:033380) carries loxP sites flanking exons 2–3, which include the ATG start codon of the Atf4 gene. These mice have been previously described and were crossed with Emx1-Cre and/or p53 null mice to knockout the Atf4 expression.

The p53 null (p53-/-) line (B6.129S2-Trp53tm1Tyj/J, JAX:002101) carries a neomycin cassette replacing exons 2–6 (including the start codon) of the Trp53 gene. These mice were crossed with Emx1-Cre and Atf4 floxed mice to block p53-dependent cell death.
The LSL-H2B-GFP line (B6.Cg-Gt(ROSA)26Sortm8(CAG-HIST1H2BB/EGFP)Zjh/J JAX:036761) harbors a targeted mutation in the Gt(ROSA)26Sor locus with a loxP-flanked STOP cassette preventing transcription of a CAG promoter–driven enhanced green fluorescent protein (EGFP).

IUE experiment was done with time-plugged pregnant animal with ECM BTX830, according to previous published paper(PMID: 28524856).

Embryonic brains were collected from timed-pregnant mice at the indicated stages, and genotyping was performed using tissue from the same embryos.
For neonatal samples, the day of birth was designated postnatal day 0 (P0), and brains were collected at the indicated postnatal stages with genotyping performed on tissue from the same animals.

| | |
|---|---|
| Wild animals | NO |
| Reporting on sex | Animals were used unbaised to sex. |
| Field-collected samples | No |
| Ethics oversight | All animal protocols were in accordance with the regulations of the National Institute of Health and approved by the University of California San Francisco Institutional Animal Care and Use Committee (IACUC). |

Note that full information on the approval of the study protocol must also be provided in the manuscript.

# Plants

| | |
|---|---|
| Seed stocks | No |
| Novel plant genotypes | No |
| Authentication | No |

