## [Peer Review file · Nature]

Expansion of outer cortical Cux2+ neurons requires selective adaptations for DNA repair

Corresponding Author: Professor Stephen Fancy

Version 0:

Reviewer comments:

Referee #1

(Remarks to the Author)

A. Summary of the key results

The authors report that during mammalian brain development, upper cortical layer 2/3 excitatory neurons expand significantly, making them vulnerable to DNA damage from replicative stress. This study identifies Activating Transcription Factor 4 (Atf4) as a key regulator of the DNA damage response, particularly in promoting DNA repair genes. Loss of Atf4 impairs the development of Cux2+ layer 2/3 neurons by increasing DNA damage and cell death in neural progenitors. The findings highlight Atf4's essential role in the DNA damage response in these neurons during cortical development. Overall this is a well-performed paper that highlights the important role of DNA repair in upper layer 2/3 neurons.

B. Originality and significance: if not novel, please include reference

The originality and significance of this abstract lies in identifying a novel, layer-specific role for Atf4 in regulating DNA damage response pathways critical for the development and survival of upper cortical layer 2/3 neurons, highlighting a unique molecular vulnerability during mammalian cortical expansion.

C. Data & methodology: validity of approach, quality of data, quality of presentation

The authors use a number of mouse models and experimental assays to address this question with high data quality.

D. Appropriate use of statistics and treatment of uncertainties

No issues.

E. Conclusions: robustness, validity, reliability

No issues.

F. Suggested improvements: experiments, data for possible revision

In the section in methods, "EdU lableing and staining" labeling is spelt incorrectly and should be labeling, not lableing. In the section Single nucleus RNA-seq Alignment and Filtering, the text states cellranger count pipeline provided by 10x Genomics (version X.Y.Z). The version should be written out not as version X.Y.Z.

In the text "Cell Bender (version X.Y.Z)" the version should be written out, not as version X.Y.Z.

"For highly_variable_genes function in scanpy (version Y.Y.Y), the version should be written out not as y.y.y.

For the curated "Human DNA Repair Genes" list by R. Wood and M. Lowery, the list should be provided or the reference to the source.

Nrf2 Target Gene Set: This set includes 37 genes identified as Nrf2 targets, based on a curated list from a published review article. The authors should list the genes.

For the RNAscope section in methods, the tense of methods should be written out, see "If combining RNAscope with immunostaining for various antibodies is needed, the standard immunohistochemistry (IHC) protocol should be followed."

Please write out the names as much as possible for example, write out DNA Damage Response each time rather than DDR and double strand breaks rather than DSB, neural progenitors rather than NPs, Neural Stem Cells rather than NSCs, reactive oxygen species rather than ROS, Unfolded Protein Response rather than UPR, excitatory neurons rather than ENs, integrated Stress Response rather than ISR, ventricular zone rather than VZ. Many readers outside the field may not be familiar with the jargon used and it would make the manuscript easier to read to write out the words in full each time.

All the mouse lines should be described in the methods. For example, in figure 1a and extended data fig 1 the mouse line LSL-H2B-GFP was used; however, it's not defined what LSL is in the text of the manuscript, the figure legend, and the source for this mouse line should be provided in methods.

For the line "or p53 related genes upregulated in NPs in the absence of Atf4, such as Cdkn1a Eda2r," there should be a comma after Cdkn1a.

For the line, "We also examined the RG sequencing data". Please define what RG stands for.

In the UMAP plot there are clusters RG1, RG2, RG3, RG4. Please define in the manuscript or the figure legend what RG stands for? Similarly it should be defined in the figure legend or manuscript what Ext1 and Int1 clusters refer to.

Overall these are minor points and the paper is outstanding and I would be supportive of publication if the authors address them.

Referee #3

(Remarks to the Author)

Expansion of outer cortical layer Cux2+ neurons required selective adaptations for DNA repair

Summary/Key Results:

The authors highlight that the expansion of the cortex, especially of Layers 2/3, correlates with advances in cognition throughout evolution. This rapid expansion during embryonic development subjects these neural progenitors to replication stress and oxidative damage/ROS stemming from elevated cellular metabolism. While neural progenitors of the cortex likely have to overcome these insults, little is known about the maintenance of genomic integrity of the neural progenitor pool. Furthermore, whether progenitors are fate restricted to become L2/3 neurons or influenced by niche and birthdate remains unknown. L2/3 neurons are less viable in multiple sclerosis (MS) and disproportionately affected in autism, suggesting a unique vulnerability of this population. Thus, the authors investigate what conditions lead to this selective vulnerability. They identify that knockout of activating transcription factor 4 (Atf4) in all embryonic precursors of upper layer cortical neurons leads to increased DNA double stranded breaks (DSBs) and apoptosis selectively in L2/3 neurons, supporting a unique vulnerability of this population. The authors demonstrate that L2/3 neuronal loss in Atf4 knockout (KO) mice is p53 dependent. Finally, they demonstrate that Atf4 KO leads to changes in transcripts for the DNA damage response which are capable of partially rescuing L2/3 cell death when upregulated. The authors identify a novel role for Atf4 not only in regulation of L2/3 cortical expansion, but also as a transcriptional regulator of DDR. Furthermore, the authors conclude that neural progenitors (NP) of L2/3 are prepatterned, rather than influenced by their niche and birthdate. Thus, the evolutionary expansion of the cortex is facilitated by the recruitment of DNA repair proteins in select progenitors which give rise to L2/3 neurons.

The authors present convincing results demonstrating the susceptibility of L2/3 neurons to Atf4 KO, indicating an essential role of Atf4 to neural progenitor survival, and they clearly demonstrate that the selective loss of Atf4 knockout neurons is p53-dependent. Beyond these findings, however, many of the authors' conclusions are not convincingly substantiated, which detracts from the significance of this work. Major issues include lack of substantial evidence for the mechanisms contributing to DNA damage and L2/3 susceptibility to apoptosis, mechanisms that underly the dependence of L2/3 on ATF4, lack of direct evidence for the evolutionary relevance of this pathway to humans/its role in cortical expansion, and lack of data demonstrating the consequences of the observed L2/3 cortical phenotype on brain function and behavior.

Major Concerns

General

1. The disappearance of L2/3 neurons in ATF4 knockouts looks very convincing as is its dependence on P53-induced apoptosis. However, the claim that this is due to disrupted ATF4-dependent DNA damage repair (DDR) resulting in DSB and apoptosis can be challenged for several reasons:
 - protein translation is severely disrupted in ATF4 KO, which can suffice to induce DSB, DDR down-regulation, and apoptosis
 - although ROS is referred to as a candidate DSB mechanism, there is no evidence for higher oxidative stress in L2/3 NPs relative to other neurons at a comparative proliferation stage
 - many established DDR genes are up-regulated
 - there is no direct demonstration that the candidate DDR genes (Uba52, Cirbp, and Ebf1) can also induce L2/3 cortical

thinning

- these genes have many functions other than DDR, and most are not DDR genes themselves but regulators of DDR.

2. ATF4 is ubiquitously expressed in neurons, especially in interneurons. Mechanisms for the particular vulnerability of L2/3 neurons to ATF4 KO are not demonstrated and the underlying causes of their vulnerability remain unknown. It is not clear, for example, whether ATF4 is needed for the expression of the aforementioned DDR-related genes only in L2/3, or, it also controls them in other neuron types, which tolerate these effects better.

3. This mouse model offers a unique opportunity to establish the significance of L2/3 loss. However, the consequences of this loss on brain function and behavior are completely unexplored.

Significance

1. L2/3 structural and functional deficits have been linked to a range of neurodevelopmental, neurodegenerative, and psychiatric (schizophrenia in particular). The association with MS has been the least well established, with some reports highlighting L5 rather than L2/3 pathologies. It remains unclear why only MS and autism are mentioned and why there was no effort to make associations to disease-relevant phenotypes, especially during adolescence, when much of the cortical analyses were performed.

2. In studies that do report L2/3 changes, they do not appear homogeneously distributed throughout the cortex. Regional differences are not addressed in the paper, and it is not clear in which cortical area were the measurements performed.

Methodology

1. In Figure 2, the authors use H2AX and pKAP1 to label DNA DSBs. While this provides clear evidence of DNA break induction, the amount of H2AX+ cells in representative images appears too sparse to induce the extreme atrophy observed in L2/3 of Atf4 KO mice, nor does it constitute “overwhelming DNA damage” as described. For example, in figure 2g, only 30% of Pax6+ cells are gH2AX+. Furthermore, evidence is needed to support the claim that DNA DSBs are the cause of cell death, or further investigation into other mechanisms inducing apoptosis is necessary.

2. Figure 4j indicates the upregulation of many DDR related GO terms in the Atf4 KO. While this demonstrates alteration of the DNA damage response in the KO model, as stated by the authors, it weakens the claim that Atf4 is needed to facilitate DDR.

Conclusions

1. The conclusion that impaired DDR is the main cause of excessive DSB and apoptosis is not directly demonstrated and alternatives (eg inhibited translation) are a strong alternative

2. The final claim that evolutionary expansion of the cortex requires recruitment of DNA repair proteins is unsubstantiated. In figure 2b-c, the authors demonstrate expression of Atf4 at GW15 and 17 in human tissue. However, the presence of Atf4 is not enough to claim that Atf4 driven DNA damage response contributes to evolutionary cortical expansion.

3. The authors claim that the evidence presented suggests that Cux2+ NP are pre-patterned to become L2/3 cortical neurons. Despite a significant number of Cux2+gH2AX+ NPs (60%) following Atf4 KO, suggesting a unique vulnerability of this population, there is not enough evidence to claim they are predestined.

Minor Concerns

1. The rationale for investigating Atf4 is unclear. Although Atf4 is known to regulate several cellular stress response pathways, the authors openly state that there is little evidence for Atf4 in the regulation of genomic stability.

2. The sentence in the introduction stating DNA DSBs are the most common lesion in neurogenesis is not cited.

Referee #4

(Remarks to the Author)

I co-reviewed this manuscript with one of the reviewers who provided the listed reports.

Referee #5

(Remarks to the Author)

I co-reviewed this manuscript with one of the reviewers who provided the listed reports.

Version 1:

Reviewer comments:

Referee #1

(Remarks to the Author)

This is an important work that is complementary to the accompanying more clinically related MS paper showing that Atf4 is a critical regulator of the DNA Damage Response, directly activating components of double stranded DNA repair

All of the comments have been properly addressed

Referee #3

(Remarks to the Author)

The authors adequately showed that neural progenitors destined to become L2/3 Cux2+ neurons acquired an ATF4 dependent mechanism to mediate DNA damage. Additionally, the decrease in phosphoATM and regulation of C/EBP which mimics decrease in phospho ATM is a novel and convincing mechanism. All major concerns are addressed with new evidence or sufficient explanation.

Minor concerns:

1. How do the authors rationalize the increase in DDR genes? Especially since ATM is downregulated, what is mediating the DNA damage response?

This was somewhat addressed in response to reviewers, Methodology point 2. I believe it could be expanded.

2. In Fig2, ATM labeling does not overlap with gH2AX labeling. Is that a concern?

3. How was the statistics for snRNAseq performed? There are n = 3 biological replicates, but it is unclear whether this was accounted for using pseudobulk or other approaches in their analyses?

Referee #4

(Remarks to the Author)

I co-reviewed this manuscript with one of the reviewers who provided the listed reports.

Referee #5

(Remarks to the Author)

I co-reviewed this manuscript with one of the reviewers who provided the listed reports.

made.

Response to Reviewers

We very sincerely thank all reviewers for their constructive and insightful comments. These have helped us to significantly strengthen this paper. All changes to the main manuscript are now written in blue. Below is our point by point response to comments:

Referees' comments:

Referee #1 (Remarks to the Author):

A. Summary of the key results

The authors report that during mammalian brain development, upper cortical layer 2/3 excitatory neurons expand significantly, making them vulnerable to DNA damage from replicative stress. This study identifies Activating Transcription Factor 4 (Atf4) as a key regulator of the DNA damage response, particularly in promoting DNA repair genes. Loss of Atf4 impairs the development of Cux2+ layer 2/3 neurons by increasing DNA damage and cell death in neural progenitors. The findings highlight Atf4's essential role in the DNA damage response in these neurons during cortical development. Overall this is a well-performed paper that highlights the important role of DNA repair in upper layer 2/3 neurons.

B. Originality and significance: if not novel, please include reference

The originality and significance of this abstract lies in identifying a novel, layer-specific role for Atf4 in regulating DNA damage response pathways critical for the development and survival of upper cortical layer 2/3 neurons, highlighting a unique molecular vulnerability during mammalian cortical expansion.

C. Data & methodology: validity of approach, quality of data, quality of presentation

The authors use a number of mouse models and experimental assays to address this question with high data quality.

D. Appropriate use of statistics and treatment of uncertainties

No issues.

E. Conclusions: robustness, validity, reliability

No issues.

F. Suggested improvements: experiments, data for possible revision

In the section in methods, "EdU lableing and staining" labeling is spelt incorrectly and should be labeling, not lableing.

Response: We have corrected this in the text.

In the section Single nucleus RNA-seq Alignment and Filtering, the text states cellranger count pipeline provided by 10x Genomics (version X.Y.Z). The version should be written out not as version X.Y.Z.

Response: We have corrected this in the text.

In the text "Cell Bender (version X.Y.Z)" the version should be written out, not as version X.Y.Z.

Response: We have corrected this in the text.

"For highly_variable_genes function in scanpy (version Y.Y.Y), the version should be written out

not as y.y.y.

Response: We have corrected this in the text.

For the curated “Human DNA Repair Genes” list by R. Wood and M. Lowery, the list should be provided or the reference to the source.

Response: We have now listed all genes in supplementary tables 1 to 7.

Nrf2 Target Gene Set: This set includes 37 genes identified as Nrf2 targets, based on a curated list from a published review article. The authors should list the genes.

Response: We have now listed all genes in supplementary tables 1 to 7.

For the RNAscope section in methods, the tense of methods should be written out, see “If combining RNAscope with immunostaining for various antibodies is needed, the standard immunohistochemistry (IHC) protocol should be followed.”

Response: We have rewritten this methods section.

Please write out the names as much as possible for example, write out DNA Damage Response each time rather than DDR and double strand breaks rather than DSB, neural progenitors rather than NPs, Neural Stem Cells rather than NSCs, reactive oxygen species rather than ROS, Unfolded Protein Response rather than UPR, excitatory neurons rather than ENs, integrated Stress Response rather than ISR, ventricular zone rather than VZ. Many readers outside the field may not be familiar with the jargon used and it would make the manuscript easier to read to write out the words in full each time.

Response: We have corrected this in the text by spelling out abbreviations in full.

All the mouse lines should be described in the methods. For example, in figure 1a and extended data fig 1 the mouse line LSL-H2B-GFP was used; however, it’s not defined what LSL is in the text of the manuscript, the figure legend, and the source for this mouse line should be provided in methods.

Response: We have corrected this in the text. All detailed animal information is now provided in the methods.

For the line “or p53 related genes upregulated in NPs in the absence of Atf4, such as Cdkn1a Eda2r,” there should be a comma after Cdkn1a.

Response: We have corrected this in the text.

For the line, “We also examined the RG sequencing data”. Please define what RG stands for.

Response: We have corrected this in the text.

In the UMAP plot there are clusters RG1, RG2, RG3, RG4. Please define in the manuscript or the figure legend what RG stands for? Similarly it should be defined in the figure legend or manuscript what Ext1 and Int1 clusters refer to.

Response: All of these have now been defined in the figure 4 legend and new extended data fig. 5a.

Overall these are minor points and the paper is outstanding and I would be supportive of publication if the authors address them.

We thank the reviewer for these positive and encouraging comments.

Referee #3 (Remarks to the Author):

Expansion of outer cortical layer Cux2+ neurons required selective adaptations for DNA repair

Summary/Key Results:

The authors highlight that the expansion of the cortex, especially of Layers 2/3, correlates with advances in cognition throughout evolution. This rapid expansion during embryonic development subjects these neural progenitors to replication stress and oxidative damage/ROS stemming from elevated cellular metabolism. While neural progenitors of the cortex likely have to overcome these insults, little is known about the maintenance of genomic integrity of the neural progenitor pool. Furthermore, whether progenitors are fate restricted to become L2/3 neurons or influenced by niche and birthdate remains unknown. L2/3 neurons are less viable in multiple sclerosis (MS) and disproportionately affected in autism, suggesting a unique vulnerability of this population. Thus, the authors investigate what conditions lead to this selective vulnerability. They identify that knockout of activating transcription factor 4 (Atf4) in all embryonic precursors of upper layer cortical neurons leads to increased DNA double stranded breaks (DSBs) and apoptosis selectively in L2/3 neurons, supporting a unique vulnerability of this population. The authors demonstrate that L2/3 neuronal loss in Atf4 knockout (KO) mice is p53 dependent. Finally, they demonstrate that Atf4 KO leads to changes in transcripts for the DNA damage response which are capable of partially rescuing L2/3 cell death when upregulated. The authors identify a novel role for Atf4 not only in regulation of L2/3 cortical expansion, but also as a transcriptional regulator of DDR. Furthermore, the authors conclude that neural progenitors (NP) of L2/3 are prepatterned, rather than influenced by their niche and birthdate. Thus, the evolutionary expansion of the cortex is facilitated by the recruitment of DNA repair proteins in select progenitors which give rise to L2/3 neurons.

The authors present convincing results demonstrating the susceptibility of L2/3 neurons to Atf4 KO, indicating an essential role of Atf4 to neural progenitor survival, and they clearly demonstrate that the selective loss of Atf4 knockout neurons is p53-dependent. Beyond these findings, however, many of the authors' conclusions are not convincingly substantiated, which detracts from the significance of this work.

Major issues include lack of substantial evidence for the mechanisms contributing to DNA damage and L2/3 susceptibility to apoptosis, mechanisms that underly the dependence of L2/3 on ATF4, lack of direct evidence for the evolutionary relevance of this pathway to humans/its role in cortical expansion, and lack of data demonstrating the consequences of the observed L2/3 cortical phenotype on brain function and behavior.

Major Concerns

General

1. The disappearance of L2/3 neurons in ATF4 knockouts looks very convincing as is its dependence on P53-induced apoptosis. However, the claim that this is due to disrupted ATF4-dependent DNA damage repair (DDR) resulting in DSB and apoptosis can be challenged for several reasons:

- protein translation is severely disrupted in ATF4 KO, which can suffice to induce DSB, DDR down-regulation, and apoptosis.

Response: We thank the reviewer for these comments which have helped us to significantly strengthen the findings of the paper.

1. We assessed whether translation was altered in neural progenitors in the absence of Atf4 at E11.5. Gene set score analysis of the eukaryotic translation factor gene set (69 genes, including initiation, elongation, termination, and ribosome recycling factors) revealed no significant changes in activity (NEW Extended Data Fig. 5a). To directly assess protein synthesis, we performed O-propargyl-puromycin (OPP) labeling on acutely isolated neural progenitor cells from E11.5 Atf4

fl/fl (control) and Emx1-Cre:Atf4fl/fl embryos. Relative OPP intensity showed no difference between Atf4-deficient and control neural progenitors, indicating that global translation rates were unchanged despite the observed increase in DNA damage (NEW Extended Data Fig. 5b, c, d).

2. We now provide new mechanistic data that demonstrates that the DNA Damage Response is disrupted at its earliest stage at the onset of corticogenesis in a subset of radial glia in E11 cortex in Emx1cre:ATF4fl/fl. Ataxia-Telangiectasia Mutated (ATM) is a central regulator of DNA double strand break repair and, once activated by DNA Double Strand Breaks (DSB), goes on to phosphorylate many downstream factors to initiate the cascade of double strand repair. Phosphorylation of ATM at serine 1981 is a key event for sustained retention of ATM at double strand DNA breaks and also for its ability to phosphorylate its downstream targets to trigger the cascade of DSB repair. We show that the phosphorylation of ATM Serine1981 is very significantly disrupted in a subset of radial glia in Emx1cre:ATF4fl/fl at E11. The proliferation and the number of radial glia at E11 was unaltered in the absence of Atf4 (NEW Extended Data Fig. 3m, n, o, p), but this loss of ATM phosphorylation was associated with significantly increased γ H2AX punctae. This suggests a fundamental failure at the initiation of double strand break DNA repair in neural progenitors lacking Atf4. This is shown in NEW Fig. 2t and 2u, and discussed in the results and discussion.

3. We now also demonstrate a direct link between Atf4 transcriptional activity, failure of ATM phosphorylation, and DNA repair failure. Cold Inducible RNA binding protein (Cirbp) has been shown in cultured HeLa cells to be important for phosphorylated ATM association with chromatin at DNA double strand breaks (*PNAS* 2018; 115 (8) E1759-E1768). Interestingly, high Cirbp expression and species-specific structure has also been implicated recently in the bowhead whale as responsible for the incredibly accurate and efficient DNA repair seen in this species, which allows for its long and cancer-free lifespan (*Nature* 2025 October). We have shown that Cirbp is a direct transcriptional target of Atf4 in the DNA Damage Response in neuronal progenitors. We now show that knockdown of Cirbp in vivo in cortical progenitors using in-utero electroporation leads to disrupted phosphorylation of ATM at Serine-1981. We used in utero electroporation (IUE) of Cirbp shRNA (in a GFP expression plasmid) to knock down Cirbp in wild-type (WT) brain cortex, and observed a very significant reduction in the total number of radial glia with phosphorylated ATM (Ser1981) (compared to IUE with control empty vector) and also a very significant reduction in the proportion of GFP and phospho-ATM (Ser1981) double positive cells relative to the total phospho-ATM(Ser1981)+ population along the ventricular surface (NEW Fig. 5r, s, t).

4. We also performed Cirbp shRNA IUE and assessed the tissue at later times for an effect on cortex Layer 2/3 thickness (NEW Fig. 5u, v, w). We found that knock down of Cirbp alone in vivo led to significant cortical layer 2/3 thinning, highlighting the importance of this factor in corticogenesis. Our data shows that Cirbp knockdown in vivo in developing cortex mimics the phenotype of Atf4 loss in neuronal progenitors, both in terms of disrupted ATM phosphorylation in radial glia and subsequent thinning of layer 2/3. We provide the first functional assessment of Cirbp in vivo in the nervous system. We have also added an entire new discussion section on the emerging role of Cirbp.

- although ROS is referred to as a candidate DSB mechanism, there is no evidence for higher oxidative stress in L2/3 NPs relative to other neurons at a comparative proliferation stage
many established DDR genes are up-regulated

Response: It is generally accepted that the cortical progenitors destined for layer 2/3, as they are the last to emerge from the ventricular zone, spend longer in the ventricular zone and so are exposed to ROS for a greater length of time and are exposed to a greater cumulative oxidative stress and DNA damage load. We have discussed this in the text and added two references for this.

- there is no direct demonstration that the candidate DDR genes (Uba52, Cirbp, and Ebf1) can also induce L2/3 cortical thinning

Response: We have now performed in utero electroporation with Cirbp shRNA or Uba52 shRNA for in vivo knockdown of either Cirbp or Uba52 in developing embryonic cortex, and assessed for subsequent L2/3 thinning. We found significant thinning of GFP+ layer 2/3 thickness as well as significant reductions in Cux1/2 and GFP+ cells when either of these factors was knocked down in vivo. This is shown in NEW Fig. 5u-w and NEW Extended data Fig. 8 m-p, and discussed in the text.

- these genes have many functions other than DDR, and most are not DDR genes themselves but regulators of DDR.

Response: Many factors and accessory factors related to DNA repair are upregulated in neuronal progenitors lacking Atf4. This is because double strand DNA repair fails fundamentally at its initiation due to a failure to phosphorylate ATM (as now discussed in Point 1 above). The cells make an attempt to compensate for this failure by upregulating many factors related to the DNA Damage Response, but are unable to overcome the failure of ATM phosphorylation. We have now clarified this in the text.

2. ATF4 is ubiquitously expressed in neurons, especially in interneurons. Mechanisms for the particular vulnerability of L2/3 neurons to ATF4 KO are not demonstrated and the underlying causes of their vulnerability remain unknown. It is not clear, for example, whether ATF4 is needed for the expression of the aforementioned DDR-related genes only in L2/3, or, it also controls them in other neuron types, which tolerate these effects better.

Response: We have evaluated a publicly available single cell Nuc-seq dataset for ATF4 expression in different cell types in E18 cortex. This analysis is shown as NEW Extended data Fig. 3a. We agree that Atf4 is more widely expressed, suggesting that other neuron types tolerate its loss better than Layer 2/3.

3. This mouse model offers a unique opportunity to establish the significance of L2/3 loss. However, the consequences of this loss on brain function and behavior are completely unexplored.

Response: We agree with the reviewer that this mouse could be of use for assessing behavior changes and learning more about the function of layer 2/3 neurons and their contribution to brain function. Layer 2/3 functions however are very complex, and this is beyond the scope of this current study, which is to understand their developmental vulnerability. We agree however that this is a very interesting area for future research.

Significance

1. L2/3 structural and functional deficits have been linked to a range of neurodevelopmental, neurodegenerative, and psychiatric (schizophrenia in particular). The association with MS has been the least well established, with some reports highlighting L5 rather than L2/3 pathologies. It remains unclear why only MS and autism are mentioned and why there was no effort to make

associations to disease-relevant phenotypes, especially during adolescence, when much of the cortical analyses were performed.

Response: We apologize for this omission. We agree that Layer 2/3 pathologies have been linked to other neurodevelopmental and psychiatric conditions. We have added references about dysregulation of these layers in schizophrenia and epilepsy. In addition, recent high impact papers have implicated Layer 2/3 pathologies and dysregulation in human ageing (*Nature*. 2025. September 3rd), head trauma in young athletes (*Nature*. 2025. September 19th), Frontotemporal dementia (*PNAS*. 2025 Mar 4;122(9)) and Alzheimer's (*Cell*. 2025. Volume 188, Issue 18P4980-5002.). It is becoming clear that this population of cortical neurons is fundamentally affected in many human disorders. We have now added text to the introduction and an entire new discussion section describing the importance of L2/3 dysfunction to a multitude of neurological conditions.

2. In studies that do report L2/3 changes, they do not appear homogeneously distributed throughout the cortex. Regional differences are not addressed in the paper, and it is not clear in which cortical area were the measurements performed.

Response: We have added data and text clarification to explain that we do see these L2/3 changes homogeneously throughout the cortex. Please see NEW Extended Data Fig.2 and text clarification in the results section. "All analyses were performed in the primary somatosensory cortex using level-matched coronal brain sections aligned to the Allen Mouse Brain Atlas, but we also confirmed a similar selective loss of Cux2+ neurons in Emx1-cre:Atf4fl/fl throughout the cortex at different rostral and caudal levels (Extended Data Fig. 2)".

Methodology

1. In Figure 2, the authors use γ H2AX and pKAP1 to label DNA DSBs. While this provides clear evidence of DNA break induction, the amount of γ H2AX+ cells in representative images appears too sparse to induce the extreme atrophy observed in L2/3 of Atf4 KO mice, nor does it constitute "overwhelming DNA damage" as described. For example, in figure 2g, only 30% of Pax6+ cells are γ H2AX+. Furthermore, evidence is needed to support the claim that DNA DSBs are the cause of cell death, or further investigation into other mechanisms inducing apoptosis is necessary.

Response: The pan nuclear staining of γ H2AX and pKap1 that the reviewer refers to represents cells actually in the process of undergoing apoptosis due to overwhelming DNA damage (*Nat Rev Cancer*. 2008 Dec;8(12):957-67). To assess DNA damage more accurately it is necessary to assess the number of γ H2AX punctae in the nuclei of cells that do not have pan nuclear staining. This showed very significant increases in punctae for γ H2AX and 53BP1 (which colocalized) in neural progenitors at E12.5 from Emx1-cre:Atf4fl/fl compared to controls (NEW Fig.2q, r, s). This is clarified in the text and added figures, and the above reference is added.

2. Figure 4j indicates the upregulation of many DDR related GO terms in the Atf4 KO. While this demonstrates alteration of the DNA damage response in the KO model, as stated by the authors, it weakens the claim that Atf4 is needed to facilitate DDR.

Response: Response is the same as above for Point 1. Many factors related to DNA repair are upregulated in neuronal progenitors lacking Atf4. This is because double strand DNA repair fails at its initiation due to disruption of ATM phosphorylation (as now discussed in Point 1 above). The cells make a futile attempt to compensate for this failure by upregulating many factors related to the DNA Damage Response, but are unable to overcome the failure of ATM phosphorylation.

Conclusions

1. The conclusion that impaired DDR is the main cause of excessive DSB and apoptosis is not directly demonstrated and alternatives (eg inhibited translation) are a strong alternative

Response: The response is the same as above for section 'General Point 1.' We now show that the Atf4 transcriptional target Cirbp is required in vivo for the proper phosphorylation of ATM in neuronal progenitors. We have added an entire new discussion section on the emerging role of Cirbp.

2. The final claim that evolutionary expansion of the cortex requires recruitment of DNA repair proteins is unsubstantiated. In figure 2b-c, the authors demonstrate expression of Atf4 at GW15 and 17 in human tissue. However, the presence of Atf4 is not enough to claim that Atf4 driven DNA damage response contributes to evolutionary cortical expansion.

Response: We agree and have rewritten this discussion section in a more speculative way. We have added discussion about the very recent findings that the level of expression and species specific structure of Cirbp appears to have been an evolutionary adaptation in animals requiring exquisitely robust DNA repair (*Nature* 2025 October).

3. The authors claim that the evidence presented suggests that Cux2+ NP are pre-patterned to become L2/3 cortical neurons. Despite a significant number of Cux2+gH2AX+ NPs (60%) following Atf4 KO, suggesting a unique vulnerability of this population, there is not enough evidence to claim they are predestined.

Response: We agree and have removed this section from the discussion.

Minor Concerns

1. The rationale for investigating Atf4 is unclear. Although Atf4 is known to regulate several cellular stress response pathways, the authors openly state that there is little evidence for Atf4 in the regulation of genomic stability.

Response: We try to provide better rationale for why we looked at Atf4 function. We have tried to clarify this better in the text. Cux2 neurons showed selective loss in MS, accompanied by significant dysregulation of Atf4 (companion paper). We initially questioned whether Atf4 functioned in their developmental resilience and assumed this would be via roles in ISR or UPR. The single nuclei RNA seq suggested DNA repair was altered rather than ISR or UPR.

2. The sentence in the introduction stating DNA DSBs are the most common lesion in neurogenesis is not cited.

Response: We have added a reference for this in the introduction.

Referee #4 (Remarks to the Author):

I co-reviewed this manuscript with one of the reviewers who provided the listed reports.

Response: We very much thank the reviewer for co-reviewing this.

Referee #5 (Remarks to the Author):

I co-reviewed this manuscript with one of the reviewers who provided the listed reports.

Response: We sincerely thank the reviewer for their co-review.

Response to Reviewers

We sincerely thank all reviewers for their constructive and insightful comments. Below is our response to the remaining minor concerns of referee 3:

Referee 3:

Minor concerns:

1. How do the authors rationalize the increase in DDR genes? Especially since ATM is downregulated, what is mediating the DNA damage response?

This was somewhat addressed in response to reviewers, Methodology point 2. I believe it could be expanded.

Response: We thank the reviewer for the comment. The activated genes of the DDR are mostly accessory factors in *Emx1-cre:Atf4^{fl/fl}*, and genes critical for double strand DNA repair such as *Cirbp* are downregulated. There are many other arms of the DNA Damage Response that may attempt to compensate for the loss of phospho-ATM to rescue the cell. For instance the mechanisms that sense DNA damage may upregulate alternative arms such as Ataxia telangiectasia and Rad3 related (ATR) kinase, which is a master regulator of single-stranded DNA break repair. This will be accompanied by significant upregulation of accessory factors in the DDR, but will ultimately fail to repair the double strand DNA breaks in the absence of phosphorylated ATM. Additionally activated P53 is also known to upregulate factors in the DDR, to attempt to rescue the cell. We have added an additional explanation to the text of the manuscript about this.

2. In Fig2, ATM labeling does not overlap with γ H2AX labeling. Is that a concern?

Response: This would be our expectation. In cells with phosphorylated ATM labeling, DNA damage will be very quickly repaired. In cells that are unable to phosphorylate ATM (no ATM labeling) there will be massive DNA damage marked by γ H2AX. Note that γ H2AX can be activated in the absence of phospho ATM by other phosphoinositide 3-kinase related protein kinases (PIKKs) such as ATM-and Rad3-related (ATR), and/or DNA dependent protein kinase (DNA-PKcs), but DNA double strand repair will still fail in the absence of phospho-ATM.

3. How was the statistics for snRNAseq performed? There are n = 3 biological replicates, but it is unclear whether this was accounted for using pseudobulk or other approaches in their analyses?

Response: Thank you for raising this point. Differential expression analysis was performed at the single-nucleus level using the Wilcoxon rank-sum test as implemented in Scanpy. In this analysis, individual nuclei were treated as independent observations, and biological replicates (n = 3) were not explicitly modeled in the statistical testing. We acknowledge that this approach does not directly account for replicate-to-replicate variability and may therefore increase statistical power at the expense of conservative inference. However, the Wilcoxon test was used here as an exploratory method to identify robust, cell-type-associated transcriptional differences, and key findings were supported by consistency across biological replicates and downstream analyses. We have clarified this analytical choice and its limitations in the Methods section.